# AXIS: EXPLAINABLE TIME SERIES ANOMALY DETECTION WITH LARGE LANGUAGE MODELS

## ABSTRACT

Time-series anomaly detection (TSAD) increasingly demands explanations that articulate not only if an anomaly occurred, but also what pattern it exhibits and why it is anomalous. Leveraging the impressive explanatory capabilities of Large Language Models (LLMs), recent works have attempted to treat time series as text for explainable TSAD. However, this approach faces a fundamental challenge: LLMs operate on discrete tokens and struggle to directly process long, continuous signals. Consequently, naïve time-to-text serialization suffers from a lack of contextual grounding and representation alignment between the two modalities. To address this gap, we introduce AXIS, a framework that conditions a frozen LLM for nuanced time-series understanding. Instead of direct serialization, AXIS enriches the LLM's input with three complementary hints derived from the series: (i) a symbolic numeric hint for numerical grounding, (ii) a context-integrated, step-aligned hint distilled from a pretrained time-series encoder to capture fine-grained dynamics, and (iii) a task-prior hint that encodes global anomaly characteristics. Furthermore, to facilitate robust evaluation of explainability, we introduce a new benchmark featuring multi-format questions and rationales that supervise contextual grounding and pattern-level semantics. Extensive experiments, including both LLM-based and human evaluations, demonstrate that AXIS yields explanations of significantly higher quality and achieves competitive detection accuracy compared to general-purpose LLMs, specialized time-series LLMs, and time-series Vision Language Models. The code is available in `https://anonymous.4open.science/r/TimeSemantic-1742/main.py`

## 1 INTRODUCTION

Time Series Anomaly Detection (TSAD) is essential for safeguarding critical systems across domains (Iqbal et al., 2019; Zeufack et al., 2021; Hundman et al., 2018). While deep learning models can detect anomalies with high accuracy (Fig. 1(a)), their adoption in real-world systems is limited by two challenges. First, their reasoning process is essentially a black box. Experts remain in the dark when asking the most practical question: why was this anomaly flagged? Post-hoc attribution methods such as SHAP (Fig. 1(b)) attempt to fill this void, but they merely repackage correlations into statistical features. Such attributions reveal what inputs influenced the model, but they stop short of offering why the underlying anomaly event occurred. Second, these models are brittle. Trained narrowly - often on a single dataset - they capture dataset-specific features rather than generalizable patterns. In fast-changing environments where anomalies manifest in diverse forms, this rigidity is crippling: models must be

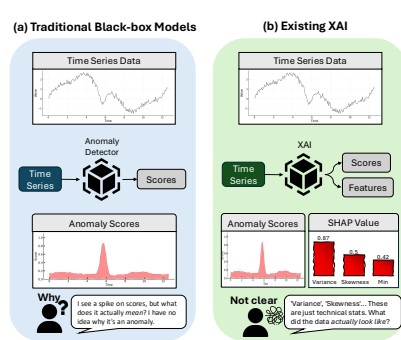

Figure 1: Deep learning method for TSAD: (a) Opaque anomaly scores fail to explain why; (b) XAI features lack intuitive semantics;

retrained at high cost, yet still fail to transfer across domains. What is missing is the ability to handle anomalies universally—to recognize and adapt across diverse failure patterns without exhaustive retraining.

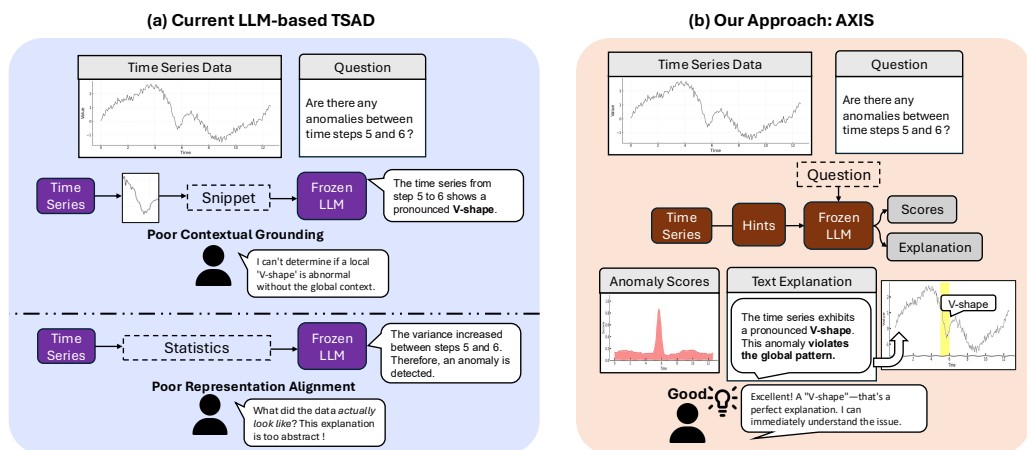

Figure 2: Bridging the Semantic Gap in Time Series Anomaly Explanation. (a) Current LLM-based methods fail due to: (i) poor **Contextual Grounding**, where observing a local pattern (e.g., the "V-shape") in isolation prevents a meaningful diagnosis; and (ii) **Representation Misalignment**, where inputs of abstract statistics (e.g., "variance increased") lead to uninformative, circular explanations. (b) Our approach overcomes these limitations by producing contextualized, pattern-level explanations that align with expert reasoning.

In response to these limitations, the community has turned to Large Language Models (LLMs), celebrated for their fluency and generalization. Yet critical obstacles remain: LLMs operate on discrete tokens, making them ill-suited for the long, continuous nature of time series (Dong et al., 2024). Attempting to fit these signals into tokenized inputs often incurs lossy serialization, forcing workarounds that undercut the LLM's capabilities. Common strategies—feeding isolated fragments or pre-aggregated statistics—reduce the model to a mere post-hoc translator. As our motivating example illustrates (Fig. 2(a)), this naive approach, however, suffers a critical semantic gap, driven by two fundamental failures.

The first is a lack of **Contextual Grounding**. By analyzing only a narrow snippet of the series, the LLM is deprived of the broader temporal context required to discern whether a local pattern is genuinely anomalous or merely a benign fluctuation. The second is a failure of **Representation Alignment**, which creates a chasm between the model's analytical basis and human intuition. When an LLM is fed abstract statistical summaries instead of the data's intrinsic shape, its explanations degenerate into shallow echoes of its inputs, failing to provide the qualitative, pattern-level insights that domain experts require to understand what truly happened in the data.

Overcoming these failures requires a paradigm shift. Explanations must move beyond statistical paraphrasing toward a native integration of temporal dynamics and linguistic reasoning. This reduces to two core challenges: the **Contextual Grounding Challenge**, which demands interpreting local events in the context of the full series to explain not only what the data looks like but **why** it is abnormal; and the **Representation Alignment Challenge**, which requires bridging the semantic gap between low-level numerical signals and the rich, shape-based concepts underlying human reasoning.

In this paper, we introduce AXIS, a framework designed to address these challenges and unlock the explanatory potential of LLMs for TSAD. Our approach rest on two synergistic contributions. First, to establish the necessary semantic foundation, we construct a novel benchmark with pattern-level labels and rich contextual cues, providing the semantic foundation essential for both grounding and alignment. Second, at the core of our framework is a Hint Tuner that systematically tackles both challenges. For contextual grounding, it distills global time-series information into a compact, informative "hint." For representation alignment, it maps this temporal hint into the LLM's native semantic space. This integrated design transforms a frozen, general-purpose LLM into a context-aware diagnostic expert, capable of generating correct and high reasonal quality answers for TSAD, as illustrated in Fig. 2(b). In summary, our main contributions are threefold:

- **A Benchmark for Semantic Explanations:** To bridge the "semantic gap" between raw time series signals and linguistic concepts, we construct the first benchmark dedicated to semantic time series anomaly explanation. This benchmark ensures both anomaly diversity and explanation fidelity, providing a principled testbed for evaluating the semantic explainability of TSAD.

- **A Novel Cross-Modal Alignment Framework:** We present AXIS, a framework that aligns a frozen LLM with time-series dynamics. It conditions the LLM on three synergistic inputs: a symbolic numeric hint for numerical grounding, a context-integrated step-aligned hint for fine-grained dynamics, and a task-prior hint for global task priors.

- **Extensive Empirical Validation:** Comprehensive experiments show that AXIS establishes a new state of the art in semantic anomaly explanation, substantially outperforming strong baselines including general LLM, specialized time-series LLM, time series VLM.

## 2 RELATED WORK

**State-of-the-Art TSAD Models and Interpretability Challenges**   Classical and statistical methods remain competitive baselines yet produce pointwise scores with weak semantics and limited support for multivariate structure (Liu et al., 2008; Yeh et al., 2016). Deep TSAD improves accuracy via reconstruction (Audibert et al., 2020; Su et al., 2019; Zhang et al., 2019), prediction-residual modeling (Tuli et al., 2022), and attention-centric architectures (Xu et al., 2021; Yang et al., 2023; Shen et al., 2020; Lan et al., 2025), with recent work exploring unified/foundation-style formulations (Shentu et al., 2024; Gao et al., 2024) and diffusion-based detectors (Wang et al., 2025b). Explanations, however, are largely post hoc and tied to low-level contributions (e.g., time-step or feature importance), limiting mechanism-oriented diagnosis. Time-series XAI extends attribution (Bento et al., 2021) and investigates prototype/shapelet/motif views and counterfactual recourse (Bahri et al., 2022; Yeh et al., 2016), but explanations remain grounded in signal-level statistics rather than pattern-level concepts. This motivates treating TSAD explainability as a semantic alignment problem.

**Large Language Models for TSAD**   LLMs can function as zero-shot anomaly detectors under appropriate prompting and input scaling (Alnegheimish et al., 2024; Dong et al., 2024; Zhou & Yu, 2024; Wang et al., 2025a). Performance degrades on long-horizon series due to context-length limits, lossy time-to-text serialization, and chunked inference, which together induce memory decay and boundary artifacts. A complementary line uses LLMs as post-hoc reasoners that verbalize anomaly scores, SHAP attributions, or raw subsequences, or coordinate multi-agent annotation (Liu et al., 2025; Lin et al., 2024). In both paradigms, LLMs act mainly as summarizers of low-level signals, yielding descriptive rather than semantically grounded explanations. Our approach directly aligns temporal representations with language via soft-prompt-based conditioning, aiming for faithful, pattern-level explanations.

**Benchmarks for TSAD**   Existing benchmarks for time-series question answering, which are adjacent to our task, can be broadly categorized into two paradigms. The first relies on fully synthetic data generation, where normal time series are composed from trends, seasonality, and noise, after which localized anomalies are injected to generate templated or LLM-augmented labels (Cai et al., 2024; Xie et al., 2024; Wang et al., 2025a; Kong et al., 2025). The second paradigm uses real-world datasets, pairing authentic time-series data with corresponding semantic information to create evaluation suites (Kim et al., 2024; Cai et al., 2025; Liu et al., 2024a; Williams et al., 2024; Chen et al., 2025). However, synthetic benchmarks often lack the contextual richness required for robust grounding and representation alignment, while real-world data yields domain-specific explanations that limit model generalizability. To our knowledge, a dedicated benchmark for semantic time series anomaly explanation has remained a critical gap, which our work directly addresses by introducing a benchmark designed for this task.

## 3 METHODOLOGY

This section presents our AXIS framework for the semantic anomaly explanation task. We begin by formalizing the task in Sec. 3.1. Next, we introduce the core architecture in Sec. 3.2. Sec. 3.3

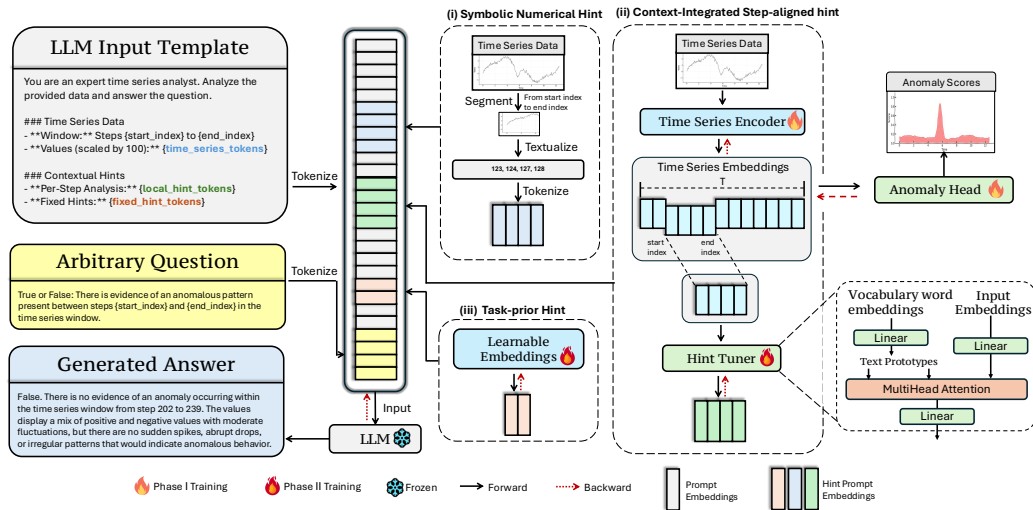

Figure 3: AXIS constructs the prompt by three representation pathways: (i) symbolic numeric grounding via window values, (ii) context-integrated local dynamics through step-aligned hints to capture contextual information, and (iii) task-prior hints encoding global priors.

describes the two-phase training paradigm—encoder pretraining followed by hint tuning with the LLM frozen—and the inference procedure. Finally, to enable systematic supervision, we synthesize a benchmark with pattern-level annotations in Sec. 3.4.

## 3.1 PROBLEM FORMULATION

Conventional TSAD methods typically output point-wise anomaly scores for a series of length $T$, but such signals rarely provide human-understandable insights. In practice, anomalies often span contiguous intervals rather than isolated timestamps, and users are chiefly concerned with understanding **why** an interval is anomalous. To address this, we reformulate the task by introducing a target interval $(s, e)$ and defining the goal as generating a natural-language explanation for it. This window-based formulation respects the temporal continuity of anomalies and makes the explanation task well-posed by localizing reasoning to a specific region within the series. We formalize the problem as follows:

> Semantic time series anomaly explanation
>
> Given a univariate time series $\mathbf{x}_{1:T} \in \mathbb{R}^T$ and a natural-language query $q$, the objective is to explain the pattern within an interval $[s, e]$; in our setup, $(s, e)$ is provided as input. The model learns a mapping $\mathcal{G}$ that, while conditioning on the entire series $\mathbf{x}_{1:T}$ to leverage global context, generates an explanation $\mathbf{y}$ for the target window:
>
> $$\mathcal{G} : (\mathbf{x}_{1:T}, q, s, e) \mapsto \mathbf{y}.$$

## 3.2 AXIS FRAMEWORK

We now propose our novel framework called AXIS for semantic anomaly explanation task. AXIS conditions a frozen LLM through three representation pathways: symbolic numeric hint, context-integrated step-aligned hint, and task-prior hint. The overall framework is shown in Fig. 3. We instantiate this conditioning through three pathways that jointly provide numeric grounding, step-aligned dynamics under global context, and compact task-level priors, without expanding the context length or modifying the LLM.

**Symbolic Numeric Hint.** LLMs possess native reasoning capabilities over discrete numerals when presented symbolically, even in zero-shot settings. To exploit this capability without exhaust-

ing the context budget, we textualize only the target window $[s, e]$ after Z-score normalization of the full series $\mathbf{x}_{1:T}$. Values are scaled by a factor $r$ (default $r=100$) to preserve precision while avoiding decimal tokens (Liu et al., 2024b), rounded to integers, and serialized as a delimiter-separated string (e.g., "123, 124, 127, 128") to constitute {time_series_tokens}. This pathway is compact—its position cost scales as $\alpha(e - s) + c$ where $\alpha$ is the average subword tokens per integer and $c$ the delimiter overhead—yet it preserves step-wise numeric grounding.

**Context-Integrated Step-aligned Hint.** While the above textualization provides direct numeric access, it cannot capture long-range dependencies essential for TSAD such as regime shifts, seasonality interactions, and boundary effects. We therefore condense global information into step-aligned local representations via a pretrained time-series encoder and a *Hint Tuner*, in the spirit of (Jin et al., 2023). A Transformer encoder $f_\theta$ consumes $\mathbf{x}_{1:T}$ and outputs embeddings $\mathbf{H}_{1:T} \in \mathbb{R}^{T \times d_{\text{proj}}}$ where $d_{\text{proj}} = 256$ is the projection dimension (the details are shown in Appx. B.1). We slice $\mathbf{H}_{s:e}$ and map it into the LLM space using a multi-head cross-attention mechanism over a prototype bank derived from the LLM vocabulary. The prototype bank is defined as $\mathbf{S}_{\text{proto}} = \mathbf{M}\mathbf{E}_{\text{vocab}} \in \mathbb{R}^{P \times d_h}$, where $\mathbf{E}_{\text{vocab}} \in \mathbb{R}^{|\mathcal{V}| \times d_h}$ denotes fixed word embeddings and $\mathbf{M} \in \mathbb{R}^{P \times |\mathcal{V}|}$ is a learnable linear projection ($P = 1024$). Specifically, we employ scaled dot-product attention with $H_{\text{tuner}} = 8$ heads. The queries, keys, and values are computed as: $\mathbf{Q} = \mathbf{H}_{s:e}\mathbf{W}_q$, $\mathbf{K} = \mathbf{S}_{\text{proto}}$, $\mathbf{V} = \mathbf{S}_{\text{proto}}$, where $\mathbf{W}_q \in \mathbb{R}^{d_{\text{proj}} \times d_h}$ projects the encoder output to the LLM hidden dimension ($d_h = 4096$). The attention output is then:

$$\tilde{\mathbf{H}}_{s:e} = \text{MultiHeadAttn}(\mathbf{Q}, \mathbf{K}, \mathbf{V}) \in \mathbb{R}^{(e-s) \times d_h}.$$

The resulting $\tilde{\mathbf{H}}_{s:e}$ acts as step-aligned local hints that inject global context and temporal structure into the LLM embedding space while keeping both the LLM and $f_\theta$ frozen. This pathway adds $(e - s)$ positions linearly while supplying detailed global context and temporal alignment.

**Task-Prior Hint.** To regularize decoding and inject task-level priors that remain stable across instances, we introduce a small set of shared queries $\mathbf{P}_{\text{fix}} \in \mathbb{R}^{K \times d_h}$ that attend to the same prototype source: $\tilde{\mathbf{F}} = \text{Attn}(\mathbf{P}_{\text{fix}}, \mathbf{S}_{\text{proto}}, \mathbf{S}_{\text{proto}}) \in \mathbb{R}^{K \times d_h}$.

**Final Prompt.** The three hint pathways are integrated into a unified prompt that conditions the frozen LLM, as illustrated in Fig. 3. The final input sequence is constructed from a template containing the user's query $q$, the textualized window values, and special placeholder tokens. At input time, the embeddings for the $K$ task-prior hints ($\tilde{\mathbf{F}}$) and the $(e-s)$ step-aligned hints ($\tilde{\mathbf{H}}_{s:e}$) replace the embeddings of $K + (e-s)$ placeholder tokens. The symbolic numeric hint is inserted directly as text. This process yields a single, coherent input sequence for the LLM that combines natural language with rich, multi-faceted temporal information, all without requiring architectural changes to the base model.

## 3.3 TRAINING OBJECTIVE

Our method is trained in two phases. First, we pretrain the time-series encoder $f_\theta$ using a joint objective that combines masked reconstruction and anomaly classification to learn robust temporal representations. The total loss is defined as:

$$\mathcal{L}_{\text{pretrain}} = \lambda_{\text{recon}}\mathcal{L}_{\text{recon}} + \lambda_{\text{class}}\mathcal{L}_{\text{class}},$$

where $\mathcal{L}_{\text{recon}}$ is the Mean Squared Error (MSE) over masked timesteps, $\mathcal{L}_{\text{class}}$ is the Binary Cross-Entropy (BCE) loss, and we set $\lambda_{\text{recon}} = 1.0$ and $\lambda_{\text{class}} = 1.0$.

In the second phase, we freeze both the encoder and the LLM, training only the Hint Tuner and its associated parameters ($\mathbf{M}, \mathbf{P}_{\text{fix}}$). This phase optimizes a standard next-token prediction objective:

$$\mathcal{L}_{\text{LM}} = -\sum_{i=1}^{N} \log P(y_i \mid y_{<i}, q, \tilde{\mathbf{H}}_{s:e}, \tilde{\mathbf{F}}),$$

where $\mathbf{y}$ is the target explanation sequence. This two-phase strategy maintains computational efficiency by training only the lightweight Hint Tuner while preserving the time series encoder's pretrained capabilities. The decoupled design enables stable training by separating temporal representation learning from cross-modal alignment. The detailed training process and additional hyperparameters are given in Appx. B.2 and B.3 .

## 3.4 A BENCHMARK FOR SEMANTIC TIME SERIES ANOMALY EXPLANATION

Existing methods reveal a foundational limitation: the community lacks a benchmark that teaches models to speak the language of temporal patterns. To address both the contextual grounding and representation alignment challenges outlined earlier, we synthesize a benchmark specifically designed to train models to reason about anomalies like human experts. Rather than a mere collection of time series, our benchmark constitutes a carefully curated curriculum built around three core design principles.

**Pattern-Level Anomaly Vocabulary.** To address the representation alignment challenge, we introduce a procedural engine that moves beyond abstract statistical deviations to a vocabulary of interpretable, pattern-level anomalies. As illustrated in Figure 4, our engine synthetically composes canonical anomaly primitives—such as *sudden spikes*, *level shifts*, and *periodicity breaks*—onto clean baseline series. A key advantage of our approach is the generation of **paired time series**: for every abnormal series created, a corresponding normal counterpart is preserved. This methodology establishes an unambiguous, verifiable link between anomaly time series and its linguistic label, forming the bedrock for teaching models to reason about the semantics of temporal events.

**Contextual and Comparative Reasoning.** To overcome the contextual grounding challenge, we designed our benchmark to compel models to reason about local events within a global and comparative framework. Naively presenting isolated time-series windows is insufficient. Instead, our engine first generates a **global descriptor**, a textual summary of the series' overall dynamics (e.g., trends, seasonality), which provides essential context. Second, we employ a comparative windowing strategy. A model is presented not only with a window containing a potential anomaly but also with the corresponding temporal window from its "healthy" paired series. This core design choice is a significant advantage, as it inherently frames the task as a discriminative one: the model must learn to articulate **why** a specific pattern deviates from an explicit, provided norm, rather than merely describing a segment in isolation.

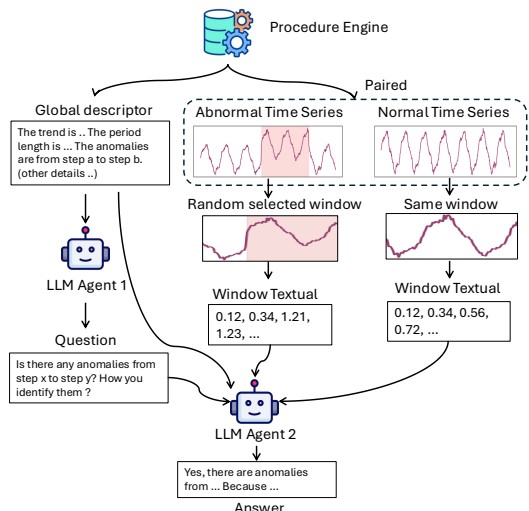

Figure 4: The architecture of our procedural engine for generating context-aware and comparative anomaly explanation benchmarks.

**LLM-Powered Explanation Generation.** Building on this structured foundation, we leverage LLMs to generate high-quality, multi-format supervision signals. As depicted in our pipeline, this is a multi-agent process. One LLM agent uses the global descriptor to formulate a targeted diagnostic question. A second, more powerful agent is then tasked with answering this question, conditioned on the global descriptor, the abnormal and normal window data. The primary motivation here is to generate rich, conceptual explanations. Our prompts are meticulously engineered to discourage superficial strategies (e.g., quoting raw values) and instead elicit reasoning based on the intrinsic, morphological characteristics of the anomaly. This process yields a diverse and consistent set of questions and detailed rationales, creating a powerful supervisory signal for training.

**Ensuring Benchmark Integrity.** To guarantee the scientific utility and integrity of our benchmark, we implement a rigorous quality control pipeline. This process verifies the agreement between ground-truth labels and generated answers, enforces stylistic consistency, and filters potential redundancies. We provide comprehensive dataset statistics and are releasing all generation metadata to ensure the full reproducibility of our benchmark. The detailed integrity check protocols are described in Appx. D. The result is not merely a dataset but a robust training environment engineered to finally bridge the semantic gap in time series anomaly explanation. Some examples of the generated Q&A pairs are given in Appx. C.2.

## 4 EXPERIMENTS

To validate the effectiveness of AXIS, we conduct a series of experiments designed to answer three central research questions: **RQ1: Explanation Quality.** How does AXIS compare against state-of-the-art LLM-based methods in generating high-quality, semantic anomaly explanations? **RQ2: Component Importance.** How do the core components of our framework—the symbolic numeric hint, the context-integrated step-aligned hint, and the task-prior hint—contribute to the final explanation quality? **RQ3: Architectural Generality.** How robust is the AXIS framework when applied to different underlying frozen LLMs? **RQ4: Generalization.** How well does AXIS generalize to external public datasets representing real-world scenarios? Finally, in Appx. B.2, we demonstrate that our Phase I TSAD model achieves results comparable to state-of-the-art methods on real-world public TSAD datasets.

### 4.1 EXPERIMENTAL SETUP

**Dataset.** All experiments are conducted on our newly created **Semantic Anomaly Benchmark** (detailed in Sec. 3.4 and Appx. F). This benchmark is specifically designed for the task of semantic time series anomaly explanation, containing diverse anomaly patterns, multi-format questions (Multiple Choice, True/False, Open-Ended), and detailed, pattern-level ground-truth explanations. The all hyperparameters for AXIS is given in Appx. B.3.

**Baselines.** We compare AXIS with a comprehensive set of strong baselines, categorized as follows: **Timeseries VLM:**(He et al., 2025) `Image LLM` is supported by gpt-4o, which analyzes plots of the full time series with highlighted window, treating the explanation task as a visual reasoning problem. **Specialized TS-LLM Methods:** We include several recent models designed for time series analysis with LLMs: `ChatTS` (Xie et al., 2024), `LLMAD` (Liu et al., 2025), `ChatTime` (Wang et al., 2025a), and `AnomLLM` (Dong et al., 2024). We evaluate `AnomLLM` in two settings: providing the full series (`AnomLLM(Full)`) and providing only the target window (`AnomLLM(Window)`). **Heuristic Baselines:** To contextualize benchmark difficulty, we introduce two non-learning baselines: (i) `Random Template`, which fills sophisticated-sounding templates with random values to test if the task is solvable by "hallucination"; and (ii) `Simple Heuristic`, a rule-based system using statistical thresholds (z-score, volatility) to flag anomalies.

**Evaluation Metrics.** Following recent work on evaluating LLM-generated content, we use an LLM-as-a-judge approach, specifically **G-eval** (Liu et al., 2023) with Gemini-2.5 as the arbiter. The quality of explanations is assessed across multiple dimensions tailored to each question type, including Correctness (Corr.), Reasoning Quality (Rsn. Qual.), Accuracy (Acc.), Completeness (Comp.), Relevance (Rel.), and Justification Quality (Justif.). A final, holistic score (Final) is also computed. The detailed definition for evaluation metrics is given in Appx. G

### 4.2 MAIN RESULTS: EXPLANATION QUALITY (RQ1)

Table 1 presents the main results comparing our model, AXIS, against all baselines. Our method demonstrates superior performance across all metrics and question types, establishing a new state-of-the-art for the task.

Specifically, AXIS achieves the highest final scores on Multiple Choice (4.19), Open-Ended (3.02), and True/False (3.65) questions. This consistent top-ranking performance highlights its robust ability to generate accurate, complete, and well-reasoned explanations regardless of the question format. Compared to specialized TS-LLM baselines like `ChatTS` and the `AnomLLM` variants, our method shows a significant improvement, underscoring the effectiveness of our proposed hint-based conditioning strategy. The strong performance against the `Image LLM` baseline further suggests that our multi-pathway representation provides richer, more aligned signals for the LLM than raw visual serialization.

**Visualization.** To qualitatively illustrate these performance gains, Fig. 5 presents a comparative case study. In the Fig. 5(a), the target window (steps 444 to 473) exhibits pronounced oscillations. AXIS correctly contextualizes these dynamics against the broader series, identifying them as part of a normal periodic pattern and concluding there is no anomaly. In stark contrast, a baseline like

Table 1: Main Results: AXIS vs Baselines

| Model | Multiple Choice | | | Open Ended | | | | True False | | |
| --- | --- | --- | --- | --- | --- | --- | --- | --- | --- | --- |
| | Final | Corr. | Rsn. Qual. | Final | Acc. | Comp. | Rel. | Final | Corr. | Justif. |
| **AXIS** | **4.19** | **4.21** | **4.14** | **3.02** | **2.87** | **2.93** | **3.31** | **3.65** | **3.60** | **3.74** |
| Image LLM | 4.09 | 4.12 | 4.02 | 2.68 | 2.53 | 2.49 | 3.07 | 2.64 | 2.57 | 2.74 |
| ChatTS | 3.29 | 3.40 | 3.05 | 2.19 | 1.67 | 2.13 | 2.87 | 2.79 | 2.76 | 2.83 |
| LLMAD | 2.73 | 2.70 | 2.79 | 2.09 | 2.09 | 1.89 | 2.31 | 2.49 | 2.52 | 2.43 |
| ChatTime | 1.33 | 1.49 | 0.98 | 0.96 | 0.95 | 0.98 | 0.95 | 1.04 | 1.07 | 1.00 |
| AnomLLM(Full) | 3.13 | 2.98 | 3.49 | 2.86 | 2.53 | 2.89 | 3.20 | 2.88 | 2.60 | 3.31 |
| AnomLLM(Window) | 3.78 | 3.81 | 3.70 | 2.84 | 2.78 | 2.55 | 3.24 | 3.32 | 3.45 | 3.12 |
| Baseline 1 (Random) | 1.02 | 1.03 | 1.00 | 1.21 | 1.21 | 1.05 | 1.41 | 1.29 | 1.36 | 1.18 |
| Baseline 2 (Heuristic) | 2.44 | 2.81 | 1.58 | 1.72 | 1.98 | 1.15 | 2.07 | 2.64 | 2.74 | 2.50 |

Table 2: Ablation Studies of Hint Components

| Model | Multiple Choice | | | Open Ended | | | | True False | | |
| --- | --- | --- | --- | --- | --- | --- | --- | --- | --- | --- |
| | Final | Corr. | Rsn. Qual. | Final | Acc. | Comp. | Rel. | Final | Corr. | Justif. |
| AXIS | **4.19** | **4.21** | **4.14** | **3.02** | **2.87** | **2.93** | **3.31** | **3.65** | **3.60** | **3.74** |
| w/o-task-hint | 3.82 | 3.93 | 3.56 | 2.33 | 2.13 | 2.22 | 2.69 | 3.25 | 3.31 | 3.17 |
| w/o-context-hint | 4.09 | 4.16 | 3.91 | 2.75 | 2.56 | 2.58 | 3.16 | 2.44 | 2.48 | 2.38 |
| w/o-windows | 3.95 | 4.00 | 3.84 | 2.41 | 2.00 | 2.36 | 2.95 | 2.87 | 2.83 | 2.93 |

AnomLLM or ChatTS, when limited to the window view, lacks this broader context and erroneously flags the internal deviations as potential outliers. In Fig. 5(b), AXIS provides a precise characterization by explicitly identifying the brief increase at steps 6–7 (2.27, 1.73, 2.38) and correctly interpreting it as a transient fluctuation rather than a sustained anomaly, concluding that the pattern is stable and anomaly-free. In contrast, alternative approaches tend to give vague or generalized descriptions, often noting fluctuations or moderate oscillations without distinguishing whether they indicate normal behavior or anomalies. This comparison highlights that our representation alignment framework enables fine-grained, context-aware interpretation of time series behavior, avoiding ambiguous assessments and ensuring robust anomaly detection.

**Human evaluation.** To further validate these quantitative results from a human-centric perspective, we also performed a statistical analysis based on expert rankings of the model-generated explanations. We conducted a survey where human evaluators were asked to rank the outputs from all competing models for each question type. Fig. 6 corroborates the findings from Table 1. Note that `ChatTime` is excluded from this analysis as it exhibited fundamental task-comprehension failures (e.g., inability to parse prompts), rendering its outputs non-comparable. Our model, AXIS, consistently achieves the best (lowest) average rank across all three tasks, positioning it furthest to the right. The details for experiment setup and results are shown in Appx. H.

**Validity Check via Heuristic Baselines.** As shown in Table 1, `Baseline 1 (Random)` performs near the lower bound across all metrics (1.00–1.41), confirming that fabricated explanations fail. `Baseline 2 (Heuristic)` shows modest improvement but still lags significantly behind semantic models, particularly on reasoning quality (1.58 vs 4.14 for AXIS), demonstrating that rule-based systems cannot generate coherent explanations. The substantial gap between these baselines and AXIS validates that the benchmark effectively discriminates between trivial heuristics and genuine semantic reasoning.

## 4.3 ABLATION STUDIES (RQ2)

To understand the contribution of each component in AXIS, we conducted an ablation study, with results summarized in Table 2. The study confirms that all three input pathways are crucial for optimal performance. Removing the context-integrated step-aligned hint ('w/o-context-hint') degrades performance most severely on True/False questions (Final score drops from 3.65 to 2.44), confirming that binary judgments rely on precise, step-aligned temporal cues. In contrast, Open-Ended questions are most sensitive to the symbolic numeric hint from the window ('w/o-windows'); without it, accuracy drops markedly (2.00 vs 2.87) and overall quality declines (Final 2.41 vs 3.02), indicating

(a) Visualization for contextual grounding ability

**Question**: When examining the time series window from step 444 to 473, what evidence would you look for to support or refute the presence of subtle anomalies, and how would you differentiate between normal fluctuations and genuine outlier behavior near the window boundaries?

**Ours (Score 5.00)** : There is **no evidence of anomalies** in the time series window from step 444 to 473. The pattern shows a smooth transition from positive to negative values, followed by a gradual recovery and stabilization. Near the window boundaries, the values remain within a predictable range, indicating normal behavior rather than outlier activity. **Thus, the observed fluctuations are characteristic of typical time series dynamics** and do not suggest the presence of subtle or genuine anomalies in this segment.

**AnomLLM (Score 1.65):** To identify subtle anomalies from steps 444 to 473, I would examine deviations beyond typical fluctuation ranges, such as sudden spikes or dips inconsistent with prior patterns. Differentiating normal fluctuations from outliers involves assessing whether boundary points significantly deviate from the local mean or trend, considering the context of surrounding data. Consistent, **isolated deviations near boundaries may suggest genuine outliers**, whereas gradual, minor variations are likely normal fluctuations.

**ChatTS (Score 1.65):** In this window, the primary concern is the transition from the steady trend to the final downward trend. **The periodic fluctuation amplitude is approximately 3.3, with a low noise Level (standard deviation of 0.06).** To distinguish between normal fluctuations and outliers, monitor deviations exceeding +2.0 from the baseline. The sudden decrease near the end of the time series (around point 528) may appear as an anomaly in this window.

**LLMAD (Score 3.60):** There are no segments with persistent high or low values that deviate significantly from the rest of the sequence. **The brief dip to -2 at index 11 is immediately surrounded by values close to zero and quickly returns to baseline,** which does not constitute a prolonged or abrupt anomaly. Upon careful review, all observed fluctuations are minor and fall within the expected range of normal data. There are no sequences or points that meet the criteria for any anomaly type with high confidence.

(b) Visualization for representation alignment ability

**Question**: Which of the following most accurately describes the pattern observed in the time series window from step 0 to 30, in terms of anomaly detection? A)The window shows abrupt spikes and unstable fluctuations suggesting sudden data corruption. B)The window displays cyclical irregularities that indicate potential recurring anomalies. C)The window contains gradual shifts that may signal the onset of a hidden anomaly. D)The window exhibits consistently stable behavior with no apparent anomalies near its boundaries.

**Ours (Score 5.00)** D) The window exhibits consistently stable behavior with no apparent anomalies near its boundaries. Explanation: The time series window from step 0 to 30 primarily displays values fluctuating within a moderate range, with no evidence of sudden spikes, abrupt changes, or irregular patterns that would suggest anomalous behavior. **While there is a brief increase at steps 6 and 7 (2.27, 1.73, 2.38), this is quickly followed by a return to lower values, indicating a short-lived fluctuation rather than a sustained anomaly.** Overall, the pattern remains stable, and there are no signs of anomalies within this window. Therefore, the correct answer is D.

**AnomLLM Window (Score 4.30):** The time series from steps 0 to 30 exhibits fluctuations with some localized spikes, notably around step 6(2.27)and step 8(2.37),which are significantly higher than surrounding values, indicating potential anomalies. However overall, the series shows no persistent abrupt spikes or cyclical patterns, and the fluctuations appear relatively contained, suggesting no clear signs of instability or cyclical irregularities. **The data mostly oscillates within a moderate range, with no consistent trend of gradual shifts.** Therefore, the most accurate description is that the window exhibits stable behavior with isolated anomalies rather than sustained or cyclical irregularities.

**ChatTS (Score 4.00):** The time series exhibits a steady trend without any significant local characteristics or abrupt changes. **The amplitude** of periodic fluctuation is approximately 2.7, indicating consistent square periodic patterns. **With a noise standard deviation** of around 0.70, the curve shows random variations but remains stable overall. Therefore, the correct answer is D. The window from step 0 to 30 does not show any abrupt spikes, cyclical irregularities, or gradual shifts, maintaining a stable pattern consistent with the overall series.

**LLMAD (Score 3.30):** There are no segments with continuous high or low values deviating from the anticipated norm. The data remains within a narrow band, and the few occurrences of -1 do not persist for a prolonged duration nor do they represent a significant deviation. Upon careful review, no data points or intervals meet the criteria for anomaly types such as spikes, dips, or level shifts. The data sequence is consistent with normal operational volatility, and there is no evidence of rare or severe anomalies.

Figure 5: Visualization of (a) contextual grounding and (b) representation alignment ability

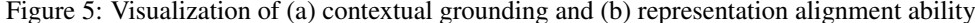
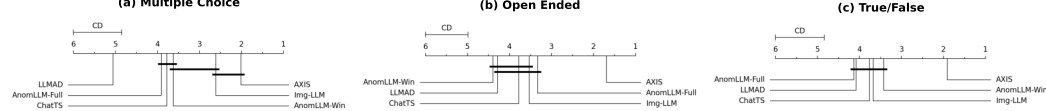

Figure 6: Critical Difference diagrams illustrating the statistical comparison of model performance based on human rankings for (a) Multiple Choice, (b) Open-Ended, and (c) True/False questions.

that direct numeric access provides the fine-grained grounding needed for detailed answers. Eliminating the task-prior hint ('w/o-task-hint') also substantially harms Open-Ended completeness and relevance (2.22/2.69 vs 2.93/3.31), suggesting that these global priors help structure the explanation and ensure comprehensive coverage. The full model consistently yields the best scores, validating our design choices. Additional experiments for causal contribution of hints are given in Appx. I.

## 4.4 ANALYSIS OF ARCHITECTURAL VARIANTS (RQ3)

To assess its generality, we instantiated AXIS across multiple LLM families and settings, using a standardized *Family + Size + Variant* naming scheme and a fixed data schedule (R1) to isolate architectural effects. As shown in Table 3, our framework demonstrates robust cross-family adaptation (Qwen, Llama, Mistral) and its performance scales with model size (e.g., Qwen-14B > 7B > 1.5B). The results also reveal complementary strengths among model variants: code-pretrained models like *Qwen2.5-7B Coder* excel at structured discrimination tasks, achieving the highest Multiple Choice score (4.40), whereas instruction-tuned versions such as *Qwen2.5-7B Instruct* lead in free-form explanatory quality, with top scores in Open-Ended relevance and completeness (3.50/3.00). This

Table 3: AXIS variants across LLM families and settings (standardized naming: family + size + variant; *Instruct* denotes instruction-tuned, *Coder* denotes code-pretrained

| Family | Variant | Multiple Choice | | | Open Ended | | | | True False | | |
|---|---|---|---|---|---|---|---|---|---|---|---|
| | | Final | Corr. | Rsn. Qual. | Final | Acc. | Comp. | Rel. | Final | Corr. | Justif. |
| Deepseek-Llama | 8B (Instruct) | 4.28 | 4.30 | 4.23 | 3.02 | 2.84 | 2.84 | 3.45 | 3.64 | 3.55 | **3.79** |
| Deepseek-Qwen | 14B (Instruct) | 4.31 | 4.28 | 4.37 | 3.03 | 2.80 | 2.93 | 3.42 | 3.60 | 3.55 | 3.69 |
| Deepseek-Qwen | 7B (Instruct) | 4.19 | 4.21 | 4.14 | 3.02 | **2.87** | 2.93 | 3.31 | 3.65 | 3.60 | 3.74 |
| Deepseek-Qwen | 1.5B (Instruct) | 4.07 | 4.12 | 3.95 | 2.72 | 2.65 | 2.55 | 3.00 | 3.18 | 3.17 | 3.19 |
| Qwen2.5 | 7B (Coder) | **4.40** | **4.40** | **4.42** | 2.72 | 2.64 | 2.58 | 2.98 | 2.96 | 2.98 | 2.93 |
| Qwen2.5 | 7B (Base) | 4.30 | 4.37 | 4.12 | 2.75 | 2.73 | 2.45 | 3.13 | **3.66** | **3.69** | 3.62 |
| Qwen2.5 | 7B (Instruct) | 4.17 | 4.22 | 4.06 | **3.08** | 2.80 | **3.00** | **3.50** | 3.11 | 3.00 | 3.27 |
| Mistral | 7B (Base) | 2.97 | 3.05 | 2.79 | 2.89 | 2.69 | 2.82 | 3.20 | 2.69 | 2.55 | 2.90 |
| Mistral | 7B (Instruct) | 3.36 | 3.33 | 3.44 | 2.77 | 2.58 | 2.45 | 3.35 | 3.27 | 3.17 | 3.43 |

Table 4: Performance comparison on Multiple Choice (MC) and True/False (TF) tasks.

| Dataset | Metric | AnomLLM | | LLMAD | ChatTS | AXIS 14B | Image LLM |
|---|---|---|---|---|---|---|---|
| | | Window | Full | | | | |
| YAHOO (W=0.651) | MC Score | 2.78 | 1.69 | 2.48 | 2.83 | **3.07** | 2.80 |
| | MC Acc. | 0.48 | 0.18 | 0.47 | 0.52 | **0.55** | 0.52 |
| | TF Score | 2.62 | 1.90 | 2.54 | 2.13 | **3.28** | 2.29 |
| | TF Acc. | 0.53 | 0.38 | 0.51 | 0.45 | **0.67** | 0.52 |
| TODS (W=0.672) | MC Score | 2.07 | 1.77 | 2.33 | 2.67 | **2.90** | 2.53 |
| | MC Acc. | 0.37 | 0.30 | 0.33 | 0.47 | **0.53** | 0.47 |
| | TF Score | 2.67 | 2.53 | 2.87 | 2.53 | **3.07** | 2.27 |
| | TF Acc. | 0.53 | 0.52 | 0.58 | 0.53 | **0.62** | 0.50 |
| NEK (W=0.713) | MC Score | 3.31 | 1.94 | 1.44 | 3.06 | 3.25 | **3.41** |
| | MC Acc. | **0.63** | 0.31 | 0.13 | 0.56 | **0.63** | **0.63** |
| | TF Score | 2.88 | 1.69 | 2.94 | 2.25 | **3.69** | 2.25 |
| | TF Acc. | 0.56 | 0.35 | 0.59 | 0.47 | **0.74** | 0.50 |

highlights that AXIS not only universally enhances different base models but also allows for trade-offs between discriminative and explanatory objectives through strategic variant selection.

## 4.5 GENERALIZATION TO PUBLIC REAL-WORLD DATASETS (RQ4)

To rigorously evaluate the generalization capability of AXIS beyond our synthetic benchmark, we extend our evaluation to established public datasets representing real-world operational scenarios. We construct a curated evaluation set derived from 108 time series sampled from three distinct sources: YAHOO, TODS, and NEK (details in Appendix H.3). We formulate two specific tasks—True/False and Multiple-Choice Questions—and employed expert annotation to establish high-quality ground truth, ensuring a robust testbed for real-world explanation quality.

The results, summarized in Table 4, demonstrate that AXIS consistently delivers superior explanation quality across these diverse domains. Our model achieves the highest scores in both MC and TF categories on the YAHOO and TODS datasets, and leads in TF performance on NEK. Notably, the strong inter-rater agreement (Kendall's W > 0.65) across all datasets confirms the reliability of our human evaluation. These findings validate that AXIS effectively generalizes to real-world data distributions, bridging the gap between synthetic training and practical operational deployment.

## 5 CONCLUSION

We introduce a novel cross-modal framework that effectively adapts frozen Large Language Models for semantic time series anomaly explanation. By using a three-stream conditioning strategy that combines a symbolic numeric hint, a context-integrated step-aligned hint, and a task-prior hint, our method achieves strong performance in both detection accuracy and explanation quality. Future work will explore incorporating domain-specific knowledge graphs to enhance causal reasoning and generating multi-modal explanations that include visualizations alongside text.

## ETHICS STATEMENT

This work focuses on explainable time series anomaly detection and does not involve personally identifiable information or other sensitive attributes. Our benchmark primarily uses procedurally synthesized data and publicly available datasets; no private logs are collected, and no re-identification is attempted. Human evaluation was conducted with informed consent, anonymized responses, and fair compensation in line with institutional guidelines, without storing any personally identifying information.

## REPRODUCITY STATEMENT

We aim for full reproducibility. Upon publication, we will release code, configuration files, and scripts to reproduce: (i) the benchmark synthesis pipeline (including prompts, fixed random seeds, and parameter settings); (ii) Phase I encoder pretraining and Phase II hint tuning with exact hyperparameters, token budgets, and training schedules (see Appendix B.2 and Appendix B.3); and (iii) evaluation pipelines, including baseline configurations, G-Eval judge prompts, scoring scripts, and human-evaluation materials. We will provide model checkpoints where licensing permits or otherwise specify exact model identifiers and initialization procedures. The code is available in `https://anonymous.4open.science/r/TimeSemantic-1742/main.py`

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
