# A USE OF LLMS

Our manuscript preparation involved the use of large language models (LLMs) primarily for refining language, improving grammar, readability, and stylistic elements. Crucially, LLMs also constituted a core component of our research process. They were leveraged as agents to participate in data generation, established as benchmarks for experimental baselines, and contributed to the comprehensive evaluation of experimental results. For a detailed account of LLMs' specific applications within our methodology, please refer to Section 3.4 and Section 4. We confirm that the contributions of LLMs, whether in writing or research, do not impede the reproducibility of our reported findings.

# B DETAILS FOR AXIS

## B.1 TIME-SERIES ENCODER DETAILS

This section details the architecture of the time-series encoder $f_\theta$ used in AXIS. The overall structure is shown in Fig. 7.

**Patchify.** Let $P$ denote the patch size. We pad $\mathbf{x}_{1:T}$ to length $T' = \lceil T/P \rceil P$ with zeros and form $N = T'/P$ contiguous, non-overlapping patches:

$$\mathbf{X}^{(n)} = [x_{(n-1)P+1}, \ldots, x_{nP}] \in \mathbb{R}^P, \quad n = 1, \ldots, N.$$

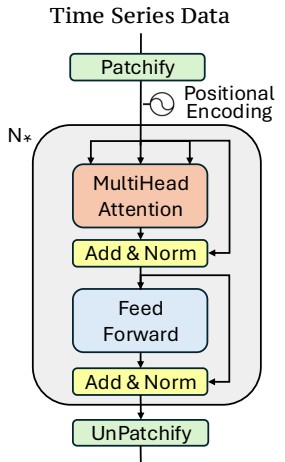

Time Series Data

Figure 7: The structure of time-series encoder.

Stacking yields $\mathbf{X}_p \in \mathbb{R}^{N \times P}$. The patch-level attention mask is obtained by aggregating the step mask: a patch is valid if it contains at least one valid step,

$$\mathbf{m}_n^{\text{patch}} = \mathbb{I}\Big( \sum_{t=(n-1)P+1}^{nP} m_t > 0 \Big), \quad n = 1, \ldots, N.$$

**Patch Embedding.** Each patch is projected to the model dimension $d_{\text{model}}$ with a linear layer

$$\mathbf{Z} = \mathbf{X}_p \mathbf{W}_e + \mathbf{1}\, \mathbf{b}_e^\top \in \mathbb{R}^{N \times d_{\text{model}}}, \quad \mathbf{W}_e \in \mathbb{R}^{P \times d_{\text{model}}}.$$

**Positional Encoding via RoPE.** The encoder uses rotary positional encoding (RoPE) applied *inside* self-attention to the query and key of each head. Concretely, for a sequence of $N$ patch tokens, RoPE generates frequency pairs that rotate the $d_{\text{model}}$ channels in 2D subspaces, providing translation-friendly relative position information.

**Transformer Encoder (Non-causal).** We stack $L$ pre-norm Transformer encoder layers. Each layer comprises: (i) multi-head self-attention with $H$ heads and head dimension $d_{\text{head}} = d_{\text{model}}/H$; (ii) an MLP block of the LLaMA style with gated GELU activation. RMSNorm is used before attention and MLP. Attention is *non-causal* and operates over all $N$ patches, enabling global context aggregation. The attention mask derived from $\mathbf{m}^{\text{patch}}$ prevents padded tokens from contributing. In the univariate case there is no inter-feature bias term; the attention is standard scaled dot-product with RoPE.

**UnPatchify and Projection to Step Alignment.** The encoder output $\mathbf{U} \in \mathbb{R}^{N \times d_{\text{model}}}$ is mapped to a patch-level representation of size $P \times d_{\text{proj}}$ through a linear projection

$$\mathbf{Y} = \mathbf{U} \mathbf{W}_p \in \mathbb{R}^{N \times (P\, d_{\text{proj}})}, \quad \mathbf{W}_p \in \mathbb{R}^{d_{\text{model}} \times (P\, d_{\text{proj}})}.$$

Reshaping and reordering yield step-aligned embeddings $\tilde{\mathbf{H}} \in \mathbb{R}^{T' \times d_{\text{proj}}}$. We finally drop the padded tail to obtain

$$\mathbf{H}_{1:T} = \tilde{\mathbf{H}}_{1:T} \in \mathbb{R}^{T \times d_{\text{proj}}}.$$

## B.2 TRAINING PROCESS DETAILS

Our model is trained in a two-phase process designed to first build a robust time-series representation and then align it with the frozen LLM's semantic space.

### B.2.1 PHASE I: TIME-SERIES ENCODER PRETRAINING

In the first phase, we pretrain the time-series encoder $f_\theta$ to learn a versatile representation of temporal dynamics. This is achieved by optimizing a joint objective function that combines masked reconstruction and anomaly classification. The encoder is a Transformer-based architecture that processes the entire input time series $\mathbf{x}_{1:T}$.

**Masked Reconstruction.** Following the principles of self-supervised learning for sequential data, we employ a masked reconstruction objective. A portion of the input time series is randomly masked, and the encoder is tasked with reconstructing the masked values. Let $\mathbf{x}$ be the input series and $\mathbf{m}$ be a binary mask where $m_i = 1$ if the $i$-th timestep is masked and 0 otherwise. The encoder $f_\theta$, followed by a reconstruction head $g_\psi$, outputs a reconstructed series

$$\hat{\mathbf{x}} = g_\psi\big(f_\theta(\mathbf{x} \odot (\mathbf{1} - \mathbf{m}))\big).$$

The reconstruction loss is the Mean Squared Error (MSE) over the masked timesteps:

$$\mathcal{L}_{\text{recon}} = \frac{1}{\sum_i m_i} \sum_{i=1}^{T} m_i (x_i - \hat{x}_i)^2.$$

This objective forces the encoder to learn the underlying patterns and dependencies within the time series.

**Anomaly Classification.** To explicitly teach the encoder to distinguish between normal and anomalous patterns. Given a time series $\mathbf{x}$ and its corresponding binary anomaly label $\mathbf{l} \in \{0, 1\}^T$, the model predicts the probability of an anomaly $\hat{\mathbf{l}}_t = h_\zeta(f_\theta(\mathbf{x}))$. We use the BCE loss:

$$\mathcal{L}_{\text{class}} = -\sum_{t=1}^{T} \big(l_t \log(\hat{l}_t) + (1 - l_t) \log(1 - \hat{l}_t)\big).$$

**Joint Objective.** The total loss for the pretraining phase is a weighted sum of the reconstruction and classification losses:

$$\mathcal{L}_{\text{pretrain}} = \lambda_{\text{recon}} \mathcal{L}_{\text{recon}} + \lambda_{\text{class}} \mathcal{L}_{\text{class}},$$

where $\lambda_{\text{recon}}$ and $\lambda_{\text{class}}$ are hyperparameters that balance the two objectives. After this phase, the weights of the encoder $f_\theta$ are frozen (only $f_\theta$ is retained for the next phase).

### B.2.2 PHASE II: HINT TUNER TRAINING

In the second phase, we focus on aligning the pretrained time-series representations with the LLM. Both the time-series encoder $f_\theta$ and the LLM are kept frozen to maintain their powerful, pre-existing capabilities. The only trainable components are the Hint Tuner and its associated parameters, namely the prototype bank selection matrix $\mathbf{M}$ and the learnable task-prior queries $\mathbf{P}_{\text{fix}}$.

The goal of this phase is to train the model to generate a natural language explanation $\mathbf{y} = (y_1, y_2, \ldots, y_N)$ conditioned on the time series $\mathbf{x}$, the query $q$, and the derived hints. The training objective is a standard causal language modeling loss, which maximizes the likelihood of the ground-truth explanation.

Specifically, the input to the LLM is constructed by embedding the query $q$, the symbolic-numeric hint, and replacing placeholder token embeddings with the outputs of the Hint Tuner ($\tilde{\mathbf{H}}_{s:e}$ and $\tilde{\mathbf{F}}$). Here, $s$ and $e$ denote the start and end indices of the selected window within the full series $\mathbf{x}_{1:T}$. The model then autoregressively predicts the next token in the explanation sequence. The loss function is the negative log-likelihood of the target sequence:

$$\mathcal{L}_{\text{LM}} = -\sum_{i=1}^{N} \log P\big(y_i \mid y_{<i}, q, \tilde{\mathbf{H}}_{s:e}, \tilde{\mathbf{F}}\big).$$

By optimizing this objective, the Hint Tuner learns to map temporal features from the frozen encoder into meaningful "soft prompts" that effectively guide the frozen LLM to generate accurate and relevant explanations for the given time-series window. During this phase, gradients update only the Hint Tuner and its associated parameters ($\mathbf{M}$ and $\mathbf{P}_{\text{fix}}$); both $f_\theta$ and all LLM parameters remain frozen.

### B.3 AXIS HYPERPARAMETER CONFIGURATION

This section provides a comprehensive specification of the hyperparameter configuration used in our AXIS framework. The architecture consists of two primary components: the time-series encoder and the hint tuner, each with distinct parameter settings optimized through systematic ablation studies.

**Time-Series Encoder Configuration.** The time-series encoder $f_\theta$ employs a Transformer-based architecture with patch-based tokenization. The patch size $P$ is set to 16, enabling efficient processing of long sequences while preserving temporal granularity. The encoder utilizes $L = 6$ transformer layers, each with a model dimension $d_{\text{model}} = 512$ and $H = 8$ attention heads, resulting in a head dimension $d_{\text{head}} = 64$. The projection dimension $d_{\text{proj}}$ is configured to 256.

**Hint Tuner Architecture.** The Perceiver-based hint tuner serves as the cross-modal alignment module between time-series representations and the frozen LLM embedding space. The prototype bank consists of $P = 1024$ prototype embeddings derived from the LLM vocabulary through a learned linear mapping. The number of fixed task-prior tokens $K$ is set to 8, providing sufficient capacity for global task-level priors while maintaining computational efficiency. The cross-attention mechanism within the hint tuner employs 8 attention heads, matching the encoder configuration for architectural consistency.

**Language Model Integration.** Our framework supports multiple LLM families with varying parameter scales. The experiments primarily utilize Qwen2.5-7B-Instruct as the base model, with the hidden dimension $d_h = 4096$ for the 7B variant. The vocabulary size varies by model family, typically ranging from 32,000 to 152,000 tokens. Special tokens `<|local_hint|>` and `<|fixed_hint|>` are added to the vocabulary for hint injection, with corresponding token IDs dynamically assigned during initialization.

**Training Configuration.** The two-phase training procedure employs distinct hyperparameter settings for each phase. During Phase I (encoder pretraining), the reconstruction loss weight $\lambda_{\text{recon}}$ and classification loss weight $\lambda_{\text{class}}$ are both set to 1.0, providing balanced supervision across objectives. The masking ratio for reconstruction is configured to 0.25, following established practices in self-supervised time-series learning. Phase II (hint tuner training) utilizes standard causal language modeling with a learning rate of $2 \times 10^{-4}$ and weight decay of 0.01, applied exclusively to the trainable parameters of the hint tuner and prototype bank.

**Inference Parameters.** During generation, the model employs beam search with 5 beams and a repetition penalty of 1.15 to ensure diverse and coherent explanations. The maximum generation length is capped at 1000 tokens to accommodate detailed explanations while preventing excessive verbosity. No-repeat n-gram size is set to 3 to avoid repetitive patterns, and length penalty is maintained at 1.0 to balance explanation completeness with conciseness.

## C PROMPT TEMPLATES AND Q&A EXEMPLARS

### C.1 PROMPT TEMPLATES

#### C.1.1 QUESTION GENERATION PROMPT

```
System: You generate precise and relevant questions for time series
    anomaly detection.

User:
```

```
Generate a {question_type} question focused on anomaly detection for the
    window [ {window_start}, {window_end} ].

Context:
- Task: time series anomaly detection on a windowed segment
- The window {may / does} contain anomalies: {has_anomaly}
- Canonical tag (if available): {canonical_tag}
- Global information: {global_information}

Requirements:
1) Output ONLY the question text (no answers, no explanations).
2) Focus on anomaly identification and pattern analysis within the window
    .
3) Consider boundary effects near window edges.
4) Multiple choice (if applicable):
   - Provide exactly 4 options (A, B, C, D).
   - Options must be mutually exclusive, same style.
   - Include both normal and anomalous descriptions; avoid exact numeric
       values.
5) True/False (if applicable):
   - Make a specific statement about a potential anomaly pattern.
6) Open-ended (if applicable):
   - Ask about pattern evidence and reasoning for/against anomalies.
```

### C.1.2   ANSWER GENERATION PROMPT

```
System: You analyze time series patterns and generate concise answers.

User:
You are given a time series window [ {window_start}, {window_end} ],
    which belongs to a longer series.
Global information: {global_information}
Canonical tag (if available): {canonical_tag}
Anomaly description (if any): {anomaly_description}
Data [current_value(normal_value)]: [{data_str}]
Question: {question}

Constraints:
- Focus on the pattern of current_values; avoid relying on normal_values.
- Keep the answer concise (<= 150 words), pattern-first (e.g., sustained
    level change, volatility burst).
- MCQ: start with the correct option letter, then explanation (e.g., "B)
    ...").
- True/False: start with "True" or "False", then explanation.
- Do not quote exact numeric values; reason from shape, persistence,
    variability.
- If no anomaly, state it clearly with supporting evidence.
```

### C.2   Q&A EXEMPLARS

### C.2.1   MULTIPLE CHOICE

```
Type: multiple_choice
Window: [668, 707]
Has anomaly: false
Anomaly descriptions: []

Question:
Which of the following best describes the anomaly detection results for
    the time series window from step 668 to 707?
```

```
A) The window contains a sudden spike suggesting an isolated anomaly near
    its center.
B) The window shows a consistent pattern without any signs of anomalous
    behavior.
C) The window includes a gradual drift at the boundary, indicating a
    possible transition to an anomaly.
D) The window exhibits irregular fluctuations that are characteristic of
    multiple overlapping anomalies.

Answer:
B) The window shows a consistent pattern without any signs of anomalous
    behavior.
The time series window from step 668 to 707 displays a steady, gradual
    decrease in values, with each step typically dropping by 0.01 or
    remaining constant for a brief interval before continuing the gradual
     trend. There are no sudden spikes, irregular fluctuations, or abrupt
     changes that would indicate an anomaly. The pattern is smooth and
    consistent throughout the window, suggesting stable behavior and the
    absence of any anomalous events within this interval.
```

### C.2.2 TRUE/FALSE

```
Type: true_false
Window: [418, 458]
Has anomaly: true
Anomaly descriptions:
["A local continuous upward spike anomaly, featuring 5 consecutive spikes
    with amplitudes from 3.03 to 5.93"]

Question:
True or False: The presence of a cluster of consecutive upward spikes
    within the window from step 418 to 458 suggests an anomalous
    deviation from the expected time series pattern.

Answer:
True. Within the window from step 418 to 458, there is a cluster of
    consecutive upward spikes-specifically, a sequence of five elevated
    values (2.26, 2.05, 2.05, 2.05, 0.71) that stand out sharply against
    the otherwise stable and low baseline pattern (around 0.7). This
    sudden, localized series of spikes deviates significantly from the
    expected behavior of the time series, which is generally flat and
    consistent. The presence of these consecutive upward anomalies
    suggests a notable and abrupt deviation from the normal pattern,
    indicating an anomalous event within this segment of the data.
```

### C.2.3 OPEN-ENDED

```
Type: open_ended
Window: [294, 311]
Has anomaly: false
Anomaly descriptions: []

Question:
How would you assess the presence or absence of anomalies in the time
    series window from step 294 to 311, and what evidence from the data
    supports your conclusion regarding the normality or abnormality of
    this segment?

Answer:
There is no evidence of anomalies in the time series window from step 294
     to 311. The values show a gradual decrease from 0.88 to 0.62,
    followed by a mild recovery back up to 0.73. This pattern reflects a
```

```
        smooth and consistent trend without sudden spikes, abrupt drops, or
        irregular fluctuations. The transitions between values are steady and
         do not indicate any unusual or unexpected behavior. Overall, the
        segment demonstrates normal and stable behavior within the observed
        window.
```

## D  BENCHMARK INTEGRITY CHECK PROTOCOLS

To ensure the scientific validity and reproducibility of our benchmark, we implemented a rigorous three-stage automated filtering pipeline to identify and remove low-quality or biased samples.

**1. Label Consistency Check.**   This step ensures that the semantic explanation aligns with the ground-truth anomaly label. We parse the generated answer to extract the definitive judgment (e.g., "Yes, there is an anomaly" or "No anomaly detected"). We then compare this judgment against the procedural injection log. If the generator claims an anomaly exists in a window that the procedural engine marked as normal (or vice versa), the sample is discarded. This eliminates "hallucinated" anomalies where the LLM invents patterns not present in the signal.

**2. Information Leakage Check.**   Since the question generation agent has access to metadata about the injected anomaly, there is a risk that the question itself might reveal the answer (e.g., "Why is there a sudden spike at step 50?"). We use a heuristic filter to detect questions that contain specific anomaly types or location indicators that should be part of the answer. Questions that inadvertently leak the ground truth are removed to ensure the task remains a valid test of detection and reasoning.

**3. Stylistic Normalization.**   LLM-generated text can sometimes contain distinct stylistic artifacts ("LLM-isms") or repetitive phrasing. If left unchecked, these could serve as spurious correlations that a model might exploit (e.g., learning that longer answers always correspond to anomalies). We apply a normalization step to standardize the output format, remove repetitive filler phrases, and ensure a neutral tone. This minimizes the risk of the evaluator model relying on superficial textual cues rather than the semantic content of the explanation.

## E  PHASE I TSAD RESULTS

**Datasets.**   We benchmark nine public univariate time-series anomaly detection corpora: IOPS, MGAB, NAB, NEK, Power, SED, TODS, UCR, and YAHOO. These datasets span industrial KPIs, synthetic chaotic dynamics, web traffic, power consumption, network/system logs, and curated mixed-domain series. Each series is standardized via z-score normalization using training-set statistics.

**Baselines.**   The comparison covers classical, deep generative, and foundation-style models. IForest (Liu et al., 2008) and LOF (Breunig et al., 2000) serve as nonparametric unsupervised baselines. OmniAnomaly (Su et al., 2019), USAD (Audibert et al., 2020), and TranAD (Tuli et al., 2022) represent deep unsupervised anomaly detectors. DADA (Shentu et al., 2024) and TSPulse (Ekambaram et al., 2025), MOMENT_ZS (Goswami et al., 2024), Chronos (Ansari et al., 2024), TimesFM (Das et al., 2024), and Time_MOE (Xiaoming et al., 2025) are pretrained time-series models evaluated in zero-shot or lightly adapted settings. AXIS denotes a semantic representation-based detector. Implementations follow public releases or authors' configurations when feasible.

**Metrics.**   We report three standard metrics. (i) PA-F1: the point-adjusted F1 that credits a hit if any point within an anomalous window is flagged, computed from precision and recall after point adjustment; threshold for PA-F1 is selected on a validation split when available, or via an unsupervised percentile heuristic on training scores. (ii) AUC-ROC: area under the receiver operating characteristic, threshold-free and aggregated over all test points. (iii) AUC-PR: area under the precision–recall curve, which is more informative under severe class imbalance. Higher values indicate better performance for all metrics.

Table 5: Model Performance Comparison Across Univariate Datasets

| Dataset | Metric | Foundation Models | | | | | | | Deep Learning | | | Classical | |
|---|---|---|---|---|---|---|---|---|---|---|---|---|---|
| | | AXIS | Chronos | DADA | MOMENT_ZS | TS_Pulse | Time_MOE | TimesFM | OmniAnomaly | TranAD | USAD | IForest | LOF |
| IOPS | PA-F1 | 57.19 | 64.17 | 55.52 | 44.76 | 50.59 | 41.58 | **87.89** | 53.50 | 53.04 | 41.24 | 15.31 | 41.38 |
| | AUC-ROC | 71.13 | 72.16 | 78.61 | **82.67** | 49.71 | 61.87 | 70.12 | 80.89 | 69.44 | 81.31 | 48.29 | 66.80 |
| | AUC-PR | 18.13 | 25.65 | 29.59 | 22.45 | 1.65 | 24.15 | 28.33 | **41.63** | 25.51 | 21.96 | 6.49 | 25.63 |
| MGAB | PA-F1 | 7.13 | 10.16 | 3.11 | 2.90 | 11.39 | 2.64 | 7.74 | 7.62 | 7.80 | 3.49 | 5.85 | **12.31** |
| | AUC-ROC | **69.97** | 50.78 | 54.12 | 47.47 | 49.94 | 37.73 | 49.97 | 60.15 | 61.04 | 51.12 | 43.88 | 50.27 |
| | AUC-PR | 0.55 | 0.31 | 0.31 | 0.25 | 0.27 | 0.20 | 0.27 | 0.41 | 0.42 | **1.74** | 0.24 | 0.34 |
| NAB | PA-F1 | 93.73 | 97.08 | 96.18 | 94.61 | 97.76 | 90.48 | **99.09** | 98.02 | 97.93 | 93.42 | 42.81 | 86.79 |
| | AUC-ROC | 51.79 | 53.82 | 56.23 | 64.01 | 50.28 | 50.82 | 53.33 | 56.26 | 54.33 | **73.09** | 55.26 | 49.93 |
| | AUC-PR | 21.00 | 20.62 | 21.87 | 42.73 | 13.91 | 20.32 | 21.12 | 24.87 | 21.92 | 20.48 | **53.14** | 18.55 |
| NEK | PA-F1 | 87.19 | 98.44 | 98.68 | 81.10 | 88.14 | 50.82 | **98.79** | 88.50 | 86.78 | 66.26 | 72.47 | 87.35 |
| | AUC-ROC | 53.28 | 57.60 | 82.51 | **92.02** | 53.94 | 37.41 | 62.64 | 88.85 | 81.25 | 91.65 | 85.66 | 81.38 |
| | AUC-PR | 30.32 | 30.63 | 44.90 | 55.75 | 16.34 | 7.13 | 34.05 | **73.10** | 57.39 | 52.91 | 52.76 | 55.63 |
| Power | PA-F1 | **98.94** | 95.97 | 89.53 | 90.25 | 98.77 | 83.29 | 97.58 | 86.66 | 87.51 | 68.06 | 0.00 | 87.92 |
| | AUC-ROC | 62.47 | 48.82 | 46.19 | 43.23 | 50.68 | 38.92 | 46.78 | 59.88 | 57.05 | **66.46** | 50.00 | 38.93 |
| | AUC-PR | **20.52** | 10.44 | 10.07 | 10.68 | 11.09 | 9.01 | 9.98 | 13.64 | 12.37 | 17.30 | 10.97 | 8.98 |
| SED | PA-F1 | **93.59** | 38.77 | 18.69 | 1.92 | 70.26 | 33.29 | 32.49 | 2.25 | 4.49 | 0.00 | 31.62 | 0.00 |
| | AUC-ROC | **96.35** | 53.03 | 14.85 | 1.43 | 49.23 | 69.08 | 29.64 | 28.93 | 12.89 | 10.71 | 27.96 | 27.35 |
| | AUC-PR | **73.76** | 4.80 | 2.77 | 2.57 | 4.95 | 7.45 | 3.26 | 3.35 | 2.78 | 2.68 | 3.26 | 3.25 |
| TODS | PA-F1 | 77.50 | **83.40** | 61.20 | 26.86 | 53.03 | 44.92 | 81.26 | 38.82 | 40.31 | 32.67 | 23.02 | 54.62 |
| | AUC-ROC | **92.40** | 68.45 | 68.16 | 46.17 | 50.20 | 51.68 | 72.96 | 49.69 | 48.87 | 53.92 | 51.16 | 53.12 |
| | AUC-PR | **68.20** | 35.60 | 20.33 | 10.03 | 6.98 | 10.36 | 36.69 | 6.89 | 6.74 | 17.37 | 7.43 | 19.74 |
| UCR | PA-F1 | 47.58 | 43.71 | 34.90 | 25.55 | 56.57 | 31.63 | **60.88** | 31.44 | 32.67 | 23.02 | 23.02 | 32.43 |
| | AUC-ROC | **79.34** | 55.79 | 56.27 | 54.92 | 50.57 | 55.38 | 57.21 | 50.67 | 50.64 | 53.74 | 51.93 | 55.86 |
| | AUC-PR | 24.06 | 6.11 | 2.19 | 6.92 | 0.85 | 2.12 | 6.35 | 2.91 | 2.32 | 8.08 | 2.54 | 2.50 |
| YAHOO | PA-F1 | 53.88 | 54.09 | 50.82 | 12.54 | 7.99 | 21.39 | **87.18** | 18.66 | 8.09 | 7.40 | 2.86 | 29.78 |
| | AUC-ROC | **96.55** | 94.64 | 95.95 | 74.37 | 49.02 | 63.08 | 96.35 | 77.71 | 56.47 | 62.54 | 51.93 | 72.85 |
| | AUC-PR | **86.07** | 70.75 | 76.66 | 4.84 | 1.49 | 22.08 | 79.23 | 16.28 | 2.90 | 3.72 | 1.66 | 44.26 |
| Overall | Avg. Rank | **3.81** | 4.74 | 5.52 | 7.48 | 7.70 | 8.70 | 4.41 | 5.48 | 7.00 | 6.78 | 9.07 | 7.30 |

Table 6: Model Performance Comparison Across Multivariate Datasets

| Dataset | Metric | Foundation Models | | | | | | | Deep Learning | | | Classical | |
|---|---|---|---|---|---|---|---|---|---|---|---|---|---|
| | | AXIS | Chronos | DADA | MOMENT_ZS | TS_Pulse | Time_MOE | TimesFM | OmniAnomaly | TranAD | USAD | LOF | IForest |
| MSL | PA-F1 | 82.99 | 81.68 | 86.93 | 88.46 | 82.30 | 82.08 | 75.88 | 50.15 | 83.35 | 51.16 | **89.56** | 56.66 |
| | AUC-ROC | 72.17 | 69.79 | **75.32** | 52.15 | 51.05 | 64.59 | 70.45 | 62.28 | 49.02 | 61.98 | 56.65 | 52.88 |
| | AUC-PR | 20.56 | 20.21 | **26.42** | 11.60 | 7.46 | 19.83 | 20.34 | 17.19 | 7.42 | 17.24 | 15.18 | 11.31 |
| PSM | PA-F1 | 93.50 | 96.82 | **98.99** | 93.97 | 96.10 | 94.32 | 95.18 | 60.53 | 84.45 | 62.26 | 96.87 | 80.10 |
| | AUC-ROC | 56.48 | 54.82 | 60.84 | **67.85** | 49.83 | 61.34 | 54.47 | 53.10 | 45.04 | 42.95 | 52.34 | 51.50 |
| | AUC-PR | 19.47 | 34.56 | 39.92 | **42.83** | 14.45 | 42.46 | 33.83 | 17.18 | 13.38 | 12.64 | 17.00 | 18.49 |
| SMAP | PA-F1 | **86.76** | 66.89 | 69.18 | 81.03 | 81.55 | 71.00 | 64.56 | 25.74 | 62.93 | 26.23 | 85.58 | 58.47 |
| | AUC-ROC | **62.61** | 58.53 | 51.24 | 47.89 | 49.83 | 40.23 | 56.46 | 51.96 | 47.99 | 51.38 | 51.97 | 54.43 |
| | AUC-PR | **24.13** | 15.21 | 12.08 | 11.90 | 4.03 | 10.56 | 13.87 | 4.07 | 3.93 | 4.18 | 7.73 | 7.93 |
| SMD | PA-F1 | **96.15** | 83.72 | 84.03 | 75.79 | 75.88 | 77.02 | 84.10 | 30.58 | 57.05 | 27.89 | 92.56 | 55.02 |
| | AUC-ROC | **74.26** | 69.88 | 72.02 | 66.28 | 49.93 | 66.57 | 70.15 | 55.76 | 52.59 | 49.78 | 65.75 | 60.53 |
| | AUC-PR | **39.44** | 11.68 | 13.16 | 11.33 | 4.58 | 11.31 | 13.32 | 5.96 | 5.45 | 5.47 | 15.72 | 11.04 |
| SWaT | PA-F1 | 83.85 | 83.72 | 89.55 | 88.08 | **97.32** | 84.44 | 86.07 | 70.26 | 72.80 | 69.93 | 90.72 | 0.00 |
| | AUC-ROC | 55.49 | 29.85 | 79.96 | **81.96** | 50.38 | 80.74 | 24.67 | 42.56 | 41.55 | 41.99 | 50.38 | 50.00 |
| | AUC-PR | 17.48 | 8.84 | 50.60 | 70.79 | 15.69 | **70.85** | 8.58 | 15.21 | 14.72 | 16.04 | 16.64 | 15.49 |
| Overall | Avg. Rank | **3.20** | 5.53 | 3.27 | 5.13 | 7.87 | 5.13 | 5.53 | 9.07 | 9.93 | 9.73 | 4.93 | 8.60 |

**Results on Univariate Benchmarks.** The experimental results, summarized in Table 5, demonstrate that AXIS achieves competitive performance across nine public univariate benchmarks. To rigorously evaluate the statistical significance of these rankings, we present a Critical Difference (CD) diagram in Figure 8. AXIS achieves a Average Rank of 3.81. This validates that our Phase I encoder learns robust temporal representations essential for anomaly detection.

**Extensibility to Multivariate Contexts.** While our current work primarily focuses on univariate settings to establish the core "step-aligned hint" reasoning mechanism, our framework is inherently extensible to multivariate contexts. Specifically, we incorporate the Any-variate Attention mechanism (Woo et al., 2024), which allows the model to learn cross-channel interactions. By integrating this mechanism, we construct AXIS-multi, which effectively handles multivariate attention by enabling each time step to receive contextual hints incorporating information from related channels. To validate this extensibility, we further evaluated AXIS-multi on five standard multivariate datasets (MSL, PSM, SMAP, SMD, SWaT). As shown in Table 6, AXIS-multi achieves the best overall performance with an Average Rank of 3.20, outperforming recent time-series foundation models (e.g., Chronos, MOMENT). This confirms that our step-aligned hint mechanism scales effectively to high-dimensional signals, capturing cross-channel dependencies without performance degradation.

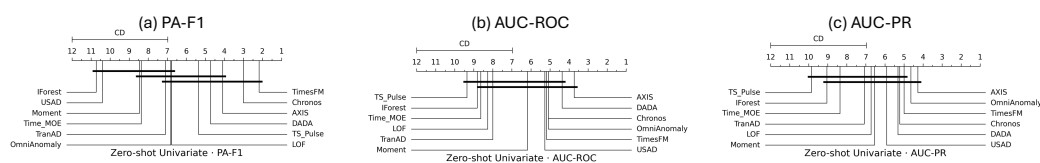

Figure 8: Critical Difference (CD) diagram comparing the statistical performance of AXIS against baselines across nine univariate datasets.

# F  BENCHMARK ANALYSIS AND VALIDITY

**Dataset Composition and Scale.** Our benchmark comprises **140 unique time series** generating **280 evaluation windows**, of which **98 (35.0%)** contain anomalies and **182 (65.0%)** are normal. The dataset spans diverse domains and exhibits significant variability in temporal structure: original series lengths range from 64 to 1,020 time steps (median: 617), while evaluation windows range from 15 to 40 steps (median: 28). This distribution ensures the benchmark tests both short-term pattern recognition and longer-context reasoning capabilities. The benchmark is carefully balanced across three question types: **open-ended (34.64%), multiple-choice (32.86%)**, and **true/false (32.50%)**. Importantly, the anomaly ratio is consistent across question types (31.87%–37.11%), preventing any systematic bias where certain question formats disproportionately test anomaly vs. normal cases.

Table 7: Performance of standard anomaly detection algorithms on benchmark synthetic data.

| Dataset | F1-PA | ROC-AUC | ROC-PRC |
|---------|-------|---------|---------|
| IForest | 0.3912 | 0.5851 | 0.1616 |
| KNN | 0.4708 | 0.6762 | 0.1993 |
| LOF | 0.4596 | 0.7011 | 0.2072 |
| PCA | 0.3537 | 0.5908 | 0.1666 |

**Difficulty Validation via Synthetic Time Series.** We evaluated four standard unsupervised anomaly detection algorithms implemented by TSB-AD on the same synthetic time series data used in our benchmark. Table 7 summarizes their performance using F1-score with point adjustment (F1-PA), ROC-AUC, and Precision-Recall AUC (ROC-PRC). None of the traditional algorithms achieve high scores across the board, with the best average F1-PA score being only 0.4708 (KNN). These metrics indicate that the underlying anomaly patterns are non-trivial and challenging for established methods to detect reliably.

# G  DETAILED PROCEDURE FOR G-EVAL

This appendix documents the evaluation methodology, criteria, and prompts used by our G-Eval variant.

## G.1  METHODOLOGICAL OVERVIEW

We adopt an advanced G-Eval judge that evaluates system answers along multiple dimensions and calibrates discrete scores using token-level log-probabilities. For each question-answer instance:

1. Determine the question type (e.g., multiple choice, open-ended, true/false) and instantiate the corresponding set of evaluation dimensions.

2. For each dimension, query a LLM judge with a criterion-specific instruction (see the exact prompt in Appx.) to produce a reasoning-based assessment and a final integer score $s \in \{1, 2, 3, 4, 5\}$. If score extraction is ambiguous, a conservative default is used.

3. Obtain token-level probabilities assigned to the score tokens $\{1, \ldots, 5\}$. Exponentiate and normalize these probabilities to construct a score distribution. Use its expectation as a log-probability-weighted score and compute a confidence via normalized entropy (Appx. G.3).

## G.2 EVALUATION CRITERIA AND WEIGHTS

For each `question_type`, the evaluator applies a fixed set of dimensions with 5-point guidelines and combines them via type-specific weights:

Table 8: Dimension Weights by Question Type.

| | |
|---|---|
| **multiple_choice** | correctness: 0.70, reasoning_quality: 0.30 |
| **open_ended** | relevance: 0.30, completeness: 0.35, accuracy: 0.35 |
| **true_false** | correctness: 0.60, justification_quality: 0.40 |

The dimension descriptions and scoring guidelines (1–5) are:

**multiple_choice**

- **correctness**: How accurate is the generated response compared to the expected answer?
  - 5: Perfect match, completely correct
  - 4: Mostly correct; minor deviations not affecting core meaning
  - 3: Partially correct; captures some key aspects but misses important details
  - 2: Somewhat relevant but with significant errors or omissions
  - 1: Incorrect or completely irrelevant

- **reasoning_quality**: How well does the response demonstrate logical reasoning and explanation?
  - 5: Clear, logical, comprehensive reasoning fully explains the choice
  - 4: Good reasoning with minor gaps
  - 3: Adequate reasoning; lacks depth or has inconsistencies
  - 2: Weak reasoning; significant gaps or flawed logic
  - 1: No clear reasoning or completely flawed logic

**open_ended**

- **relevance**: How relevant and on-topic is the generated response?
  - 5: Completely relevant; directly addresses all aspects
  - 4: Highly relevant; minor omissions
  - 3: Moderately relevant; addresses core aspects but misses details
  - 2: Somewhat relevant; off-topic content or key omissions
  - 1: Irrelevant or off-topic

- **completeness**: How complete and comprehensive is the response?
  - 5: Fully comprehensive; covers all necessary aspects
  - 4: Mostly complete; minor gaps
  - 3: Adequately complete; missing some important details
  - 2: Incomplete; significant information missing
  - 1: Very incomplete

- **accuracy**: How factually accurate is the response?
  - 5: Completely accurate; no factual errors
  - 4: Mostly accurate; very minor inaccuracies
  - 3: Generally accurate; some notable errors
  - 2: Several factual errors affecting reliability
  - 1: Major factual errors or mostly inaccurate

**true_false**

- **correctness**: How correct is the T/F judgment and supporting explanation?
    - 5: Perfect judgment; excellent supporting explanation
    - 4: Correct judgment; good explanation
    - 3: Correct judgment; adequate explanation
    - 2: Incorrect judgment; reasonable attempt at explanation
    - 1: Incorrect judgment; poor or no explanation
- **justification_quality**: How well does the response justify the decision?
    - 5: Excellent justification with clear evidence and reasoning
    - 4: Good justification with solid evidence
    - 3: Adequate justification with some evidence
    - 2: Weak justification with little evidence
    - 1: No meaningful justification

## G.3 ADVANCED SCORING FROM LOG-PROBABILITIES

Let $s \in \{1, 2, 3, 4, 5\}$ denote the raw score extracted from the LLM's evaluation output. We also obtain token-level log-probabilities for score tokens $\{1, \ldots, 5\}$. The evaluator constructs a distribution over scores by exponentiating and normalizing these log-probabilities:

$$p(s) \propto \exp(\log p_s), \quad \tilde{p}(s) = \frac{\exp(\log p_s)}{\sum_{k=1}^{5} \exp(\log p_k)}.$$

The **weighted score** is the expectation under $\tilde{p}$:

$$\text{WeightedScore} = \sum_{s=1}^{5} s \cdot \tilde{p}(s).$$

## G.4 ADVANCED G-EVAL PROMPT

```
You are an expert evaluator for natural language generation systems. Your
    task is to evaluate the quality of generated responses using the G-
    Eval methodology with chain-of-thought reasoning.

Control the Maximum Length to 500 words.

**Evaluation Criterion: {criterion.dimension}**
{criterion.description}

**Scoring Guidelines:**
{guidelines_text}

**Question:** {question}

**Expected Answer:** {expected_answer}

**Generated Response:** {generated_response}

**Instructions:**
1. Analyze the generated response step by step using chain-of-thought
    reasoning
2. Compare it against the expected answer for the specified criterion
3. Consider both content quality and alignment with the expected answer
4. Provide detailed reasoning for your evaluation
5. Conclude with a single score from 1-5
6. Ignore error in index mismatch, just focus on the content

Please follow this exact format for your response:
```

```
**Step-by-step Analysis:**
[Provide detailed chain-of-thought analysis]

**Comparison with Expected Answer:**
[Compare generated response with expected answer]

**Final Assessment:**
[Summarize your evaluation]

**Score:** [Single integer: 1, 2, 3, 4, or 5]
```

### G.5 DETAILED RESULTS OF G-EVAL

**Visualize and pair-wised comparsion** Figures 9 to 11 present forest plots to visually summarize the pairwise model performance comparisons from Tables 1 to 3. The confidence intervals are calculated from the variance of the pairwise total scores.

As shown in Fig. 9, our AXIS model demonstrates a strong performance. On Multiple-choice questions, it significantly outperforms all models except for Image-LLM and Anon-LLM(Window). For True/False questions, AXIS significantly surpasses all other models, highlighting its advantage in objective answering. Furthermore, on Open-Ended questions, AXIS outperforms other models while achieving performance comparable to the General LLM (AnomLLM).

Fig. 10 details an ablation study of our proposed AXIS model. The results demonstrate that AXIS consistently and significantly outperforms its variants ("w/o-windows", "w/o-task-hint", and "w/o-context-hint") across all three question categories. The confidence intervals for the mean paired difference are entirely below zero in all subplots, indicating that the removal of any of these components leads to a statistically significant degradation in performance. This underscores the integral contribution of each component to the overall efficacy of the AXIS model.

Fig. 11 provides a comparative analysis of various open-source models against the Deepseek Llama 8B (Inst) baseline. In multiple-choice questions, Deepseek Llama 8B shows a significant performance advantage over the Mistral 7B models, while its performance is statistically comparable to the Qwen2.5 7B series. For open-ended and true/false questions, Deepseek Llama 8B shows comparable performance to most Qwen and Deepseek-Qwen variants, showing that the robustness of our AXIS design.

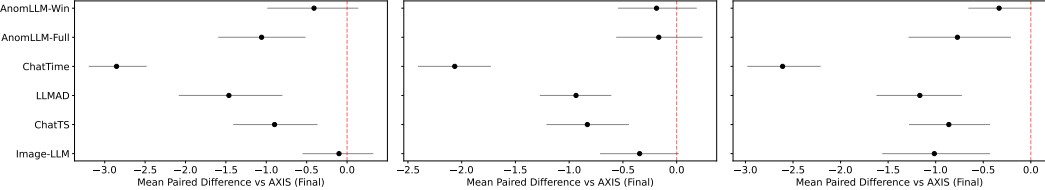

Figure 9: Comparative analysis of baseline models against AXIS via forest plot. Each horizontal line represents the 95% bootstrap confidence interval for the mean paired score difference relative to the AXIS baseline. The central dot marks the point estimate of the mean difference. The vertical red dashed line at zero indicates no difference; confidence intervals crossing this line suggest that the model's performance is not statistically different from the baseline.

**Analysis for different LLM judge:** We introduce gpt-4.1 as the second LLM judge to complement our primary judge, Gemini-2.5. We then conduct a comprehensive inter-rater reliability analysis using a human-annotated subset. Our analysis, summarized in Table 9, confirms the robustness of our evaluation methodology. First, we observe high consistency between our two LLM judges, Gemini-2.5 and Gpt-4.1, which achieve an overall Kendall's W of 0.882. This demonstrates that our evaluation prompts and criteria are stable and reproducible across different state-of-the-art models. Second, and more importantly, both LLM judges show strong alignment with human expert ratings. Our primary judge, Gemini-2.5, achieves a strong agreement of W=0.743 with human ratings, while

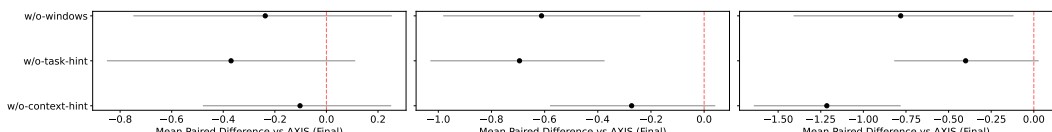

Figure 10: Forest plot comparing AXIS variants against the AXIS baseline. The plot displays the mean paired score difference and the 95% bootstrap confidence interval for each variant. The vertical red line at zero indicates no performance difference relative to AXIS.

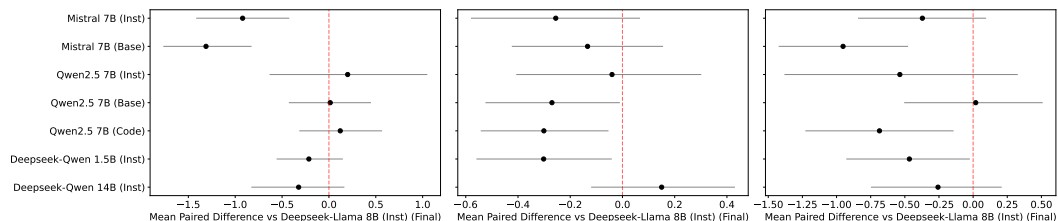

Figure 11: Forest plot for ablation studies, benchmarked against AXIS. This plot illustrates the mean paired score difference and the corresponding 95% bootstrap confidence interval for each ablated variant. The vertical red line at zero serves as the reference for no difference from the baseline.

our second judge, Gpt-4.1, also achieves a strong agreement of W=0.750. This alignment remains robust even for the most challenging open-ended questions, where the judges score W=0.670 and W=0.679, respectively. To contextualize these results, we measure the inter-rater agreement among the human experts themselves, finding a Kendall's W of 0.587. This indicates that our LLM judges not only agree with each other but also perform at a level of consistency comparable to human experts for this subjective task. We report all correlation coefficients using Kendall's W, as shown in the table below.

Table 9: Inter-rater agreement (Kendall's W) between LLM judges and human experts.

| Question Type | Gpt-4.1 vs. Human | Gemini-2.5 vs. Gpt-4.1 | Gemini-2.5 vs. Human | Human Inter-rater Agreement |
|---|---|---|---|---|
| Overall | 0.750 | 0.882 | 0.743 | 0.587 |
| Multiple Choice | 0.857 | 0.833 | 0.750 | 0.667 |
| Open-Ended | 0.679 | 0.912 | 0.670 | 0.587 |
| True/False | 0.749 | 0.856 | 0.784 | 0.535 |

### G.6 COMPUTATIONAL EFFICIENCY ANALYSIS

We analyze the computational overhead of AXIS compared to baseline methods by measuring average prompt token counts and generation latency across question types (see Table 11).

**Token Efficiency.** AXIS (14B) demonstrates high token efficiency, averaging ∼335 prompt tokens. This is comparable to `ChatTS` (14B) (∼363 tokens) and significantly more efficient than text-serialization approaches like `LLMAD` (∼1370 tokens) and `Image LLM` (∼1479 tokens). While `AnomLLM(Window)` uses the fewest tokens, it sacrifices reasoning capability by limiting context to a local window.

**Latency and Performance Balance.** AXIS achieves a favorable balance between speed and performance. It incurs only a modest latency overhead (∼1.5–2s) compared to `ChatTS` while delivering superior reasoning, and is ∼7× faster than `ChatTime`, which suffers from prohibitive latency. Both token counts and latencies remain consistent across question types, indicating robust efficiency for diverse reasoning tasks.

Table 10: G-eval results by LLM judge gpt-4.1

| Model | Multiple Choice | | | Open Ended | | | | True False | | |
| --- | --- | --- | --- | --- | --- | --- | --- | --- | --- | --- |
| | Final | Corr. | Rsn. Qual. | Final | Rel. | Comp. | Acc. | Final | Corr. | Justif. |
| AXIS | **4.52** | **4.56** | **4.49** | **3.54** | **3.67** | **3.29** | **3.65** | **3.94** | **3.71** | **4.17** |
| Image LLM | 4.40 | 4.35 | 4.44 | 3.40 | 3.58 | 3.24 | 3.38 | 2.98 | 2.71 | 3.24 |
| ChatTS | 3.71 | 3.77 | 3.65 | 3.11 | 3.27 | 2.95 | 3.11 | 3.05 | 2.90 | 3.19 |
| LLMAD | 3.66 | 3.30 | 4.02 | 2.86 | 2.93 | 2.80 | 2.85 | 3.50 | 3.31 | 3.69 |
| ChatTime | 1.47 | 1.91 | 1.02 | 1.07 | 1.02 | 1.04 | 1.15 | 1.10 | 1.17 | 1.02 |
| AnomLLM(Full) | 4.23 | 4.12 | 4.35 | 3.13 | 3.35 | 2.98 | 3.05 | 2.76 | 2.69 | 2.83 |
| AnomLLM(Window) | 2.80 | 2.81 | 2.79 | 2.93 | 3.05 | 2.58 | 3.15 | 2.55 | 2.57 | 2.52 |

Table 11: Comparison of Computational Efficiency: Prompt Tokens, Latency, and Memory

| Model | Multiple Choice | | | Open Ended | | | True False | | |
| --- | --- | --- | --- | --- | --- | --- | --- | --- | --- |
| | Tokens | Lat (s) | Mem (GB) | Tokens | Lat (s) | Mem (GB) | Tokens | Lat (s) | Mem (GB) |
| **Image LLM** | 1548.42 | 9.57 | - | 1455.33 | 10.83 | - | 1433.33 | 9.21 | - |
| **LLMAD** | 1364.42 | 2.44 | - | 1375.16 | 2.36 | - | 1372.00 | 2.44 | - |
| **AnomLLM(Full)** | 3083.81 | 1.75 | - | 3075.64 | 2.01 | - | 2983.40 | 1.82 | - |
| **AnomLLM(Window)** | 312.53 | 0.73 | - | 258.00 | 0.74 | - | 240.43 | 0.70 | - |
| **ChatTime 14B** | 288.49 | 31.06 | 38.5 | 182.45 | 28.88 | 37.4 | 181.05 | 29.15 | 38.1 |
| **ChatTS 14B** | 411.95 | 5.02 | 32.3 | 347.24 | 5.77 | 31.9 | 330.10 | 4.75 | 32.1 |
| **AXIS 14B** | 376.05 | 13.70 | 41.6 | 321.29 | 13.76 | 40.5 | 303.90 | 12.98 | 41.0 |

# H PROCEDURE AND DETAIL RESULTS FOR HUMAN-BASED EVALUATION

## H.1 DESIGNS OF EXPERIMENTS

**Questionnaire Design and Structure.** Our human expert questionnaire is systematically designed to evaluate model explanations across multiple dimensions with rigorous controls for bias and consistency. Each questionnaire comprises: (1) **Question Information**: the original question, expected answer, and question type classification (multiple choice, open-ended, or true/false); (2) **Time Series Visualization**: a plot showing the full time series with the target window highlighted to provide visual context for assessment; (3) **Model Responses**: all baseline model outputs presented in a randomized order to eliminate position bias; (4) **Evaluation Criteria**: dimension-specific scoring guidelines adapted from the G-Eval methodology, employing 5-point Likert scales for correctness, reasoning quality, relevance, completeness, accuracy, and justification quality tailored to each question type; (5) **Scoring Tables**: structured evaluation forms for systematic criterion-based assessment; (6) **Model Ranking**: comparative ranking of all models with written justifications to capture qualitative insights.

**Expert Recruitment and Annotation Protocol.** For this study, we evaluate 140 unique questions, resulting in 280 completed questionnaires. We recruit 28 human experts through a rigorous vetting process targeting researchers with established expertise in time-series analysis and machine learning. All annotators hold advanced degrees (Ph.D. candidates or post-doctoral fellows) and are recruited from our institution and collaborating academic research laboratories. Experts are compensated at standard academic research rates for their time and effort. To ensure annotation quality and mitigate individual bias, we implement a double-blind protocol wherein each questionnaire is independently evaluated by three randomly assigned experts from the pool of 28, with neither annotator aware of the other's identity or responses. To quantify inter-rater reliability, we compute Kendall's W (Coefficient of Concordance), which is well-suited for measuring agreement among multiple raters on ordinal data. Our analysis yields Kendall's W = 0.587, indicating a "moderate" to "substantial" level of agreement among our expert panel. This coefficient is notably comparable to inter-rater agreement levels reported in similar expert evaluation studies, confirming that our human-annotated dataset provides a reliable and internally consistent gold standard for model assessment.

---

**Human Expert Evaluation Questionnaire Example**

**Question 1**
**Question Type:** multiple_choice
**Question:** Which of the following best describes the likelihood of detecting an anomaly within the time series window from step 491 to 521, given its observed behavior?
A) There is a sudden spike at the start of the window that indicates potential anomalous activity B) The window displays periodic irregularities that suggest subtle anomalies C) The window exhibits consistent and stable behavior, suggesting no detectable anomalies D) The pattern shows gradual drift, implying an evolving anomaly towards the window's end
**Expected Answer:** C) The window exhibits consistent and stable behavior, suggesting no detectable anomalies. The current_value data fluctuates around zero with both positive and negative values but does not display any abrupt spikes, sustained trends, or irregular periodicities...
**Time Series Visualization:** [Plot showing full time series with highlighted window]
**Model Responses:**
**Model 1** To determine anomaly likelihood in the window from step 491 to 521, let's examine the values: [0.12, -2.45, -1.15, ...]. Within this window, there is a sharp drop from 0.12 to -2.45 at the very start... **A) There is a sudden spike at the start of the window that indicates potential anomalous activity**
**Model 2** Within the specified window (indices 491 to 521), the data shows fluctuating values without any abrupt spikes or consistent irregular patterns...
**Model 3** The observed behavior within the time series window from step 491 to 521 indicates no sudden spikes or abrupt changes... **C) The window exhibits consistent and stable behavior, suggesting no f anomalies.**
**Model 4** The time series visualization highlights the analysis window from step 491 to 521... **C) The window exhibits consistent and stable behavior, suggesting no detectable anomalies.**
**Model 5** {"briefExplanation": {"step1_global": "The overall trend of the latest 31 data points is stable...", "is_anomaly": false, "anomalies": [], ...}
**Model 6** **Answer: C) The window exhibits consistent and stable behavior, suggesting no detectable anomalies.** The time series window from step 491 to 521 shows a mix of positive and negative values with moderate fluctuations, but there are no sudden spikes...
**Evaluation Criteria:**
**Correctness (1-5):** How accurate is the generated response compared to the expected answer? **Reasoning Quality (1-5):** How well does the response demonstrate logical reasoning and explanation?

| | Model | Correctness | Reasoning Quality | Comments |
|---|---|---|---|---|
| | Model 1 | --- | --- | --- |
| | Model 2 | --- | --- | --- |
| **Scoring Table:** | Model 3 | --- | --- | --- |
| | Model 4 | --- | --- | --- |
| | Model 5 | --- | --- | --- |
| | Model 6 | --- | --- | --- |

**Model Ranking:** Rank all models from best (1) to worst (6) with justifications.

---

H.2 DETAILED RESULTS FOR HUMAN EVALUATION

Fig. 12 provides a criterion-level breakdown of human evaluation across the three question types. AXIS ranks first consistently. On Multiple Choice questions, it achieves the best scores in both *Correctness* and *Reasoning Quality*, with a clear margin over the next best visual baseline. This indicates that our hint-based conditioning not only selects the right option but also articulates concise, logically grounded rationale.

For Open-Ended questions, AXIS leads on *Relevance*, *Completeness*, and *Accuracy*, reflecting faithful, fully supported explanations rather than surface descriptors. On True/False questions, it also tops both *Correctness* and *Justification Quality*, showing strong calibration and evidence-backed decisions. Overall, these results demonstrate that the proposed three-pathway design (symbolic

numeric grounding, context-integrated local hints, and task-prior hints) confers robust advantages across formats—surpassing specialized TS-LLMs and multimodal vision–language approaches in both accuracy and human-judged explanatory quality.

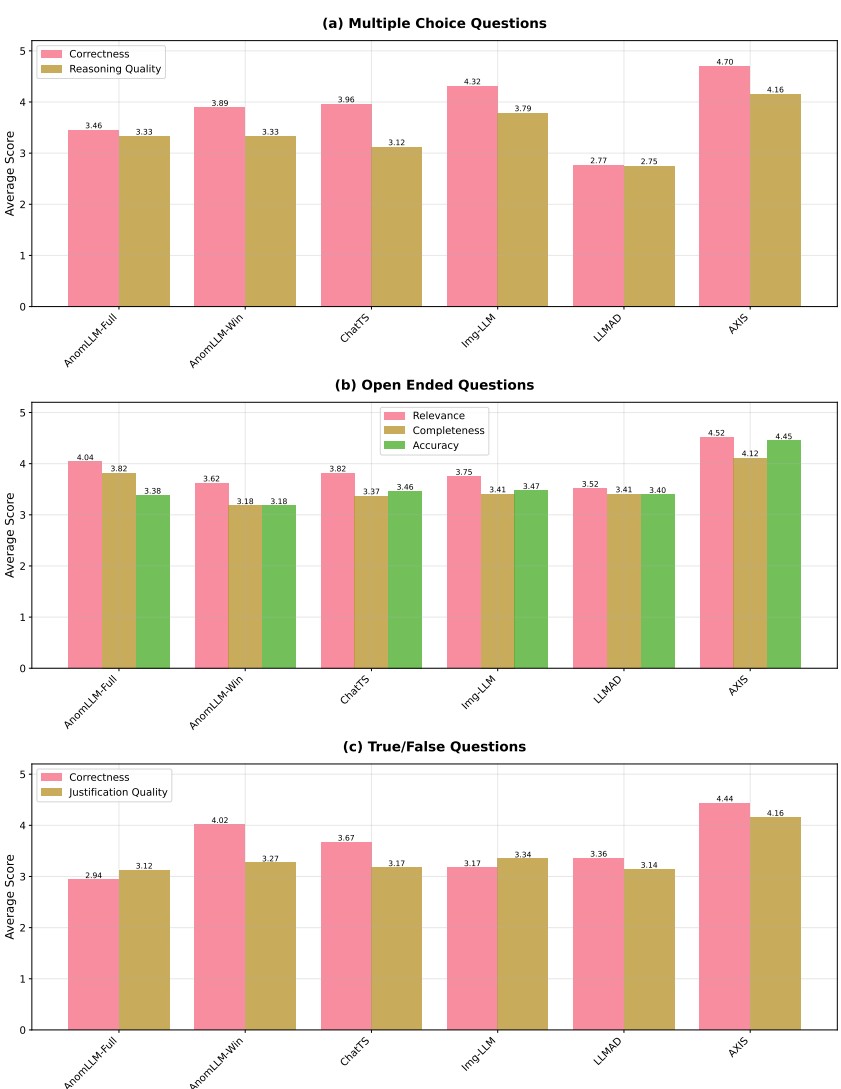

Figure 12: The mean scores by human evaluation for (a) multiple choice question, (b) open ended questions and (c) true/false questions

## H.3 BENCHMARK CONSTRUCTION & EVALUATION FOR PUBLIC DATASETS

To evaluate the generalization capabilities of our model on real-world data, we construct a comprehensive human evaluation benchmark using 108 time series from three public datasets: YAHOO, TODS, and NEK. These datasets cover diverse domains including web traffic, distributed system metrics, and sensor data. For each time series, we design two distinct question formats: a True/False (TF) question and a Multiple-Choice Question (MCQ), to assess the quality of generated explanations from different perspectives.

The benchmark construction process begins with a deterministic, seed-based sampling procedure to select two representative windows from each of the 108 time series, ensuring reproducibility. For each series, we sample:

- **An Anomaly Window**: A segment of 15 to 40 time steps that contains at least one point labeled as an anomaly.
- **A Normal Window**: A segment of the same length range (15-40 steps) containing exclusively normal data points.

Each selected window is visualized by highlighting it on a plot of the entire time series, providing essential visual context for the evaluation. These selected windows and their corresponding visualizations form the basis for both the TF and MCQ tasks.

**True/False (TF) Question Generation**    For the TF task, we formulate a standardized question for every window: *"True or False: The time series window from step {start} to {end} shows only normal behavior, with no evidence of sudden deviations or irregular patterns that would indicate an anomaly."* The ground truth answer is "False" for anomaly windows and "True" for normal windows. The ground truth explanation for an anomaly window is derived from a predefined set of detailed descriptions corresponding to expert-provided labels (e.g., "downward spike," "sudden increase"), while the explanation for a normal window confirms the absence of anomalous patterns.

**Multiple-Choice Question (MCQ) Generation**    Using the same set of windows, we design an MCQ task to assess a model's ability to not only detect but also correctly classify anomalies. The question asks: *"Which of the following best describes the anomaly detection results within the time series window spanning steps {start} to {end}?"* For each question, we generate three distinct options:

- **The Correct Option**: This option accurately describes the window's content. For an anomaly window, it corresponds to the specific, expert-labeled anomaly type (e.g., "upward spike"). For a normal window, it is the "normal" option.
- **Distractor Options**: Two incorrect options are provided to serve as distractors. If the window is anomalous, the options include "normal" and another randomly selected anomaly type from the overall pool of labels. If the window is normal, the distractors are two different, randomly selected anomaly types.

Crucially, each option is presented not just as a short label but as a full descriptive sentence (e.g., *"Upward Spike anomaly: There is a sharp upward jump in the measurements."*) to ensure clarity and prevent ambiguity. This design forces the model to discern nuanced differences between various anomaly types and normal behavior.

**Evaluation Criteria**    To ensure a granular assessment of the models' explanatory power, we establish a detailed scoring rubric for both TF and MCQ tasks, administered by expert human annotators. For the **True/False task**, we use a 6-point scale (0–5) to evaluate both the correctness of the binary answer and the quality of the accompanying justification. A score of 5 represents a correct T/F judgment with a clear, accurate, and comprehensive explanation. Lower scores are assigned for answers with less precise, generic, or irrelevant justifications. A score of 0 indicates an incorrect T/F judgment with a nonsensical explanation. For the **Multiple-Choice task**, scores are assigned based on the model's ability to not only detect but also correctly classify the anomaly:

- **4–5 (Correct Classification):** A score of 5 is awarded for correctly identifying the specific anomaly type (or 'normal') with a clear and accurate explanation. A score of 4 is given for a correct classification with a more generic or slightly flawed explanation.
- **2–3 (Correct Detection, Incorrect Classification):** A score of 3 is assigned if the model correctly identifies the window as anomalous but selects the wrong anomaly type, providing a plausible justification for its incorrect choice. A score of 2 is given for the same error but with a poor or irrelevant explanation.
- **0 (Incorrect Detection):** A score of 0 is given if the model fails to correctly determine whether an anomaly is present (e.g., classifying an anomalous window as 'normal' or vice versa).

This multi-faceted evaluation framework allows us to capture nuanced differences in the quality of explanations beyond simple accuracy metrics.

## I  CAUSAL ABLATION ANALYSIS FOR HINT TOKENS

To analyze the individual contribution of each local hint token to the model's generation quality, we implement a causal ablation study. This method systematically removes or modifies specific hint tokens to quantify their importance in producing accurate explanations.

Given hint embeddings $\tilde{\mathbf{H}}_{s:e} \in \mathbb{R}^{(e-s) \times d_h}$, we measure the contribution of each local hint token at position $i$ within the target window $[s, e)$ through the following procedure:

**Baseline Computation.**  First, we compute the baseline log-likelihood using the complete hint embeddings:

$$\mathcal{L}_{\text{baseline}} = -\log P(\mathbf{y}|\tilde{\mathbf{H}}_{s:e}, q, \tilde{\mathbf{F}}),$$

where $\mathbf{y}$ is the target explanation, $q$ is the query, and $\tilde{\mathbf{F}}$ represents the task-prior hints.

**Ablation Methods.**  For each position $i \in [s, e)$, we create an ablated version of the hint embeddings $\tilde{\mathbf{H}}_{s:e}^{(i)}$ using **Zero Replacement:** $\tilde{\mathbf{H}}_{s:e}^{(i)}[i - s] = \mathbf{0}$.

**Contribution Score.**  The contribution score for position $i$ is computed as:

$$C_i = \mathcal{L}_{\text{baseline}} - \mathcal{L}_{\text{ablated}}^{(i)},$$

where $\mathcal{L}_{\text{ablated}}^{(i)} = -\log P(\mathbf{y}|\tilde{\mathbf{H}}_{s:e}^{(i)}, q, \tilde{\mathbf{F}})$. A positive $C_i$ indicates that removing the hint at position $i$ degrades the model's performance, suggesting that this position contributes positively to explanation generation.

**Validation of Context-Integrated Hints via Causal Ablation Analysis.**  As depicted in Fig. 13 and Fig. 14, the causal ablation analysis provides empirical evidence for the efficacy of our proposed hint mechanism. A predominant observation across both examples is that the majority of the contribution scores, denoted by $C_i$, are positive ($C_i > 0$). According to the formulation presented in Appx. I, a positive $C_i$ indicates that ablating the hint token at position $i$ degrades the model's performance, as measured by an increase in the negative log-likelihood of the target explanation. This finding strongly supports our hypothesis that the context-integrated, step-aligned hints furnish valuable guidance for the model, making a net positive contribution to the generation of high-quality answers.

Furthermore, a more nuanced pattern emerges from the results. We observe that in regions where the time series exhibits significant volatility or sharp fluctuations, the corresponding contribution scores are comparatively lower. This suggests that the inherent complexity and unpredictability of volatile segments in the time series can diminish the marginal utility of individual hint tokens. In essence, while the hints remain beneficial overall, their directional impact is partially mitigated by the increased difficulty of the task in these challenging temporal regions.

## J  ADDITIONAL SENSITIVITY AND ROBUSTNESS ANALYSES

**Analysis of Distractor Design in Multiple Choice Questions.**  We investigate the hypothesis that the divergent impact of the context-integrated hint across question types stems from the distinct reasoning processes required. While True/False questions demand binary judgments often necessitating global context, Multiple Choice Questions (MCQs) allow for elimination strategies. To validate this, we constructed an evaluation set of 140 MCQs with varying numbers of options ($N \in \{2, 3, 4\}$). Table 12 presents the performance of AXIS and its ablated variants. We observe that with fewer options (2 or 3), where choices are broader, the performance gap between the full model and the 'w/o-context-hint' variant is significant. As the number of options increases to 4, distractors become more specific, allowing the model to rely more on localized features (symbolic numeric hints) for elimination, thereby reducing the relative impact of the global context hint. This confirms that the utility of context hints in MCQs is modulated by distractor complexity.

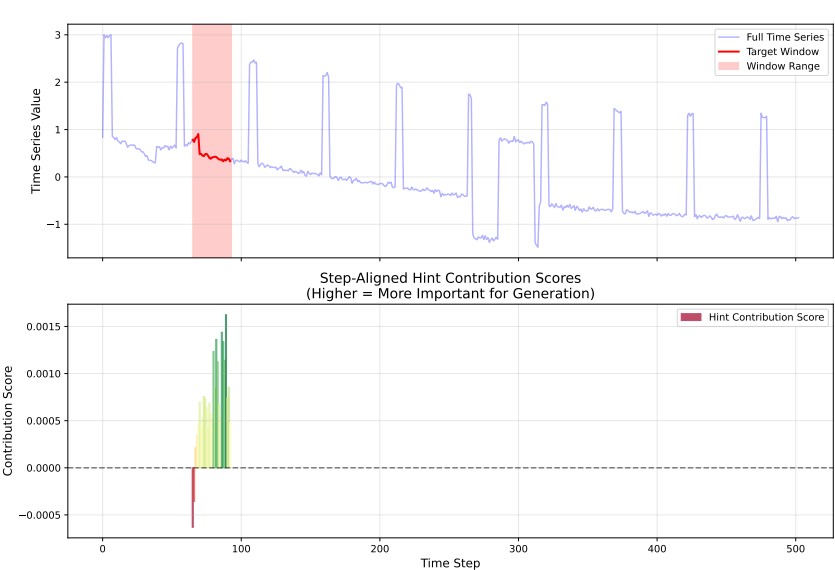

Figure 13: Causal Ablation Analysis of Step-Aligned Hint Tokens. The figure visualizes the contribution score $C_i$ for each hint token, computed via our causal ablation method, alongside the original input time series.

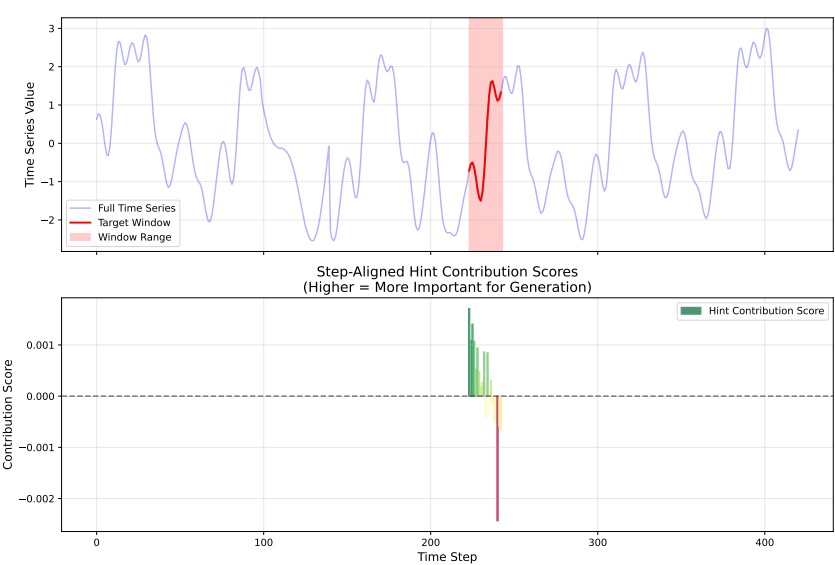

Figure 14: Causal Ablation Analysis of Step-Aligned Hint Tokens. The figure visualizes the contribution score $C_i$ for each hint token, computed via our causal ablation method, alongside the original input time series.

Table 12: Impact of Distractor Options on Model Performance across Ablations

| Model | 2 Options | 3 Options | 4 Options |
|---|---|---|---|
| **AXIS** | $3.961 \pm 0.304$ | $3.874 \pm 0.372$ | $3.983 \pm 0.344$ |
| w/o-context-hint | $3.425 \pm 0.336$ | $3.371 \pm 0.332$ | $3.791 \pm 0.334$ |
| w/o-fixed-hint | $3.249 \pm 0.346$ | $3.369 \pm 0.419$ | $3.772 \pm 0.381$ |
| w/o-windows | $3.507 \pm 0.299$ | $3.823 \pm 0.311$ | $3.734 \pm 0.344$ |

**Analysis of Window Length Sensitivity.** We further analyze the impact of the input question-window length $[s, e]$ on performance by varying it from 10 to 160 time steps. Figure 15 illustrates the sensitivity patterns across question types. For Multiple Choice and Open-Ended questions, performance follows an inverted U-shaped trend. Initial increases in window size improve reasoning by providing more information, but performance declines beyond 40-80 steps, likely due to the processing burden of long numerical sequences on the LLM. In contrast, True/False questions exhibit stability across window sizes. This robustness suggests that binary judgments rely more on the global semantic context provided by the 'context-integrated hint' rather than precise local numerical patterns, making them less sensitive to the granularity of the windowed input.

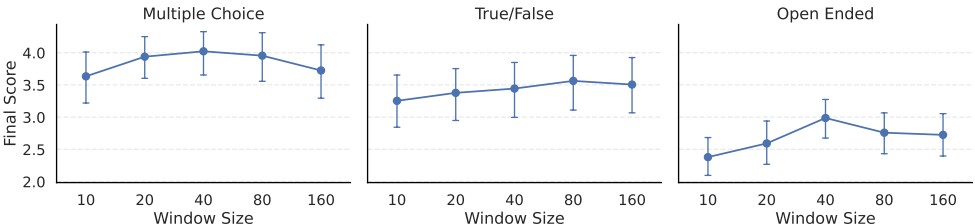

Figure 15: Performance sensitivity to variations in question-window length across different question types.

**Robustness Against Global Noise Injection.** To assess model robustness without introducing ambiguity often caused by local perturbations in anomaly detection tasks, we evaluate performance under global Gaussian noise injection. We add zero-mean Gaussian noise $\epsilon \sim \mathcal{N}(0, \sigma^2)$ with scales $\sigma \in \{0.1, 0.3, 0.7, 1.5\}$ to the input series. As shown in Figure 16, the model demonstrates strong robustness in discriminative tasks (Multiple Choice and True/False), maintaining stable performance up to noise scales of 0.7. This indicates effective filtering of high-frequency noise to focus on dominant anomalous features. Conversely, the generative Open-Ended task shows higher sensitivity, with performance decreasing monotonically as noise increases, reflecting the challenge of generating detailed semantic descriptions from obscured signal morphologies.

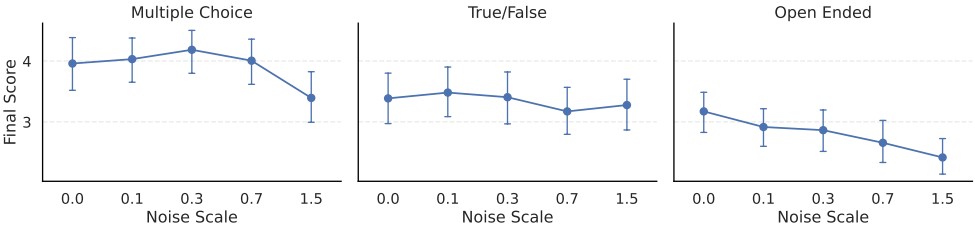

Figure 16: Model robustness evaluation under varying scales of global Gaussian noise injection.