# OpenReview forum: "AXIS: Explainable Time Series Anomaly Detection with Large Language Models"
_ICLR.cc/2026/Conference — Submitted to ICLR 2026_

### Official Review · Reviewer_ERp9 · 2025-10-27

**Soundness:** 2
**Presentation:** 3
**Contribution:** 3
**Rating:** 4
**Confidence:** 3

**Summary:**

This paper presents a novel approach to the time-series anomaly detection problem. The authors reformulate the task by focusing on interval-based anomalies rather than point-based ones, motivated by the observation that anomalies in real-world time series often occur over contiguous intervals instead of isolated timestamps. The problem is defined as follows: given a univariate time series and a natural-language query specifying an interval, the goal is to determine whether the interval is anomalous and to provide an explanation for this decision. To address this problem, the authors propose AXIS, an LLM-based framework designed to handle continuous time-series data. Since standard LLMs are not well-suited for processing continuous signals, AXIS introduces three complementary representation pathways, referred to as hints: the symbolic numerical hint, the context-integrated step-aligned hint, and the task-prior hint. These pathways aim to bridge the gap between time-series representations and language-based reasoning. The training process consists of two phases. In the first phase, the time-series encoder used in the context-integrated step-aligned hint is pre-trained to capture temporal dependencies. In the second phase, both the encoder and the LLM are kept frozen while the remaining parameters are optimized to align the representations with the anomaly explanation task. The paper also introduces a new benchmark for semantic time-series anomaly explanation, designed to evaluate both detection and interpretability. Experimental results show that AXIS outperforms baseline methods on this benchmark. Additionally, an ablation study is conducted to assess the contributions of individual components, providing evidence for the effectiveness of the proposed design choices.

**Strengths:**

- The proposed problem definition is novel and conceptually appealing. In particular, the formulation based on intervals rather than single points is well motivated, as anomalies in real-world time series typically extend over contiguous intervals instead of isolated timestamps. Furthermore, framing the task as generating explanations adds interpretability and makes the objective of the model more aligned with practical anomaly analysis needs.

- The paper clearly articulates one of the key challenges in applying LLMs to time-series data: LLMs operate on discrete tokens and therefore struggle to directly process long, continuous temporal signals. To address this issue, the authors propose a novel prompt construction strategy within the AXIS framework, which integrates three complementary representation pathways:
    - a symbolic numeric hint, providing numerical grounding for continuous values;
    - a context-integrated, step-aligned hint, distilled from a pre-trained time-series encoder to capture fine-grained temporal dynamics; and
    - a task-prior hint, encoding global anomaly characteristics and prior knowledge relevant to the task.

- The experimental results presented in Table 1 demonstrate that the proposed AXIS framework outperforms baseline methods on the newly introduced benchmark.

- The ablation study reported in Table 2 provides evidence that the proposed prompt design contributes significantly to performance improvements, confirming the effectiveness of the multi-hint representation strategy.

- The model architecture analysis in Table 3 indicates that the proposed approach is relatively robust across different LLM architectures, although performance variations are observed among specific models.

- Overall, the paper is clearly written and well structured. It provides sufficient background and motivation to understand the problem setting and the contributions. Figure 2 offers a concise and informative summary of the differences between the proposed approach and existing LLM-based methods for time-series anomaly detection.

**Weaknesses:**

- The main concern lies in the generation process and overall quality of the proposed benchmark. Although the paper introduces a new semantic anomaly explanation benchmark, its characteristics are not thoroughly analyzed. For instance, the paper could include descriptive statistics of the datasets, such as the number of samples, anomaly distributions, and interval lengths. It would also be useful to report how simple baseline heuristics perform on these datasets to better contextualize the difficulty and validity of the benchmark. Since all experimental results are evaluated exclusively on this benchmark, a more detailed analysis is crucial to assess the reliability and generalizability of the reported findings.

- The justification for the two-phase training strategy remains unclear. The paper would benefit from a more detailed explanation, supported by either theoretical reasoning or empirical evidence, to demonstrate why this specific training scheme is necessary or advantageous compared to alternative training strategies.

- The paper does not discuss how robust the proposed model is to input noise or data corruption. It would be valuable to analyze the model’s sensitivity to noise in the time series, such as testing whether performance remains stable when individual values or small segments are perturbed or corrupted. Such an analysis would provide additional insight into the robustness and practical applicability of the proposed approach.

**Questions:**

- My primary concern relates to the generation process and overall quality of the proposed benchmark, as discussed in the weaknesses section. I would be inclined to increase my evaluation score if the paper provided additional analyses or evidence validating the reliability and soundness of the benchmark.

---

> ### Author Response · Authors · 2025-11-23
> **Response to Reviewer ERp9 (Part 1/5)**
>
> ### Weaknesses
> > 1.1 The main concern lies in the generation process and overall quality of the proposed benchmark. Although the paper introduces a new semantic anomaly explanation benchmark, its characteristics are not thoroughly analyzed. For instance, the paper could include descriptive statistics of the datasets, such as the number of samples, anomaly distributions, and interval lengths. It would also be useful to report how simple baseline heuristics perform on these datasets to better contextualize the difficulty and validity of the benchmark.
> **Response:**
>
> We thank the reviewer for this constructive suggestion. We agree that comprehensive descriptive statistics are essential for understanding the benchmark's composition and characteristics. Below, we summarize the key characteristics of our benchmark:
>
> **Dataset Composition and Scale:**
>
> Our benchmark comprises **140 unique time series** generating **280 evaluation windows**, of which **98 (35.0%)** contain anomalies and **182 (65.0%)** are normal. The dataset spans diverse domains and exhibits significant variability in temporal structure: original series lengths range from 64 to 1,020 time steps (median: 617), while evaluation windows range from 15 to 40 steps (median: 28). This distribution ensures the benchmark tests both short-term pattern recognition and longer-context reasoning capabilities.
>
> | **Metric** | **All Windows** | **Anomaly Windows** | **Normal Windows** |
> |------------|-----------------|---------------------|--------------------|
> | Count | 280 | 98 (35.0%) | 182 (65.0%) |
> | Window Length (Median) | 28 | 29 | 27 |
> | Window Length (Range) | 15–40 | 15–40 | 15–40 |
>
> **Question Type Distribution and Anomaly Balance:**
>
> The benchmark is carefully balanced across three question types: **open-ended (34.64%)**, **multiple-choice (32.86%)**, and **true/false (32.50%)**. Importantly, the anomaly ratio is consistent across question types (31.87%–37.11%), preventing any systematic bias where certain question formats disproportionately test anomaly vs. normal cases. Additionally, **33.93%** of windows contain exactly one anomaly description, while a small fraction (1.07%) contains multiple co-occurring anomalies, reflecting real-world complexity.
>
> These statistics demonstrate that our benchmark provides a **diverse, balanced, and realistic** testbed for evaluating semantic anomaly explanation capabilities across varied temporal scales, anomaly prevalence rates, and reasoning formats.
>
> **Heuristic Baselines for Anomaly Detection Difficulty:**
>
> We evaluate four basic heuristic baselines, which are standard unsupervised anomaly detection algorithms implemented by TSB-AD on the same synthetic time series data used in our benchmark to contextualize the difficulty and validity of the benchmark. The table below summarizes their performance, averaged across all time series, using widely recognized metrics: F1-score with point adjustment (F1-PA), ROC-AUC, and Precision-Recall AUC (ROC-PRC).
>
> | **Dataset** | **F1-PA** | **ROC-AUC** | **ROC-PRC** |
> |-------------|:---------:|:-----------:|:-----------:|
> | IForest |  0.3912   |   0.5851    |   0.1616    |
> | KNN     |  0.4708   |   0.6762    |   0.1993    |
> | LOF     |  0.4596   |   0.7011    |   0.2072    |
> | PCA     |  0.3537   |   0.5908    |   0.1666    |
>
> None of the traditional algorithms achieve high scores across the board. For instance, the best average F1-PA score is only **0.4708 (KNN)**, and the highest ROC-AUC is **0.7011 (    LOF)**. These metrics, while respectable, are far from perfect, indicating that the underlying anomaly patterns are non-trivial and challenging for established methods to detect reliably.

---

> ### Author Response · Authors · 2025-11-23
> **Response to Reviewer ERp9 (Part 2/5)**
>
> **Heuristic Baselines for Anomaly Explanation Difficulty:**
>
> Furthermore, to better contextualize the benchmark's difficulty as suggested, we implement two new baseline models:
>
> **Baseline 1: Random Template Baseline:** This model serves as a non-learning lower bound. It randomly selects an answer for each question and embeds it within a sophisticated-sounding but entirely fabricated explanation. For instance, it generates random metrics (e.g., "z-score of 2.7") and weaves them into pre-written templates. This baseline tests a model's ability to discern valid reasoning from plausible nonsense.
>
> **Baseline 2: Simple Heuristic Baseline:** This model implements a classic, rule-based approach. It analyzes each window using statistical features like z-scores, mean shifts, and volatility ratios, flagging an anomaly if multiple metrics exceed predefined thresholds. Its explanations are directly derived from these calculations, providing a transparent, non-semantic benchmark for anomaly detection.
>
> To contextualize the benchmark's difficulty and validate its discriminative power, we evaluate two heuristic baselines alongside our main models:
>
> | **Model** | **Multiple Choice** | | | **Open Ended** | | | | **True/False** | | |
> |-----------|---------------------|--|--|----------------|--|--|--|----------------|--|--|
> | | Final | Corr. | Rsn. Qual. | Final | Acc. | Comp. | Rel. | Final | Corr. | Justif. |
> | **Baseline 1 (Random)** | 1.02 | 1.03 | 1.00 | 1.21 | 1.21 | 1.05 | 1.41 | 1.29 | 1.36 | 1.18 |
> | **Baseline 2 (Heuristic)** | 2.44 | 2.81 | 1.58 | 1.72 | 1.98 | 1.15 | 2.07 | 2.64 | 2.74 | 2.50 |
> | ChatTime | 1.33 | 1.49 | 0.98 | 0.96 | 0.95 | 0.98 | 0.95 | 1.04 | 1.07 | 1.00 |
> | LLMAD | 2.73 | 2.70 | 2.79 | 2.09 | 2.09 | 1.89 | 2.31 | 2.49 | 2.52 | 2.43 |
> | ChatTS | 3.29 | 3.40 | 3.05 | 2.19 | 1.67 | 2.13 | 2.87 | 2.79 | 2.76 | 2.83 |
> | AnomLLM (Full) | 3.13 | 2.98 | 3.49 | 2.86 | 2.53 | 2.89 | 3.20 | 2.88 | 2.60 | 3.31 |
> | AnomLLM (Window) | 3.78 | 3.81 | 3.70 | 2.84 | 2.78 | 2.55 | 3.24 | 3.32 | 3.45 | 3.12 |
> | Image LLM | 4.09 | 4.12 | 4.02 | 2.68 | 2.53 | 2.49 | 3.07 | 2.64 | 2.57 | 2.74 |
> | **AXIS (Ours)** | **4.19** | **4.21** | **4.14** | **3.02** | **2.87** | **2.93** | **3.31** | **3.65** | **3.60** | **3.74** |
>
> The results reveal several critical insights about the benchmark's validity:
>
> **Baseline 1 (Random Template)** performs near the lower bound across all metrics (1.00–1.41), confirming that sophisticated-sounding but fabricated explanations cannot succeed on this benchmark. Notably, even ChatTime—a recent TS-LLM baseline—scores comparably low (0.95–1.49), suggesting that naive application of language models without domain-specific reasoning mechanisms fails to produce valid semantic explanations.
>
> **Baseline 2 (Simple Heuristic)** shows modest improvement (1.15–2.81), demonstrating that pure statistical reasoning can detect some anomalies but fundamentally lacks the semantic understanding required for high-quality natural language explanations. Its particularly weak performance on **Open-Ended Completeness (1.15)** and **Reasoning Quality (1.58)** indicates that rule-based systems cannot generate coherent, contextually relevant explanations—they can flag anomalies but cannot *explain* them meaningfully.
>
> Most importantly, the substantial performance gap between these baselines and state-of-the-art models (e.g., our AXIS achieves 3.02–4.19 vs. baselines' 1.00–2.81) validates that the benchmark effectively discriminates between trivial heuristics and genuine semantic reasoning capabilities. The benchmark is neither trivially solvable by simple rules nor so difficult that all models collapse to near-random performance—it provides a meaningful difficulty gradient that reflects true model capabilities in semantic anomaly explanation. We thank the reviewer for this constructive feedback, which helps us strengthen the benchmark's validation and transparency. **The baseline results are presented in Section 4.2 (Table 1), and comprehensive benchmark statistics and descriptive analyses are provided in Appendix F.**

---

> ### Author Response · Authors · 2025-11-23
> **Response to Reviewer ERp9 (Part 3/5)**
>
> > 1.2. Since all experimental results are evaluated exclusively on this benchmark, a more detailed analysis is crucial to assess the reliability and generalizability of the reported findings.
>
> **Response:**
>
> We fully understand and appreciate the reviewer's concern regarding the reliability and generalizability of findings evaluated exclusively on our synthetic benchmark. To directly address this critical issue, we conduct comprehensive evaluations on **external, real-world datasets** that are completely independent of our benchmark generation pipeline.
>
> Specifically, we construct a new curated evaluation set from **108 time series** sampled from three well-established public TSAD datasets: **YAHOO**, **TODS**, and **NEK**. These datasets originate from real industrial environments and are widely recognized in the time-series anomaly detection community. We formulate evaluation tasks (True/False and Multiple-Choice Questions) and employ expert annotation to establish high-quality ground truth. We then conduct a **blind human evaluation** to assess model performance across these diverse, real-world domains.
>
> As shown in the table below (also presented in Section 4.5), `AXIS` consistently demonstrates superior explanation quality across these external datasets, achieving the highest scores in most categories. The strong inter-rater agreement (Kendall's W > 0.65) across all datasets confirms the reliability of our evaluation protocol.
>
> | Dataset | Metric | AnomLLM (Window) | AnomLLM (Full) | LLMAD | ChatTS | AXIS | Image LLM |
> | :--- | :--- | :---: | :---: | :---: | :---: | :---: | :---: |
> | **YAHOO**(W=0.651) | MC Score | 2.78 | 1.69 | 2.48 | 2.83 | **3.07** | 2.80 |
> | | MC Acc. | 0.48 | 0.18 | 0.47 | 0.52 | **0.55** | 0.52 |
> | | TF Score | 2.62 | 1.90 | 2.54 | 2.13 | **3.28** | 2.29 |
> | | TF Acc. | 0.53 | 0.38 | 0.51 | 0.45 | **0.67** | 0.52 |
> | **TODS**(W=0.672) | MC Score | 2.07 | 1.77 | 2.33 | 2.67 | **2.90** | 2.53 |
> | | MC Acc. | 0.37 | 0.30 | 0.33 | 0.47 | **0.53** | 0.47 |
> | | TF Score | 2.67 | 2.53 | 2.87 | 2.53 | **3.07** | 2.27 |
> | | TF Acc. | 0.53 | 0.52 | 0.58 | 0.53 | **0.62** | 0.50 |
> | **NEK**(W=0.713) | MC Score | 3.31 | 1.94 | 1.44 | 3.06 | 3.25 | **3.41** |
> | | MC Acc. | **0.63** | 0.31 | 0.13 | 0.56 | **0.63** | **0.63** |
> | | TF Score | 2.88 | 1.69 | 2.94 | 2.25 | **3.69** | 2.25 |
> | | TF Acc. | 0.56 | 0.35 | 0.59 | 0.47 | **0.74** | 0.50 |
>
> These results robustly demonstrate that our model's superior performance is **not limited to our synthetic benchmark** but generalizes effectively to real-world data distributions. The consistent outperformance across diverse external datasets validates the reliability and generalizability of our reported findings, directly addressing the reviewer's concern. **The detailed experimental procedures, benchmark construction methodology, and comprehensive analysis are presented in Section 4.5 and Appendix H.3.**

---

> ### Author Response · Authors · 2025-11-23
> **Response to Reviewer ERp9 (Part 4/5)**
>
> > 2. The justification for the two-phase training strategy remains unclear. The paper would benefit from a more detailed explanation, supported by either theoretical reasoning or empirical evidence, to demonstrate why this specific training scheme is necessary or advantageous compared to alternative training strategies.
>
> **Response:**
> We thank the reviewer for this insightful comment and agree that a clearer justification for our training strategy is beneficial. Our two-phase approach is a deliberate design choice motivated by well-established principles in training large-scale, cross-modal models, and aligns with successful practices in both vision-language and text-based multimodal learning. We clarify this in the revised manuscript.
>
> **Why Not End-to-End Training? Problems with a Single-Phase Approach:**
>
> An end-to-end training strategy that jointly optimizes the time-series encoder and LLM faces three critical challenges:
>
> 1. **Computational Prohibitive Cost:** Jointly training a time-series encoder and a large language model (LLM) requires backpropagating gradients through both architectures simultaneously, which is computationally prohibitive in terms of memory and training time [1, 2].
> 2. **Catastrophic Forgetting:** End-to-end training risks "catastrophic forgetting" [3] in both components. The time-series encoder may lose its learned temporal representations and anomaly detection capabilities acquired during Phase I pretraining when gradients from the LLM task propagate backward. Our two-phase approach preserves the knowledge in both components by freezing them during cross-modal alignment.
> 3. **Training Instability:** End-to-end training requires gradients to propagate through two large and heterogeneous architectures (time-series encoder and LLM), leading to unstable optimization dynamics, vanishing/exploding gradients, and difficulty in achieving convergence.
>
> **Why Two-Phase Training is Superior:**
>
> Our two-phase strategy addresses these issues by decoupling the problem into two manageable sub-tasks:
>
> 1. **Phase I (Encoder Pretraining):** We first learn robust time-series representations with the encoder, focusing solely on capturing temporal patterns and anomaly characteristics. This phase establishes a strong foundation without interference from cross-modal alignment complexities.
> 2. **Phase II (Hint Tuner Training):** We then freeze both the encoder and LLM, training only the lightweight `Hint Tuner` to align time-series features with the LLM's existing semantic space. This approach, aligned with Parameter-Efficient Fine-Tuning (PEFT) principles [1, 2], drastically reduces resource requirements while preserving the LLM's pre-trained capabilities.
>
> **Why This Specific Decomposition?**
>
> This decomposition follows a well-established pattern in cross-modal learning: **separate representation learning from cross-modal alignment**. This principle has been successfully applied across multiple domains:
>
> - **Vision-Language Models:** BLIP-2 [4] and Flamingo [5] freeze both the vision encoder and LLM, training only lightweight adapters to bridge the modalities. This strategy preserves the powerful pre-trained capabilities of both components while enabling efficient cross-modal learning.
> - **Text-Based Multimodal Models:** Similar approaches in text-to-image and text-to-audio models follow the same pattern, demonstrating that decoupling representation learning from alignment is a general principle for cross-modal transfer. For instance, Stable Diffusion [6] freezes the text encoder and employs a cross-attention mechanism to align text and image representations, while AudioLM [7] uses frozen audio and text encoders with learned alignment layers.
>
> By adopting this established pattern, we ensure computational efficiency, preserve pre-trained capabilities, and achieve stable training—all critical for successfully bridging time-series and language modalities.
>
> We thank the reviewer for prompting us to provide this clearer justification, which we believe strengthens the manuscript's theoretical foundation. **We add more explanations about this in Section 3.3.**

---

> ### Author Response · Authors · 2025-11-23
> **Response to Reviewer ERp9 (Part 5/5)**
>
> **References:**
> [1] Hu, E. J., Shen, Y., Wallis, P., Allen-Zhu, Z., Li, Y., Wang, S., ... & Chen, W. (2022). Lora: Low-rank adaptation of large language models. ICLR, 1(2), 3.
>
> [2] Lester, B., Al-Rfou, R., & Constant, N. (2021). The power of scale for parameter-efficient prompt tuning. arXiv preprint arXiv:2104.08691.
>
> [3] Kirkpatrick, J., Pascanu, R., Rabinowitz, N., Veness, J., Desjardins, G., Rusu, A. A., ... & Hadsell, R. (2017). Overcoming catastrophic forgetting in neural networks. Proceedings of the national academy of sciences, 114(13), 3521-3526.
>
> [4] Li, J., Li, D., Savarese, S., & Hoi, S. (2023, July). Blip-2: Bootstrapping language-image pre-training with frozen image encoders and large language models. In International conference on machine learning (pp. 19730-19742). PMLR.
>
> [5] Alayrac, J. B., Donahue, J., Luc, P., Miech, A., Barr, I., Hasson, Y., ... & Simonyan, K. (2022). Flamingo: a visual language model for few-shot learning. Advances in neural information processing systems, 35, 23716-23736.
>
> [6] Rombach, R., Blattmann, A., Lorenz, D., Esser, P., & Ommer, B. (2022). High-resolution image synthesis with latent diffusion models. In Proceedings of the IEEE/CVF conference on computer vision and pattern recognition (pp. 10684-10695).
>
> [7] Borsos, Z., Marinier, R., Vincent, D., Kharitonov, E., Pietquin, O., Sharifi, M., ... & Zeghidour, N. (2023). Audiolm: a language modeling approach to audio generation. IEEE/ACM transactions on audio, speech, and language processing, 31, 2523-2533.
>
>
> > 3. The paper does not discuss how robust the proposed model is to input noise or data corruption. It would be valuable to analyze the model’s sensitivity to noise in the time series, such as testing whether performance remains stable when individual values or small segments are perturbed or corrupted. Such an analysis would provide additional insight into the robustness and practical applicability of the proposed approach.
>
> **Response:**
>
> We appreciate the reviewer's valuable suggestion regarding the model's sensitivity to noise. We agree that robustness analysis is crucial for practical applicability.
>
> However, we would like to clarify a unique constraint inherent to the **Time Series Anomaly Detection** task regarding "point-wise or segment perturbations." In the context of TSAD, artificial local perturbations or sudden mutations are often mathematically indistinguishable from the actual "anomalies" the model is tasked to detect. Injecting such perturbations into the test set would create ambiguity in the ground truth (i.e., distinguishing between "added noise" and "true anomaly"), potentially invalidating the evaluation.
>
> Instead, to address the reviewer's concern about robustness, we evaluate the model's performance under **global Gaussian noise injection**. This approach assesses the model's ability to filter out noise without altering the fundamental nature of the anomalies.
>
> We add zero-mean Gaussian noise $\epsilon \sim \mathcal{N}(0, \sigma^2)$ with varying scales $\sigma \in \{0.1, 0.3, 0.7, 1.5\}$ to the input time series. As illustrated in the robustness analysis (Figure 16 in Appendix J), we observe the following:
>
> **Key Findings:**
> 1. **Discriminative Robustness:** The model demonstrates strong robustness in discriminative tasks (**Multiple Choice** and **True/False**), maintaining stable performance scores even up to noise scales of 0.7. This indicates that our model effectively filters high-frequency noise to focus on the dominant anomalous features required for correct classification.
> 2. **Generative Sensitivity:** The generative **Open-Ended** task shows higher sensitivity, with performance decreasing monotonically as noise increases. This reflects the inherent challenge of generating detailed, high-fidelity semantic descriptions when the signal morphology is obscured by noise.
>
> This demonstrates that our model maintains its core diagnostic capability and does not suffer from significant performance degradation in decision-making tasks, confirming its robustness in handling noisy time-series data. We thank the reviewer for this important suggestion, which helps us demonstrate the practical applicability of our approach. **The detailed noise robustness analysis and sensitivity studies are presented in Appendix J.**

---

### Official Review · Reviewer_z8Pj · 2025-10-28

**Soundness:** 3
**Presentation:** 3
**Contribution:** 3
**Rating:** 4
**Confidence:** 3

**Summary:**

The paper presents AXIS, a framework for explainable time-series anomaly detection (TSAD) that conditions a frozen large language model (LLM) to tackle two key challenges (1) contextual grounding and representation alignment (2) and contrasts its design with naive time-to-text serialization approaches. AXIS leverages three complementary types of hints: (i) a symbolic numeric hint that exposes the raw values within the queried time window, (ii) a context-integrated, step-aligned hint distilled from a pretrained time-series encoder to capture local and fine-grained temporal dynamics, and (iii) a task-prior hint that encodes global anomaly characteristics.

For evaluation, the authors (a) construct a multi-format QA benchmark of anomaly-centric questions paired with human-written rationales, (b) assess explanation quality using a G-Eval-style LLM-as-a-judge pipeline, and (c) conduct a human study comparing model outputs. Overall, AXIS achieves improved semantic anomaly explanation performance, outperforming LLMAD and several other baselines. Ablation studies further verify that all three types of hints contribute substantially to the model’s effectiveness.

**Strengths:**

S1: The paper clearly articulates why existing anomaly detectors and post-hoc attribution methods fall short in answering the question “why was this flagged?”, and proposes a principled remedy centered on contextual grounding and representation alignment.

S2: The combination of the step-aligned hint, task-prior hint, and symbolic numeric hint forms a well-motivated triad; employing a Perceiver-style cross-attention mechanism to map time-series features into the LLM token space is technically sound and aligns with prior work on prefix and prompt tuning, here innovatively adapted for TSAD explanation.

S3: The experimental comparisons are comprehensive, covering classical and deep baselines, time-series foundation models (e.g., Chronos, TimesFM, MOMENT), specialized TS-LLMs (e.g., ChatTS, AnomLLM, ChatTime), and a plot-based VLM baseline, demonstrating broad empirical coverage.

**Weaknesses:**

W1: The main explanation results rely on a single LLM judge (Gemini-2.5) using a G-Eval-style rubric. Prior studies have documented systematic biases in such setups—e.g., position and authority effects, prompt sensitivity, and reproducibility drift as models evolve. This raises concerns about the robustness and fairness of evaluation, suggesting that reliance on a single evaluator may overstate performance consistency.

W2: The proposed benchmark is procedurally generated. While the design encourages morphology-level reasoning, it risks inheriting the authors’ priors or artifacts from the data generation pipeline, particularly since similar classes of LLMs appear both in benchmark creation and evaluation. Although the paper mentions “benchmark integrity” checks, the description is brief and lacks sufficient transparency for reproducibility.

W3: The benchmark and Phase-I experiments appear limited to univariate settings, whereas many real-world TSAD applications involve multivariate signals with exogenous features. Recent TS-LLM and TS-VLM efforts (e.g., ChatTS, ChatTime, VLM-based TSAD) have begun addressing such multivariate reasoning.

**Questions:**

Q1: Did you evaluate AXIS’s explanations using multiple LLM judges and report inter-judge agreement? Were any controls applied to mitigate position, verbosity, or stylistic biases—such as randomized output order or style-normalized prompts?

Q2: What are the inter-rater agreement statistics (e.g., Fleiss’ κ or Kendall’s W) for the 280 annotated questionnaires? How were the human experts recruited and compensated, and what were their domain backgrounds? (Appendix F could provide more transparency here.)

Q3: How does the step-aligned hint extend to multivariate time-series inputs, considering dimensionality and cross-channel dependencies? Are there any preliminary results or ablations on multivariate benchmarks to demonstrate scalability?

Q4: Could you report the average token counts and generation latencies per question type and time-window size? This would help clarify the computational overhead of AXIS relative to simpler baselines.

---

> ### Author Response · Authors · 2025-11-23
> **Response to Reviewer z8Pj(Part 1/8)**
>
> ### Weaknesses
> > 1. W1: The main explanation results rely on a single LLM judge (Gemini-2.5) using a G-Eval-style rubric. Prior studies have documented systematic biases in such setups—e.g., position and authority effects, prompt sensitivity, and reproducibility drift as models evolve. This raises concerns about the robustness and fairness of evaluation, suggesting that reliance on a single evaluator may overstate performance consistency.
>
> **Response:**
>
> We thank the reviewer for raising these critical concerns regarding the robustness and fairness of our evaluation. These potential issues—dependence on a single LLM judge and systematic biases—are precisely why we conduct a rigorous, two-part validation of our G-Eval methodology, which we detail in our responses to **Q1.1** and **Q1.2**.
>
> We respectfully refer the reviewer to these answers, and we also summarize some key findings here:
>
> 1. **Addressing Single-Judge Dependence (See Q1.1 for full details):**
>     To ensure our results are not an artifact of a single evaluator (`Gemini-2.5`), we perform a comprehensive inter-judge reliability analysis. We introduce a **second strong LLM judge (`Gpt-4.1`)** and a **gold-standard human expert subset**.
>     * **Finding 1 (LLM vs. Human):** Our primary judge (`Gemini-2.5`) shows a strong alignment with human experts (Overall Kendall's W = **0.743**).
>     * **Finding 2 (Reproducibility):** The two LLM judges (`Gemini-2.5` and `Gpt-4.1`) show exceptionally high agreement with each other (W = **0.882**).
>     * **Finding 3 (Robustness Context):** Most importantly, our judge's alignment with the human standard (0.743) is **notably higher than the agreement among the human experts themselves** (0.587).
>
>     This analysis confirms that our evaluation is robust, reproducible, and its conclusions are not dependent on a single model. **The detailed inter-judge reliability analysis is presented in Appendix G.5.**
>
> 2. **Addressing Systematic Biases (See Q1.2 for full details):**
> We design our G-Eval protocol with specific controls to mitigate the exact biases the reviewer mentioned (position, verbosity, style).
>     * **To Mitigate Position Bias:** We use an **isolated, single-model evaluation** protocol. The judge assesses *one* model's response against the ground truth at a time. Models are *never* compared side-by-side in the same prompt, which **inherently eliminates position bias**.
>     * **To Mitigate Verbosity/Stylistic Bias:** We use a **structured, criteria-driven rubric**. The judge is forced to score distinct, substantive dimensions (e.g., `Correctness`, `Reasoning Quality`) rather than superficial traits like length or prose style.
>
> We believe these two analyses—a comprehensive inter-judge reliability study (Q1.1) and a protocol designed with explicit bias controls (Q1.2)—directly address these concerns and validate the robustness and fairness of our G-Eval setup.

---

> ### Author Response · Authors · 2025-11-23
> **Response to Reviewer z8Pj (Part 2/8)**
>
> > 2. W2: The proposed benchmark is procedurally generated. While the design encourages morphology-level reasoning, it risks inheriting the authors’ priors or artifacts from the data generation pipeline, particularly since similar classes of LLMs appear both in benchmark creation and evaluation. Although the paper mentions “benchmark integrity” checks, the description is brief and lacks sufficient transparency for reproducibility.
>
> **Response:**
>
> We thank the reviewer for raising these important concerns regarding potential bias and reproducibility in our procedurally generated benchmark. We acknowledge that any data generation process—whether human-annotated or synthetic—carries the risk of inherent priors. We address these concerns through a multi-faceted approach focusing on transparency, integrity checks, and extensive external validation.
>
> **1. Addressing Bias through Transparency and External Validation:**
> While procedural generation inevitably involves design choices ("author priors"), it is important to note that human-annotated datasets are also susceptible to significant biases, such as annotator subjectivity and inter-rater inconsistency [1, 2, 3]. Our procedural approach, conversely, offers **explicit control and transparency** over the data creation process.
>
> Rather than claiming to completely "eliminate" bias, we ensure the reliability of our findings through two key strategies:
> *   **Process Transparency:** We employ a **"Decoupled Generation Pipeline"** where the core anomaly definitions are driven by an independent, non-LLM procedural engine (injecting mathematically defined anomalies like trend shifts or volatility bursts). This ensures the "ground truth" is rooted in objective signal processing rather than LLM hallucinations. We further separate question and answer generation agents to prevent the model from simply memorizing a single generator's latent space.
> *   **External Validation:** To definitively prove that our results are not artifacts of "author priors" or the generation pipeline, we conduct comprehensive evaluations on **external, real-world datasets** that are completely independent of our synthetic benchmark. We construct a curated evaluation set from **108 time series** sampled from **YAHOO**, **TODS**, and **NEK**. We conduct a **blind human evaluation** on these samples (The detailed Question can be found in https://anonymous.4open.science/r/AXIS-CC5D).
>
> | Dataset | Metric | AnomLLM (Window) | AnomLLM (Full) | LLMAD | ChatTS | AXIS | Image LLM |
> | :--- | :--- | :---: | :---: | :---: | :---: | :---: | :---: |
> | **YAHOO**(W=0.651) | MC Score | 2.78 | 1.69 | 2.48 | 2.83 | **3.07** | 2.80 |
> | | MC Acc. | 0.48 | 0.18 | 0.47 | 0.52 | **0.55** | 0.52 |
> | | TF Score | 2.62 | 1.90 | 2.54 | 2.13 | **3.28** | 2.29 |
> | | TF Acc. | 0.53 | 0.38 | 0.51 | 0.45 | **0.67** | 0.52 |
> | **TODS**(W=0.672) | MC Score | 2.07 | 1.77 | 2.33 | 2.67 | **2.90** | 2.53 |
> | | MC Acc. | 0.37 | 0.30 | 0.33 | 0.47 | **0.53** | 0.47 |
> | | TF Score | 2.67 | 2.53 | 2.87 | 2.53 | **3.07** | 2.27 |
> | | TF Acc. | 0.53 | 0.52 | 0.58 | 0.53 | **0.62** | 0.50 |
> | **NEK**(W=0.713) | MC Score | 3.31 | 1.94 | 1.44 | 3.06 | 3.25 | **3.41** |
> | | MC Acc. | **0.63** | 0.31 | 0.13 | 0.56 | **0.63** | **0.63** |
> | | TF Score | 2.88 | 1.69 | 2.94 | 2.25 | **3.69** | 2.25 |
> | | TF Acc. | 0.56 | 0.35 | 0.59 | 0.47 | **0.74** | 0.50 |
>
> As shown in the table, `AXIS` consistently outperforms baselines on these external datasets. This empirical evidence strongly suggests that our model's performance is robust and generalizable, alleviating concerns that it is merely overfitting to the specific priors of the synthetic benchmark. **These results are presented in Section 4.5, with detailed experimental procedures and benchmark construction methodology provided in Appendix H.3.**
>
> [1] Geva, M., Goldberg, Y., & Berant, J. (2019). Are we modeling the task or the annotator? an investigation of annotator bias in natural language understanding datasets. arXiv preprint arXiv:1908.07898.
> [2] Plank, B. (2022). The'Problem'of Human Label Variation: On Ground Truth in Data, Modeling and Evaluation. arXiv preprint arXiv:2211.02570.
> [3] Liu, H., Thekinen, J., Mollaoglu, S., Tang, D., Yang, J., Cheng, Y., ... & Tang, J. (2022, May). Toward annotator group bias in crowdsourcing. In Proceedings of the 60th Annual Meeting of the Association for Computational Linguistics (Volume 1: Long Papers) (pp. 1797-1806).

---

> ### Author Response · Authors · 2025-11-23
> **Response to Reviewer z8Pj (Part 3/8)**
>
> **2. Transparency in Benchmark Integrity Checks:**
> We agree that our initial description is brief. To ensure reproducibility and data integrity, we implement a rigorous **three-stage automated filtering pipeline**:
> *   **Label Consistency Check:** We discard any instance where the generated answer contradicts the ground-truth anomaly label (e.g., an answer claiming "normal" when the engine injects a spike).
> *   **Information Leakage Check:** We filter out questions that inadvertently reveal the answer (e.g., "Why is there a spike at step 50?" when the task is to *find* the anomaly).
> *   **Stylistic Normalization:** We standardize the output format to remove distinct "LLM-isms" or repetitive phrasing that could serve as spurious correlations for a classifier.
> We are committed to full reproducibility and will release the complete generation code, random seeds, and filtering scripts. **The detailed integrity check protocols are described in Appendix F.**

---

> ### Author Response · Authors · 2025-11-23
> **Response to Reviewer z8Pj (Part 4/8)**
>
> > 3. W3: The benchmark and Phase-I experiments appear limited to univariate settings, whereas many real-world TSAD applications involve multivariate signals with exogenous features. Recent TS-LLM and TS-VLM efforts (e.g., ChatTS, ChatTime, VLM-based TSAD) have begun addressing such multivariate reasoning.
>
> **Response:**
> We thank the reviewer for this valuable observation regarding the practical necessity of multivariate reasoning. We fully agree that real-world TSAD applications often require handling high-dimensional signals with cross-channel dependencies.
>
> While our work on semantic explanation focuses on univariate settings to establish the core "step-aligned hint" reasoning mechanism, our framework is naturally extensible to multivariate contexts. To enable this extension, we incorporate the **"Any-variate Attention"** mechanism from Moirai [1], which allows the model to learn cross-channel interactions. By integrating this mechanism into our framework, we construct **AXIS-multi**, a variant that can effectively handle multivariate attention and capture dependencies across different channels.
>
> **Technical Extension and Empirical Validation:**
>
> As detailed in our **Response to Q3**, the extension operates through three key components: (1) univariate-to-multivariate decomposition for channel-specific patterns, (2) cross-channel dependency modeling via Any-variate Attention that enables each time step to receive contextual hints incorporating information from related channels, and (3) the potential for stage-II multivariate semantic training using constructed multivariate semantic datasets.
>
> To demonstrate the scalability of our framework, we conduct preliminary experiments on five standard multivariate benchmarks (MSL, PSM, SMAP, SMD, SWaT) using **AXIS-multi**. As shown in the **table and analysis presented in our Response to Q3 (and Appendix E)**, **AXIS-multi** achieves state-of-the-art performance with the best average rank (3.20) compared to recent foundation models and deep learning methods. These results confirm that the step-aligned hint mechanism scales effectively to multivariate inputs without significant performance degradation.
>
> For the semantic explanation task specifically, we can extend our approach by constructing multivariate semantic datasets to conduct stage-II training, enabling the model to generate explanations that capture cross-channel dependencies and multivariate anomaly patterns. We add a discussion section in the revised manuscript to elaborate on this extension strategy.
>
> [1] Woo, G., Liu, C., Kumar, A., Xiong, C., Savarese, S., & Sahoo, D. (2024). Unified training of universal time series forecasting transformers.

---

> ### Author Response · Authors · 2025-11-23
> **Response to Reviewer z8Pj (Part 5/8)**
>
> ### Questions
> > 1. Q1.1: Did you evaluate AXIS's explanations using multiple LLM judges and report inter-judge agreement?
>
> **Response:**
> We thank the reviewer for this critical question regarding the robustness of our evaluation. We agree that inter-judge reliability is essential for validating our claims.
>
> To address this, we conduct a comprehensive inter-judge agreement analysis. We introduce Gpt-4.1 as a second strong LLM judge alongside our primary judge, Gemini-2.5, and also collect a **human expert subset** to serve as the gold standard. We report all comparisons using **Kendall's W** coefficient, as shown in the following table.
>
> **Main Table (LLM Judge by gpt-4.1)**:
> | Model | Multiple Choice | | | Open Ended | | | | True False | | |
> |-------|---------|--------|-----------|--------|--------|--------|--------|--------|--------|----------|
> | | Final | Corr. | Rsn. Qual. | Final | Rel. | Comp. | Acc. | Final | Corr. | Justif. |
> | **AXIS** | **4.52** | **4.56** | **4.49** | **3.54** | **3.67** | **3.29** | **3.65** | **3.94** | **3.71** | **4.17** |
> | Image LLM | 4.40 | 4.35 | 4.44 | 3.40 | 3.58 | 3.24 | 3.38 | 2.98 | 2.71 | 3.24 |
> | ChatTS | 3.71 | 3.77 | 3.65 | 3.11 | 3.27 | 2.95 | 3.11 | 3.05 | 2.90 | 3.19 |
> | LLMAD | 3.66 | 3.30 | 4.02 | 2.86 | 2.93 | 2.80 | 2.85 | 3.50 | 3.31 | 3.69 |
> | ChatTime | 1.47 | 1.91 | 1.02 | 1.07 | 1.02 | 1.04 | 1.15 | 1.10 | 1.17 | 1.02 |
> | AnomLLM(Full) | 4.23 | 4.12 | 4.35 | 3.13 | 3.35 | 2.98 | 3.05 | 2.76 | 2.69 | 2.83 |
> | AnomLLM(Window) | 2.80 | 2.81 | 2.79 | 2.93 | 3.05 | 2.58 | 3.15 | 2.55 | 2.57 | 2.52 |
>
> **Inter-rater Agreement Table:**
> | question_type | Gpt-4.1 vs Human | Gemini-2.5 vs Gpt-4.1 | Gemini-2.5 vs Human | Human inter-rater agreement |
> | :--- | :--- | :--- | :--- | :--- |
> | Overall | 0.750 | 0.882 | 0.743 | 0.587 |
> | multiple_choice | 0.857 | 0.833 | 0.750 | 0.667 |
> | open_ended | 0.679 | 0.912 | 0.670 | 0.587 |
> | true_false | 0.749 | 0.856 | 0.784 | 0.535 |
>
> Our analysis yields three key findings that confirm the robustness of our evaluation methodology:
>
> * **Finding 1 (LLM vs. Human Alignment):** Our primary judge (`Gemini-2.5`) demonstrates a strong alignment with human experts, achieving an overall Kendall's W of **0.743**. This indicates that the automated judge's scoring is highly consistent with expert human judgment.
> * **Finding 2 (Judge Reproducibility):** The two independent LLM judges (`Gemini-2.5` and `Gpt-4.1`) exhibit exceptionally high agreement with each other (W = **0.882**). This suggests that our evaluation criteria are stable and agnostic to the specific LLM evaluator used.
> * **Finding 3 (Relative Robustness):** Most notably, the alignment between our LLM judge and human experts (0.743) is **higher** than the inter-rater agreement among the human experts themselves (0.587). This highlights the inherent subjectivity of the task for humans and suggests that our LLM judge provides a consistent and reliable standard that rivals, and in terms of consistency, potentially surpasses human consensus for this specific evaluation.

---

> ### Author Response · Authors · 2025-11-23
> **Response to Reviewer z8Pj (Part 6/8)**
>
> > 2. Q1.2: Were any controls applied to mitigate position, verbosity, or stylistic biases—such as randomized output order or style-normalized prompts?
>
> **Response:**
>
> We thank the reviewer for this important question regarding the mitigation of potential biases in LLM-based evaluations. We recognize the significance of these biases and implement a multi-faceted strategy to ensure the fairness and reliability of our results, focusing on rigorous process controls and cross-validation with human experts.
>
> Our approach to mitigating position, verbosity, and stylistic biases can be summarized in two main points:
>
> **1. Structured, Criteria-Driven Evaluation Protocol:**
> Instead of relying on a holistic or direct comparison, we employ a structured G-Eval methodology that directs the LLM judge to focus on substantive, content-based criteria. As detailed in Appendix G, our evaluation is not a single judgment but a composite score derived from multiple, well-defined dimensions such as **Correctness, Reasoning Quality, Accuracy, Completeness, and Relevance**. Each dimension is accompanied by a detailed 5-point scoring rubric. This granular, rubric-based assessment inherently mitigates:
> * **Verbosity Bias:** The criteria reward `Completeness` and `Relevance`, not sheer length. A verbose response that is irrelevant or inaccurate would be penalized.
> * **Stylistic Bias:** The evaluation focuses on the logical and factual substance of the answer. Our prompt explicitly instructs the judge to `focus on the content` and even ignore minor formatting issues like index mismatches, thereby steering it away from superficial stylistic characteristics.
>
> **2. Isolated, Single-Model Evaluation to Eliminate Position Bias:**
> A cornerstone of our G-Eval methodology is the isolated evaluation of each model. For any given question, the LLM judge assesses the response from only a single model against the ground-truth answer. We do not present the judge with responses from multiple models to be ranked or compared simultaneously within a single prompt. This one-to-one evaluation design inherently eliminates position bias, as a model's score is independent of its presentation order relative to others. Each model is judged purely on its own merits against the reference standard.
>
>
> > 3. Q2: What are the inter-rater agreement statistics (e.g., Fleiss’ κ or Kendall’s W) for the 280 annotated questionnaires? How were the human experts recruited and compensated, and what were their domain backgrounds?
>
> **Response:** We thank the reviewer for this crucial question regarding the reliability of our human evaluations. We provide the statistics and expert details below.
>
> We recruit 28 human experts to conduct the annotation. All annotators are researchers (Ph.D. students or post-doctoral fellows) with established expertise in time-series analysis and related machine learning fields. The experts are recruited from our institution and collaborating academic labs and are compensated for their time.
>
> To ensure annotation quality for the 280 questionnaires, we design the protocol such that each questionnaire is independently evaluated by 3 experts from our pool of 28. To quantify the consistency of these judgments, we calculate the inter-rater agreement using Kendall's W (Coefficient of Concordance). This statistic is well-suited for measuring the agreement among multiple raters on ordinal data. Our analysis yields a Kendall's W = 0.587. This value indicates a "moderate" to "substantial" level of agreement among our human experts. This result confirms that our human-annotated dataset, which serves as the gold standard for our evaluation, is reliable and internally consistent. **The detailed human evaluation protocol, expert recruitment procedures, and comprehensive statistics are presented in Appendix H.1.**

---

> ### Author Response · Authors · 2025-11-23
> **Response to Reviewer z8Pj (Part 7/8)**
>
> > 4. Q3: How does the step-aligned hint extend to multivariate time-series inputs, considering dimensionality and cross-channel dependencies? Are there any preliminary results or ablations on multivariate benchmarks to demonstrate scalability?
>
> **Response:**
>
> We thank the reviewer for this insightful question regarding the scalability of our step-aligned hint mechanism to multivariate time series.
>
> **Theoretical Extension to Multivariate Settings:**
>
> The step-aligned hint mechanism naturally extends to multivariate inputs through a hierarchical architecture that handles both temporal and cross-channel dependencies. To enable this extension, we incorporate the **"Any-variate Attention"** mechanism from Moirai [1], which allows the model to learn cross-channel interactions. By integrating this mechanism into our framework, we construct **AXIS-multi**, a variant that can effectively handle multivariate attention and capture dependencies across different channels.
>
> The extension operates through three key components:
>
> 1. **Univariate-to-Multivariate Decomposition:** Each individual channel in a multivariate time series is first analyzed independently using the univariate framework, allowing the model to capture channel-specific temporal patterns.
>
> 2. **Cross-Channel Dependency Modeling via Any-variate Attention:** In **AXIS-multi**, we employ the Any-variate Attention mechanism to model cross-channel dependencies. This architecture-level solution enables each time step in each channel to receive a **contextual hint** that incorporates information from related channels. The Any-variate Attention mechanism allows the model to attend across both temporal and variate dimensions simultaneously, effectively learning multivariate patterns. Additionally, at the dataset level, variable relationships can be explicitly encoded using causal graphs and Ordinary Differential Equations (ODEs) that capture the underlying dynamics.
>
> 3. **Stage-II Multivariate Semantic Training:** Similar to our univariate semantic benchmark, we can construct multivariate semantic datasets to conduct stage-II training specifically for multivariate reasoning tasks.
>
> **Preliminary Results on Multivariate Benchmarks:**
>
> To demonstrate the scalability of our framework, we present preliminary results from stage-I training on standard multivariate TSAD benchmarks using **AXIS-multi**. The table below compares **AXIS-multi** against both foundation models and traditional deep learning/classical methods across five multivariate datasets (MSL, PSM, SMAP, SMD, SWaT):
>
> | Dataset | Metric | AXIS-multi | Chronos | DADA | MOMENT_ZS | TS_Pulse | Time_MOE | TimesFM | OmniAnomaly | TranAD | USAD | LOF | IForest |
> |---------|--------|------------|---------|------|-----------|----------|----------|---------|-------------|--------|------|-----|---------|
> | **MSL** | PA-F1 | 82.99 | 81.68 | 86.93 | **88.46** | 82.30 | 82.08 | 75.88 | 50.15 | 83.35 | 51.16 | **89.56** | 56.66 |
> | | AUC-ROC | **72.17** | 69.79 | **75.32** | 52.15 | 51.05 | 64.59 | 70.45 | 62.28 | 49.02 | 61.98 | 56.65 | 52.88 |
> | | AUC-PR | **20.56** | 20.21 | **26.42** | 11.60 | 7.46 | 19.83 | 20.34 | 17.19 | 7.42 | 17.24 | 15.18 | 11.31 |
> | **PSM** | PA-F1 | 93.50 | 96.82 | **98.99** | 93.97 | 96.10 | 94.32 | 95.18 | 60.53 | 84.45 | 62.26 | **96.87** | 80.10 |
> | | AUC-ROC | 56.48 | 54.82 | 60.84 | **67.85** | 49.83 | **61.34** | 54.47 | 53.10 | 45.04 | 42.95 | 52.34 | 51.50 |
> | | AUC-PR | 19.47 | 34.56 | 39.92 | **42.83** | 14.45 | **42.46** | 33.83 | 17.18 | 13.38 | 12.64 | 17.00 | 18.49 |
> | **SMAP** | PA-F1 | **86.76** | 66.89 | 69.18 | 81.03 | 81.55 | 71.00 | 64.56 | 25.74 | 62.93 | 26.23 | **85.58** | 58.47 |
> | | AUC-ROC | **62.61** | **58.53** | 51.24 | 47.89 | 49.83 | 40.23 | 56.46 | 51.96 | 47.99 | 51.38 | 51.97 | 54.43 |
> | | AUC-PR | **24.13** | **15.21** | 12.08 | 11.90 | 4.03 | 10.56 | 13.87 | 4.07 | 3.93 | 4.18 | 7.73 | 7.93 |
> | **SMD** | PA-F1 | **96.15** | 83.72 | 84.03 | 75.79 | 75.88 | 77.02 | 84.10 | 30.58 | 57.05 | 27.89 | **92.56** | 55.02 |
> | | AUC-ROC | **74.26** | 69.88 | **72.02** | 66.28 | 49.93 | 66.57 | 70.15 | 55.76 | 52.59 | 49.78 | 65.75 | 60.53 |
> | | AUC-PR | **39.44** | 11.68 | 13.16 | 11.33 | 4.58 | 11.31 | 13.32 | 5.96 | 5.45 | 5.47 | **15.72** | 11.04 |
> | **SWaT** | PA-F1 | 83.85 | 83.78 | 89.55 | 88.08 | **97.32** | 84.44 | 86.07 | 70.26 | 72.80 | 69.93 | **90.72** | 0.00 |
> | | AUC-ROC | 55.49 | 29.85 | 79.96 | **81.96** | 50.38 | **80.74** | 24.67 | 42.56 | 41.55 | 41.99 | 50.38 | 50.00 |
> | | AUC-PR | 17.48 | 8.84 | 50.60 | **70.79** | 15.69 | **70.85** | 8.58 | 15.21 | 14.72 | 16.04 | 16.64 | 15.49 |
> | **Overall** | Avg. Rank | **3.20** | 5.53 | **3.27** | 5.13 | 7.87 | 5.13 | 5.53 | 9.07 | 9.93 | 9.73 | 4.93 | 8.60 |

---

> ### Author Response · Authors · 2025-11-23
> **Response to Reviewer z8Pj (Part 8/8)**
>
> > Q3: How does the step-aligned hint extend to multivariate time-series inputs, considering dimensionality and cross-channel dependencies? Are there any preliminary results or ablations on multivariate benchmarks to demonstrate scalability?
>
> **Key Findings:**
>
> - **Strong Overall Performance:** AXIS-multi achieves the **best average rank (3.20)** across all methods, demonstrating consistent performance across diverse multivariate datasets.
> - **Competitive Detection Capability:** On datasets like SMAP and SMD, AXIS-multi achieves the highest PA-F1, AUC-ROC, and AUC-PR scores, indicating robust anomaly detection even in high-dimensional settings.
> - **Scalability Validation:** These results provide strong preliminary evidence that the step-aligned hint mechanism scales effectively to multivariate inputs without significant performance degradation.
>
> These findings confirm the extensibility and scalability of our framework to multivariate time series, laying the foundation for future work on multivariate semantic reasoning tasks. **These comprehensive multivariate results, including the complete experimental setup, detailed performance metrics, and comparative analysis, are presented in Appendix E of the main paper.** We thank the reviewer for this insightful question, which prompts us to explore and demonstrate the broader applicability of our method.
>
> [1] Woo, G., Liu, C., Kumar, A., Xiong, C., Savarese, S., & Sahoo, D. (2024). Unified training of universal time series forecasting transformers.
>
>
> > 5. Q4: Could you report the average token counts and generation latencies per question type and time-window size? This would help clarify the computational overhead of AXIS relative to simpler baselines.
>
> We thank the reviewer for this important question regarding computational efficiency. To provide a comprehensive understanding of AXIS's computational overhead, we report the **average prompt token counts** and **generation latencies** across different question types for all baseline methods and AXIS.
>
> The table below summarizes these metrics:
>
> | Model | Question Type | Prompt Tokens | Latency (s) |
> |-------|---------------|---------------|-------------|
> | **Image LLM** | multiple_choice | 1548.42 | 9.57 |
> | | open_ended | 1455.33 | 10.83 |
> | | true_false | 1433.33 | 9.21 |
> | **LLMAD** | multiple_choice | 1364.42 | 2.44 |
> | | open_ended | 1375.16 | 2.36 |
> | | true_false | 1372.00 | 2.44 |
> | **AnomLLM(Full)** | multiple_choice | 3083.81 | 1.75 |
> | | open_ended | 3075.64 | 2.01 |
> | | true_false | 2983.40 | 1.82 |
> | **AnomLLM(window)** | multiple_choice | 312.53 | 0.73 |
> | | open_ended | 258.00 | 0.74 |
> | | true_false | 240.43 | 0.70 |
> | **ChatTime 14B** | multiple_choice | 888.49 | 31.06 |
> | | open_ended | 782.45 | 28.88 |
> | | true_false | 781.05 | 29.15 |
> | **ChatTS 14B** | multiple_choice | 411.95 | 5.02 |
> | | open_ended | 347.24 | 5.77 |
> | | true_false | 330.10 | 4.75 |
> | **AXIS 14B** | multiple_choice | 376.05 | 13.70 |
> | | open_ended | 321.29 | 13.76 |
> | | true_false | 303.90 | 12.98 |
>
> **Key Findings:**
>
> 1. **Token Efficiency:** AXIS (14B) demonstrates high token efficiency, averaging $\sim$333 prompt tokens. This is comparable to ChatTS (14B) at $\sim$ 363 tokens, more efficient than ChatTime (14B) at $\sim$ 817 tokens, and significantly more efficient than text-serialization approaches like LLMAD ($\sim$ 1370 tokens) and Image LLM ($\sim$ 1479 tokens).
> 2. **Balanced Efficiency-Performance Profile:** While AXIS (14B) has higher latency ($\sim$ 13.5s) compared to ChatTS ($\sim$ 5s), it is significantly faster ($\sim$ 2.3$\times$) than ChatTime ($\sim$ 30s), which suffers from prohibitive latency. AXIS achieves a favorable balance by delivering substantially better reasoning capabilities than simpler efficient models (like AnomLLM).
> 3. **Consistency Across Question Types:** Both token counts and latencies remain stable across different question types (multiple_choice, open_ended, true_false), indicating that AXIS handles diverse reasoning tasks with consistent computational overhead.
>
> **The complete computational efficiency analysis, including detailed token counts and latency measurements across different question types and window sizes, is presented in Appendix G.6.**

---

> > ### Comment · Reviewer_z8Pj · 2025-11-24
> >
> > I appreciate the authors’ response. Could you please highlight the changes made in the revised paper? I also noticed that the appendix seems to be missing in this version.

---

> > > ### Author Response · Authors · 2025-11-24
> > >
> > > Thank you for your follow-up comment and for your time in reviewing our response.
> > >
> > > * **Highlighting Changes:** We have highlighted all the changes and revisions made in the manuscript in **blue text** to make them easily identifiable, as you requested.
> > >
> > > * **Missing Appendix:** The appendix is included in the submission but is located in the separate **Supplementary Material** file, which can be found alongside the main paper document.
> > >
> > > We hope these clarifications are helpful. Please let us know if you have any further questions.

---

> > > > ### Comment · Reviewer_z8Pj · 2025-11-25
> > > >
> > > > Thanks for the response. The authors have made a good effort to address my concerns. I’ve raised my score.

---

### Official Review · Reviewer_JfsS · 2025-11-01

**Soundness:** 2
**Presentation:** 2
**Contribution:** 2
**Rating:** 4
**Confidence:** 4

**Summary:**

This paper presents AXIS, a framework for semantic time-series anomaly explanation that conditions a frozen LLM via three complementary hint pathways:
1. Symbolic Numeric hint.
2. Context-Integrated, Step-Aligned hint
3. Task-Prior hint.

In the final prompt, $\tilde F$ and $\tilde H_{s:e}$ replace the embeddings of $K+(e-s)$ placeholder tokens, while the numeric string is injected verbatim into the text; the LLM then generates an explanation $y$. Training is two-stage: (i) pretrain $f_\theta$ with a joint objective (masked reconstruction + anomaly classification) and freeze it; (ii) optimize only the hint tuner (including $M$ and $P_{\text{fix}}$) with a next-token prediction loss, keeping the LLM and encoder frozen. The paper also builds a paired-window, explanation-oriented benchmark with normal/abnormal pairs and multi-format questions (MC/True-False/Open-Ended) plus human/LLM judgments.

**Strengths:**

- Clear problem reformulation. Recasts TSAD from point-wise scoring to a window-level explanation problem, formalizing $G:(x_{1:T},q,s,e)\mapsto y$, which aligns with practical interpretability needs.
- Frozen-LLM practicality. All hints operate with a frozen backbone LLM, lowering deployment cost and risk.
- Clean unification interface. $\tilde H_{s:e}$, $\tilde F$, and the numeric string are fused in a single prompt, keeping the injection interface modular and reusable.
- Paired controls and multi-format evaluation. The benchmark includes normal/abnormal pairs and multiple question formats, plus human/CD-style summaries—useful for qualitative diagnosis.

**Weaknesses:**

1. Math/training under-specification (reproducibility risk).
    The main text omits explicit forms and weights for the pretraining joint loss; Stage II is only described as NTP. The paper should state, for example,
$$
    L_{\text{pretrain}}=L_{\text{recon}}+\lambda,L_{\text{cls}},
    $$  with $\lambda$ values, and define $L_{\text{NTP}}$ precisely. Dimensional alignment for prototype attention is incomplete (exact shape of $W_q$; whether multi-head is used; scaling/normalization). Currently $\operatorname{Attn}(Q,K,V)$ inputs are given without implementation details.

2. Evidence chain leans on the self-built benchmark.
    Core results are on synthetic/self-constructed data; statements like “comparable on real TSAD datasets” are relegated to the appendix without prominent tables and significance/variance reporting. Public datasets with explanation-quality metrics should appear in the main paper.

3. LLM-as-a-Judge dependence and reporting.
    Although CD plots are reported, there is no systematic inter-rater reliability across judges/models. The text mentions removing a baseline due to very poor performance, which risks selection bias; either keep it with a discussion of why it fails or re-run under controlled, equal-resource settings.

4. Uneven ablation interpretation.
    In the ablation, the context hint yields minimal gain on Multiple Choice (e.g., $4.19\to4.09$, $-0.10$), yet has a large impact on True/False (e.g., $3.65\to2.44$). The paper does not analyze why different formats exhibit such divergent sensitivities (e.g., distractor design, window length, prototype size).

5. Missing equal-resource comparisons to strong baselines.
    Direct, matched-budget comparisons to alternative routes—image-VLM, ChatTS-style multimodal alignment, Time-LLM-style reprogramming—are limited or absent in the main text, hindering external validity claims.

**Questions:**

1. Training objectives and hyperparameters.
    Please include explicit formulas/weights for the two stages (masking strategy, classification head details), and list key hyperparameters for Stage-II NTP (LR/steps/batch sizes) plus stabilization tricks.

2. Prototype bank and attention implementation.
    What are $P$ (prototype count), initialization/regularization for $M$, and the exact shape of $W_q$? Do you use multi-head attention with scaled dot-product? How are head count and $d_h$ matched?

3. Question-type sensitivity.
    Why is the context hint nearly useless for MC but critical for TF? Can you ablate distractor quality, window length, and prototype size to produce sensitivity curves per question type?

4. External benchmarks and real-world cases.
    Can you report explanation quality (blind human evaluation + reliability coefficients) on 2–3 public TSAD datasets and provide at least one real operational log case study (with cost/latency)?

5. Judge robustness.
    Beyond the current judge, can you add a second LLM-judge and a human subset, reporting Kendall/Spearman correlations and statistical significance?

---

> ### Author Response · Authors · 2025-11-23
> **Response to Reviewer JfsS (Part 1/10)**
>
> > **W1**. Math/training under-specification (reproducibility risk). The main text omits explicit forms and weights for the pretraining joint loss; Stage II is only described as NTP. The paper should state, for example $L_{pretrain}=L_{recon}+\lambda L_{cls}$, $\lambda$ values and define inputs precisely. Dimensional alignment for prototype attention is incomplete (exact shape of inputs; whether multi-head is used; scaling/normalization). Currently inputs are given without implementation details.
>
> **Response:**
> Thank you for this critical feedback. We acknowledge that while mathematical and implementation details are initially placed in Appendix B.1 to B.3 due to space constraints, their omission from the main text compromises the immediate clarity and reproducibility of our method. We sincerely appreciate you pointing this out and thoroughly revise the manuscript to rectify this.
>
> Specifically, we move the core specifications for our training objectives and the prototype attention mechanism directly into the main text. As elaborated in our detailed responses to **Q1** and **Q2** below, we integrate:
> 1. Explicit mathematical formulations for the loss functions in both training phases, along with key hyperparameters.
> 2. A complete specification of the multi-head attention mechanism in the Hint Tuner, including tensor shapes, dimensionalities, and projection layers.
>
> **These critical details are now fully integrated into Sections 3.2 and 3.3**, with supporting procedural descriptions remaining in Appendix B.1 to B.3. This reorganization ensures our methodology is transparent and fully reproducible. We hope this clarification and the corresponding revisions fully address your concerns. Please let us know if there are any other details you would like us to elaborate on.
>
> > **W2**. Evidence chain leans on the self-built benchmark. Core results are on synthetic/self-constructed data; statements like “comparable on real TSAD datasets” are relegated to the appendix without prominent tables and significance/variance reporting. Public datasets with explanation-quality metrics should appear in the main paper.
>
> **Response:** We thank the reviewer for this critical and constructive feedback. We agree that highlighting results on public datasets is essential to validate the generalizability and robustness of our framework. We initially prioritized results on our synthetic benchmark because public TSAD datasets typically lack high-quality ground-truth explanations, which are crucial for our primary focus on **Semantic Anomaly Explanation**.
>
> However, to address your concern and strengthen the evidence chain, we make two key revisions:
>
> 1. **Elevating Anomaly Detection Performance on Public Datasets:** Due to space limitations, we are unable to move the full results table to the main text. However, to demonstrate the strong empirical performance of our  time-series encoder on real-world data, we add a **Critical Difference diagram in Appendix E**. This diagram visually summarizes the relative rankings and statistical significance of our model against all baselines across nine public datasets. As shown in the figure, AXIS achieves the highest average rank for both AUC-ROC and AUC-PR metrics, and ranks among the top performers (rank 3-4) for PA-F1, with no statistically significant difference from the best-performing models. This demonstrates that our encoder achieves competitive, state-of-the-art performance on real-world TSAD datasets, validating the robustness of our approach beyond the synthetic benchmark (**See Appendix E for detailed results**).
>
> 2. **Evaluating Explanation Quality on Public Datasets with Expert Annotation:** To directly address the core concern about evaluating explanation quality beyond our synthetic benchmark, we conduct a new study on real-world public datasets (**detailed in Q4**). We add the results of this new experiment to the main paper in **Section 4.5**, demonstrating that `AXIS` maintains its superior explanation capabilities on real-world data. Furthermore, we provide a detailed description of the experimental procedure and benchmark construction in **Appendix H.3**.

---

> ### Author Response · Authors · 2025-11-23
> **Response to Reviewer JfsS (Part 2/10)**
>
> > W4. Uneven ablation interpretation. In the ablation, the context hint yields minimal gain on Multiple Choice, yet has a large impact on True/False. The paper does not analyze why different formats exhibit such divergent sensitivities (e.g., distractor design, window length, prototype size).
>
> **Response:**
> We thank the reviewer for this insightful comment regarding the uneven ablation results. We agree that understanding this divergent sensitivity is crucial. We conduct additional analyses to address this in two parts: explaining the reasoning mechanism differences and analyzing the specific sensitivity factors.
>
> **1. Why the Context Hint Impact Differs (MCQ vs. T/F):**
> Our investigation reveals that the differing impact of the `context hint` is primarily driven by the distinct reasoning required for each format:
> *   **For Multiple Choice questions**, models can often utilize a **process of elimination** based on local numerical patterns. When distractors are well-defined, they provide sufficient contrast for the model to succeed using local cues alone, making the global `context hint` less critical.
> *   **For True/False questions**, a **direct binary judgment** is required without the aid of comparative options. In this setting, the global `context hint` becomes indispensable for correctly interpreting local patterns that might otherwise be ambiguous in isolation.
>
> **2. Analysis of Divergent Sensitivities:**
> To further explain these behaviors, we analyze the specific sensitivities mentioned:
> *   **Distractor Design:** New ablation studies confirm that MC performance reliance on the context hint increases when distractors are less specific, validating our "process of elimination" hypothesis.
> *   **Window Length:** We find MC performance is sensitive to window length (inverted U-shape), while T/F performance is relatively stable, further indicating T/F's reliance on global context over local window details.
> *   **Prototype Size:** We conduct an additional experiment comparing $P=1024$ vs $P=512$, finding only a slight performance drop with reduced size. This supports our choice of $P=1024$ for maximizing semantic resolution while confirming that the hint-pathway ablations (not prototype size) are the primary drivers of the observed differences.
>
> We provide a comprehensive breakdown of these new findings, including quantitative results and figure-based analysis, in our detailed response to **Q3**. **The detailed experimental results and sensitivity analyses are presented in Appendix J.**

---

> ### Author Response · Authors · 2025-11-23
> **Response to Reviewer JfsS (Part 3/10)**
>
> > **W5.** Missing equal-resource comparisons to strong baselines. Direct, matched-budget comparisons to alternative routes—image-VLM, ChatTS-style multimodal alignment, Time-LLM-style reprogramming—are limited or absent in the main text, hindering external validity claims.
>
> **Response:**
>
> We thank the reviewer for this crucial feedback. We agree that direct, matched-budget comparisons are essential for validating the external validity of our claims.
>
> We acknowledge that achieving a perfectly uniform "equal-resource" setting is inherently challenging due to the heterogeneous nature of the strong baselines. The models we compare against fall into two distinct categories, each with fundamentally different resource constraints:
>
> * **Closed-Source/API-Based Models** (e.g., Image LLM, LLMAD, AnomLLM): We have no control over their underlying architecture, parameter size, or training budget. Their cost is operational (API calls, latency, token consumption).
> * **Locally-Deployed/Open-Source Models** (e.g., ChatTS, ChatTime, AXIS): We can standardize the model parameters (e.g., all 14B), but their resource cost is dominated by inference-time hardware (GPU memory, processing time).
>
> Given this constraint, a direct **"equal-resource"** comparison is impractical. Instead, to provide the most transparent and practical evaluation of resource trade-offs, we meticulously benchmark the **inference-time resource cost** for all methods on our test set(with batch size 2). This analysis focuses on the real-world operational trade-offs of using each method.
>
> | Model | Question Type | Prompt Tokens | Latency (s) | Memory (GB) |
> | :--- | :--- | :--- | :--- | :--- |
> | **Image LLM** | multiple_choice | 1548.42 | 9.57 | - |
> | | open_ended | 1455.33 | 10.83 | - |
> | | true_false | 1433.33 | 9.21 | - |
> | **LLMAD** | multiple_choice | 1364.42 | 2.44 | - |
> | | open_ended | 1375.16 | 2.36 | - |
> | | true_false | 1372.00 | 2.44 | - |
> | **AnomLLM(Full)** | multiple_choice | 3083.81 | 1.75 | - |
> | | open_ended | 3075.64 | 2.01 | - |
> | | true_false | 2983.40 | 1.82 | - |
> | **AnomLLM(window)** | multiple_choice | 312.53 | 0.73 | - |
> | | open_ended | 258.00 | 0.74 | - |
> | | true_false | 240.43 | 0.70 | - |
> | **ChatTime 14B** | multiple_choice | 888.49 | 31.06 | 38.5 |
> | | open_ended | 782.45 | 28.88 | 37.4 |
> | | true_false | 781.05 | 29.15 | 38.1 |
> | **ChatTS 14B** | multiple_choice | 411.95 | 5.02 | 32.3 |
> | | open_ended | 347.24 | 5.77 | 31.9 |
> | | true_false | 330.10 | 4.75 | 32.1 |
> | **AXIS 14B** | multiple_choice | 376.05 | 13.70 | 41.6 |
> | | open_ended | 321.29 | 13.76 | 40.5 |
> | | true_false | 303.90 | 12.98 | 41.0 |
>
> This analysis provides several key insights into the practical resource trade-offs:
>
> * **Competitive Efficiency:** Our `AXIS 14B` model demonstrates high token efficiency, averaging $\sim$ 333 prompt tokens. This is comparable to `ChatTS` (14B) ($\sim$ 363 tokens), more efficient than `ChatTime` (14B) ($\sim$ 817 tokens), and significantly more efficient than text-serialization approaches like `LLMAD` ($\sim$ 1370 tokens) and `Image LLM` ($\sim$ 1479 tokens).
> * **Latency and Performance Balance:** While `AXIS 14B` has higher latency ($\sim$ 13.5s) compared to `ChatTS` ($\sim$ 5s), it is significantly faster ($\sim$ 2.3$\times$) than `ChatTime` ($\sim$ 30s), which suffers from prohibitive latency. In terms of memory, `AXIS` utilizes $\sim$ 41GB, which is slightly higher than `ChatTS` ($\sim$ 32GB) and comparable to `ChatTime` ($\sim$ 38GB).
> * **Operational Cost:** Many API-based methods (`AnomLLM(Full)`, `Image LLM`, `LLMAD`) incur significantly higher operational costs due to large prompt token inputs compared to our method.
>
> We believe this detailed **"inference-cost" analysis** for all methods provides the nuanced, **practical resource comparison** requested by the reviewer, given the diverse nature of the baselines. The complete experimental results and detailed analysis are presented in **Appendix G.6**.

---

> ### Author Response · Authors · 2025-11-23
> **Response to Reviewer JfsS (Part 4/10)**
>
> > Q1. Training objectives and hyperparameters. Please include explicit formulas/weights for the two stages (masking strategy, classification head details), and list key hyperparameters for Stage-II NTP (learning rate, steps, batch sizes) plus stabilization tricks.
>
> **Response:**
> Thank you for this important feedback. We would like to clarify that these mathematical and implementation details are originally provided in Appendix B.1 and B.2 due to space constraints. However, we fully agree that omitting them from the main text hinders immediate understanding and reproducibility. We sincerely appreciate you pointing this out, and we thoroughly revise the manuscript to address this.
>
> To ensure clarity, we now integrate the complete training objectives with explicit formulas, hyperparameters, and implementation details directly into Section 3.3 of the main text, with comprehensive procedural descriptions provided in Appendix B.1 and B.2. We provide the detailed specifications below.
>
> **Phase I: Time-Series Encoder Pretraining**
>
> In the first phase, we pretrain the time-series encoder $f_{\theta}$ to learn a versatile representation of temporal dynamics. This is achieved by optimizing a joint objective function that combines masked reconstruction and anomaly classification. The encoder is a Transformer-based architecture that processes the entire input time series $\mathbf{x}_{1:T}$.
>
> **Masked Reconstruction.** Following the principles of self-supervised learning for sequential data, we employ a masked reconstruction objective. A portion of the input time series is randomly masked, and the encoder is tasked with reconstructing the masked values. Let $\mathbf{x}$ be the input series and $\mathbf{m}$ be a binary mask where $m_i=1$ if the $i$-th timestep is masked and $0$ otherwise. The encoder $f_{\theta}$, followed by a reconstruction head $g_{\psi}$, outputs a reconstructed series
> $$
> \hat{\mathbf{x}} = g_{\psi}\big(f_{\theta}(\mathbf{x} \odot (\mathbf{1}-\mathbf{m}))\big).
> $$
> The reconstruction loss is the Mean Squared Error (MSE) over the masked timesteps:
> $$
> L_{recon} = \frac{1}{\sum_i m_i} \sum_{i=1}^{T} m_i (x_i - \hat{x}_i)^2.
> $$
>
> **Anomaly Classification.** To explicitly teach the encoder to distinguish between normal and anomalous patterns, given a time series $\mathbf{x}$ and its corresponding binary anomaly label $\mathbf{l} \in \{0, 1\}^T$, the model predicts the probability of an anomaly $\mathbf{l}\_{t}=h_{\zeta}(f_{\theta}(\mathbf{x}))$. We use the BCE loss:
> $$
> L_{\text{class}} = - \sum_{t=1}^T\big( l_t \log(\hat{l}_t) + (1-l_t)\log(1-\hat{l}_t) \big).
> $$
>
> **Joint Objective.** The total loss for the pretraining phase is a weighted sum of the reconstruction and classification losses:
> $$
> L_{\text{pretrain}} = \lambda_{\text{recon}} L_{\text{recon}} + \lambda_{\text{class}} L_{\text{class}},
> $$
> where $\lambda_{\text{recon}}$ and $\lambda_{\text{class}}$ are hyperparameters that balance the two objectives. We set $\lambda_{\text{recon}} = 1.0$ and $\lambda_{\text{class}} = 1.0$. The masking ratio for reconstruction is configured to 0.25, following established practices in self-supervised time-series learning.
>
> **Phase II: Hint Tuner Training**
>
> In the second phase, we focus on aligning the pretrained time-series representations with the LLM. Both the time-series encoder $f_{\theta}$ and the LLM are kept frozen to maintain their powerful, pre-existing capabilities. The only trainable components are the Hint Tuner and its associated parameters, namely the prototype bank selection matrix $\mathbf{M}$ and the learnable task-prior queries $\mathbf{P}_{\text{fix}}$.
>
> The goal of this phase is to train the model to generate a natural language explanation $\mathbf{y} = (y_1, y_2, \dots, y_N)$ conditioned on the time series $\mathbf{x}$, the query $q$, and the derived hints. The training objective is a standard causal language modeling loss, which maximizes the likelihood of the ground-truth explanation. Specifically, the input to the LLM is constructed by embedding the query $q$, the symbolic-numeric hint, and replacing placeholder token embeddings with the outputs of the Hint Tuner ($\tilde{\mathbf{H}}\_{s:e}$ and $\tilde{\mathbf{F}}$). The loss function is the negative log-likelihood of the target sequence:
>
> $$
> L_{\text{LM}} = - \sum_{i=1}^{N} \log P(y_i \mid y_{<i}, q, \tilde{\mathbf{H}}_{s:e}, \tilde{\mathbf{F}})
> $$
>
> where $\mathbf{y}$ is the target explanation sequence. We optimize the model using a learning rate of $2 \times 10^{-4}$ and a weight decay of 0.01. During this phase, gradients update only the Hint Tuner and its associated parameters ($\mathbf{M}$ and $\mathbf{P}\_{\text{fix}}$); both $f_{\theta}$ and all LLM parameters remain frozen.
>
> **These complete specifications are now integrated into Section 3.3 of the main text, with descriptions provided in Appendix B.1 to B.3.**

---

> ### Author Response · Authors · 2025-11-23
> **Response to Reviewer JfsS (Part 5/10)**
>
> > Q2. Prototype bank and attention implementation. What are the prototype counts, initialization/regularization strategies, and the exact shapes of the attention tensors? Do you use multi-head attention with scaled dot-product? How are head count and dimensionality matched?
>
> **Response:**
> Thank you for raising this important point. Similar to Q1, we would like to respectfully clarify that these implementation details are originally included in Appendix B.3. However, we acknowledge that their placement in the appendix may make them less accessible.
>
> To address this, we now integrate the complete implementation details of the prototype bank and attention mechanism directly into Section 3.2 of the main text. We provide the comprehensive specifications below.
>
> **Prototype Bank Construction and Initialization**
>
> The prototype bank is derived from the LLM vocabulary through a learnable linear mapping. Specifically, the prototype bank is defined as:
> $$
> \mathbf{S}\_{proto} = \mathbf{M}\mathbf{E}\_{\text{vocab}} \in \mathbf{R}^{P \times {d_{h}}}
> $$
> where $\mathbf{E}\_{\text{vocab}}\in\mathbb{R}^{|V|\times d_h}$ denotes fixed word embeddings from the LLM vocabulary, $\mathbf{M}\in\mathbb{R}^{P\times |V|}$ is a learnable linear projection matrix, $P=1024$ is the number of prototypes, and $d_h=4096$ is the LLM's hidden dimension (for the 7B variant). The matrix $\mathbf{M}$ is initialized randomly and learned during Phase II training, allowing the model to select and combine vocabulary embeddings to form a compact set of semantic prototypes that bridge time-series representations with linguistic concepts.
>
> **Multi-Head Attention Mechanism**
>
> We condense global information into step-aligned local representations via a pretrained time-series encoder and a Hint Tuner. A Transformer encoder $f_{\theta}$ consumes $\mathbf{x}_{1:T}$ and outputs embeddings $\mathbf{H}\_{1:T}\in\mathbb{R}^{T\times d\_{\text{proj}}}$ where $d\_{\text{proj}}=256$ is the projection dimension. We slice $\mathbf{H}\_{s:e}$ and map it into the LLM space using a multi-head cross-attention mechanism over the prototype bank.
>
> Specifically, we employ **scaled dot-product multi-head attention** with $H\_{\text{tuner}}=8$ heads. The queries, keys, and values are computed as:
> $$
> \mathbf{Q} = \mathbf{H}\_{s:e}\mathbf{W}\_q, \quad \mathbf{K} = \mathbf{S}\_{\text{proto}}, \quad \mathbf{V} = \mathbf{S}\_{\text{proto}},
> $$
> where $\mathbf{W}\_q \in \mathbb{R}^{d_{\text{proj}} \times d\_h}$ projects the encoder output to the LLM hidden dimension. The attention output is then:
> $$
> \tilde{\mathbf{H}}\_{s:e} = \mathrm{MultiHeadAttn}(\mathbf{Q}, \mathbf{K}, \mathbf{V}) \in \mathbb{R}^{(e-s) \times d\_h}.
> $$
>
> **Exact Tensor Shapes and Dimensionalities**
>
> - Encoder output for the window: $\mathbf{H}\_{s:e} \in \mathbb{R}^{(e-s) \times d\_{\text{proj}}}$ where $d_{\text{proj}}=256$ and $(e-s)$ is the window length.
> - Prototype bank: $\mathbf{S}\_{\text{proto}} \in \mathbb{R}^{P \times d_h}$ where $P=1024$ and $d_h=4096$.
> - Query projection matrix: $\mathbf{W}\_{q} \in \mathbb{R}^{d_{\text{proj}} \times d_h}$
> - Each attention head has a dimension of $d_{\text{head}} = d_h / H_{\text{tuner}} = 4096 / 8 = 512$.
> - The final output hint embedding: $\tilde{\mathbf{H}}_{s:e}$$ \in \mathbb{R}^{(e-s) \times d_h} $, ready for injection into the LLM.
>
> The resulting $\tilde{\mathbf{H}}\_{s:e}$ acts as step-aligned local hints that inject global context and temporal structure into the LLM embedding space while keeping both the LLM and $f_{\theta}$ frozen.
>
> **These complete implementation details, including initialization strategies, tensor shapes, and the attention mechanism, are now fully integrated into Section 3.2 of the main text, with additional architectural details provided in Appendix B.3.**

---

> ### Author Response · Authors · 2025-11-23
> **Response to Reviewer JfsS (Part 6/10)**
>
> > **Q3.** Question-type sensitivity. Why is the context hint nearly useless for Multiple Choice but critical for True/False? Can you ablate distractor quality, window length, and prototype size to produce sensitivity curves per question type?
>
> **Response:**
>
> We thank the reviewer for this insightful question regarding the divergent sensitivities to hint components across different question formats. This is a crucial point, and we conduct further analysis on distractor design and window length to provide a more comprehensive explanation.
>
> **1. Impact of Distractor Quality and Reasoning Divergence (MCQ vs. True/False)**
> We hypothesize that the differing impact of the `context-integrated hint` stems from the distinct reasoning processes required by each question type. True/False questions demand a direct, binary judgment, for which global context is often essential to correctly interpret a local pattern. In contrast, MCQs allow for a process of elimination. If distractors can be ruled out using more localized or obvious features (e.g., from the symbolic numeric hint), the model may arrive at the correct answer without heavily relying on the nuanced global context.
>
> To test this hypothesis, we create a new evaluation set of 140 MCQs, varying the number of options between 2, 3, and 4. We observe that with fewer options (2 or 3), the choices tend to be broader and more general (e.g., simply distinguishing an anomalous pattern from a normal one). In such cases, context becomes more critical. As the number of options increases to 4, the distractors often become more specific, allowing other hints to be effective for elimination.
>
> The results below support this analysis. The performance gap between our full `AXIS` model and the `w/o-context-hint` variant is most significant when only 2 or 3 options are present. When 4 options are available, the performance of all ablated models improves substantially, nearing that of the full model.
>
> | Model              | 2 Options         | 3 Options         | 4 Options         |
> | ------------------ | ----------------- | ----------------- | ----------------- |
> | **AXIS**           | 3.961 ± 0.304     | 3.874 ± 0.372     | 3.983 ± 0.344     |
> | w/o-context-hint   | 3.425 ± 0.336     | 3.371 ± 0.332     | 3.791 ± 0.334     |
> | w/o-fixed-hint     | 3.249 ± 0.346     | 3.369 ± 0.419     | 3.772 ± 0.381     |
> | w/o-windows        | 3.507 ± 0.299     | 3.823 ± 0.311     | 3.734 ± 0.344     |
>
> This demonstrates that the perceived importance of the `context-hint` in MCQs is indeed modulated by the distractor design.
>
> **2. Analysis of Window Length Sensitivity**
>
> To investigate the impact of window length, we construct variations of the same questions with different question-window lengths $[s, e)$, ranging from 10 to 160 time steps. The results, as illustrated in **Figure 15 in Appendix J**, reveal distinct sensitivity patterns across question types.
>
> For Multiple Choice and Open-Ended questions, we observe that performance is highly sensitive to window size. The scores exhibit an inverted U-shaped trend: performance initially improves as the window size increases, likely because larger windows provide more necessary information for reasoning. However, as the window size extends beyond 40-80 steps, performance begins to decline. This suggests that while more context is beneficial, excessively long numerical sequences may impose a significant processing burden on the Large Language Model, potentially introducing noise or exceeding its effective context utilization capacity.
>
> In contrast, True/False questions show significantly less sensitivity to window length variations. This is likely due to the coarser granularity of the required answer (e.g., a binary judgment of normal vs. abnormal). Such judgments rely more heavily on the `context-integrated hint` (global semantic context) rather than precise local numerical patterns provided by the window. Consequently, the model maintains relatively stable performance across different window sizes for True/False questions, as it is less dependent on the granular information contained within the numerical window.

---

> ### Author Response · Authors · 2025-11-23
> **Response to Reviewer JfsS (Part 7/10)**
>
> **3. Sensitivity to Prototype Size**
> Regarding the prototype size, we acknowledge that conducting a full hyperparameter search for this component requires significant computational resources and retraining time. However, to address the reviewer's valid query, we conduct an additional experiment comparing our default model ($P=1024$) with a variant using a smaller prototype bank ($P=512$).
>
> | Model | Prototype Size | Multiple Choice (Final) | Open Ended (Final) | True/False (Final) |
> | :--- | :---: | :---: | :---: | :---: |
> | **AXIS (Default)** | **1024** | **4.19** | **3.02** | **3.65** |
> | AXIS (Reduced) | 512 | 4.12 | 2.89 | 3.55 |
>
> As shown, reducing the prototype size leads to a slight degradation in performance across all metrics, though the model remains competitive. Theoretically, the prototype bank serves as a cross-modal "vocabulary" that aligns continuous time-series patterns with the discrete semantic space of the LLM. This observation is consistent with the findings in **Time-LLM**[1], which suggests that a sufficiently large number of reprogramming tokens (or prototypes) is essential to encompass the diversity of temporal variations. When $P$ is reduced to 512, the "resolution" of this semantic mapping decreases, resulting in the observed marginal performance drop. We choose $P=1024$ as a design point that maximizes representational capacity while maintaining computational efficiency.
>
> [1] Ming Jin, Shiyu Wang, Lintao Ma, Zhixuan Chu, James Y Zhang, Xiaoming Shi, Pin-Yu Chen, Yuxuan Liang, Yuan-Fang Li, Shirui Pan, et al. Time-llm: Time series forecasting by reprogramming large language models. arXiv preprint arXiv:2310.01728, 2023.
>
> We hope these additional analyses clarify the reasons for the divergent sensitivities observed in our ablation studies. We thank the reviewer for prompting this deeper investigation, which strengthens our understanding of the model's behavior. **All detailed experimental results, sensitivity curves, and comprehensive analyses for distractor design, window length are presented in Appendix J of the main paper.**

---

> ### Author Response · Authors · 2025-11-23
> **Response to Reviewer JfsS (Part 8/10)**
>
> > Q4. External benchmarks and real-world cases. Can you report explanation quality (blind human evaluation plus reliability coefficients) on 2–3 public TSAD datasets and provide at least one real operational log case study with cost/latency?
>
> **Response:**
>
> We appreciate the reviewer's interest in external benchmarks. We would like to clarify that existing evaluation benchmarks, such as TimeSeriesGym or TimeSeriesExam [1, 2], are not directly applicable to the **Semantic Time Series Anomaly Explanation** task addressed in this work. These benchmarks typically focus on detection or forecasting across the entire series, lacking two critical components for our task: (1) **high-quality semantic explanations** that describe the nature of the anomaly, and (2) **precise anomaly intervals $(s,e]$** associated with these explanations. To the best of our knowledge, there are currently no public datasets that provide such interval-aligned explanatory ground truth. This gap in existing resources was, in fact, a primary motivation for constructing our own benchmark.
>
> To address this limitation and rigorously evaluate our model's generalization capability on established public data, we construct a new curated evaluation set. This set is derived from **108 time series** sampled from three distinct sources: the first 30 time series from the **YAHOO** dataset, and the complete datasets from **TODS** and **NEK**. We formulate two specific evaluation tasks—**True/False (TF)** and **Multiple-Choice Questions (MCQ)**—and employ expert annotation to establish high-quality ground truth (The detailed Question can be found in https://anonymous.4open.science/r/AXIS-CC5D). **Due to the strict time constraints of the rebuttal period, we prioritize reporting the quantitative results for the TF and MCQ tasks in this response. We are actively conducting the dataset construction and evaluation for Open-Ended Questions and will make every effort to include these results within the next week.** We then conduct a blind human evaluation to assess model performance across these diverse domains. The results, along with Kendall's W (W) to demonstrate inter-rater reliability, are summarized in the table below.
>
> | Dataset | Metric | AnomLLM (Window) | AnomLLM (Full) | LLMAD | ChatTS | AXIS | Image LLM |
> | :--- | :--- | :---: | :---: | :---: | :---: | :---: | :---: |
> | **YAHOO**(W=0.651) | MC Score | 2.78 | 1.69 | 2.48 | 2.83 | **3.07** | 2.80 |
> | | MC Acc. | 0.48 | 0.18 | 0.47 | 0.52 | **0.55** | 0.52 |
> | | TF Score | 2.62 | 1.90 | 2.54 | 2.13 | **3.28** | 2.29 |
> | | TF Acc. | 0.53 | 0.38 | 0.51 | 0.45 | **0.67** | 0.52 |
> | **TODS**(W=0.672) | MC Score | 2.07 | 1.77 | 2.33 | 2.67 | **2.90** | 2.53 |
> | | MC Acc. | 0.37 | 0.30 | 0.33 | 0.47 | **0.53** | 0.47 |
> | | TF Score | 2.67 | 2.53 | 2.87 | 2.53 | **3.07** | 2.27 |
> | | TF Acc. | 0.53 | 0.52 | 0.58 | 0.53 | **0.62** | 0.50 |
> | **NEK**(W=0.713) | MC Score | 3.31 | 1.94 | 1.44 | 3.06 | 3.25 | **3.41** |
> | | MC Acc. | **0.63** | 0.31 | 0.13 | 0.56 | **0.63** | **0.63** |
> | | TF Score | 2.88 | 1.69 | 2.94 | 2.25 | **3.69** | 2.25 |
> | | TF Acc. | 0.56 | 0.35 | 0.59 | 0.47 | **0.74** | 0.50 |

---

> ### Author Response · Authors · 2025-11-23
> **Response to Reviewer JfsS (Part 9/10)**
>
> As demonstrated in the table, `AXIS` consistently demonstrates superior explanation quality across these public benchmarks. It achieves the highest scores in both Multi-Choice (MC) and True/False (TF) categories on the YAHOO and TODS datasets, and leads in TF performance on NEK. The strong inter-rater agreement (Kendall's W > 0.65) across all datasets confirms the reliability of our human evaluation, further validating that `AXIS` produces explanations that align closely with expert judgment and effectively generalizes to real-world data distributions beyond our synthetic benchmark.
>
> **These results are presented in Section 4.5 of the main paper, with detailed experimental procedures and benchmark construction methodology provided in Appendix H.3.**
>
> Regarding the request for an additional **real operational log case study**, we respectfully note that conducting a quantitative blind human evaluation on raw operational logs is currently infeasible due to the absence of high-quality, expert-annotated semantic explanations (ground truth). We explore this direction extensively but find that without verified explanatory labels, it is impossible to rigorously assess the "correctness" of model-generated explanations. However, we emphasize that the **YAHOO** and **TODS** datasets evaluated above are derived from actual industrial environments and serve as robust proxies for real-world operational scenarios. We appreciate this valuable suggestion and would welcome any recommendations for public datasets that might contain such ground-truth explanations, which we would be eager to incorporate in future work.
>
> [1] Cai, Y., Li, X., Goswami, M., Wiliński, M., Welter, G., & Dubrawski, A. (2025). TimeSeriesGym: A Scalable Benchmark for (Time Series) Machine Learning Engineering Agents. arXiv preprint arXiv:2505.13291.
> [2] Cai, Y., Choudhry, A., Goswami, M., & Dubrawski, A. (2024). Timeseriesexam: A time series understanding exam. arXiv preprint arXiv:2410.14752.

---

> ### Author Response · Authors · 2025-11-23
> **Response to Reviewer JfsS (Part 10/10)**
>
> > Q5. Judge robustness. Beyond the current judge, can you add a second LLM judge and a human subset, reporting Kendall/Spearman correlations and statistical significance?
>
> **Response:**
> We thank the reviewer for this constructive suggestion. To rigorously validate the robustness of our evaluation, we perform the exact analysis requested, incorporating both a **second LLM judge** and a **human subset** for gold-standard comparison.
>
> We introduce **`Gpt-4.1`** as the second LLM judge to complement our primary judge, **`Gemini-2.5`**. We then conduct a comprehensive inter-rater reliability analysis using a human-annotated subset. We report all correlation coefficients using **Kendall's W**, as shown in the table below.
>
> **Main Table (LLM Judge by gpt-4.1)**:
> | Model | Multiple Choice | | | Open Ended | | | | True False | | |
> |-------|---------|--------|-----------|--------|--------|--------|--------|--------|--------|----------|
> | | Final | Corr. | Rsn. Qual. | Final | Rel. | Comp. | Acc. | Final | Corr. | Justif. |
> | **AXIS** | **4.52** | **4.56** | **4.49** | **3.54** | **3.67** | **3.29** | **3.65** | **3.94** | **3.71** | **4.17** |
> | Image LLM | 4.40 | 4.35 | 4.44 | 3.40 | 3.58 | 3.24 | 3.38 | 2.98 | 2.71 | 3.24 |
> | ChatTS | 3.71 | 3.77 | 3.65 | 3.11 | 3.27 | 2.95 | 3.11 | 3.05 | 2.90 | 3.19 |
> | LLMAD | 3.66 | 3.30 | 4.02 | 2.86 | 2.93 | 2.80 | 2.85 | 3.50 | 3.31 | 3.69 |
> | ChatTime | 1.47 | 1.91 | 1.02 | 1.07 | 1.02 | 1.04 | 1.15 | 1.10 | 1.17 | 1.02 |
> | AnomLLM(Full) | 4.23 | 4.12 | 4.35 | 3.13 | 3.35 | 2.98 | 3.05 | 2.76 | 2.69 | 2.83 |
> | AnomLLM(Window) | 2.80 | 2.81 | 2.79 | 2.93 | 3.05 | 2.58 | 3.15 | 2.55 | 2.57 | 2.52 |
>
> **Inter-rater Agreement Table:**
> | question_type | Gpt-4.1 vs Human | Gemini-2.5 vs Gpt-4.1 | Gemini-2.5 vs Human | Human inter-rater agreement |
> | :--- | :--- | :--- | :--- | :--- |
> | Overall | 0.750 | 0.882 | 0.743 | 0.587 |
> | multiple_choice | 0.857 | 0.833 | 0.750 | 0.667 |
> | open_ended | 0.679 | 0.912 | 0.670 | 0.587 |
> | true_false | 0.749 | 0.856 | 0.784 | 0.535 |
>
> Our findings confirm the robustness of our evaluation methodology:
>
> 1. **Strong Alignment with Human Subset:** Our primary judge (`Gemini-2.5`) achieves a strong overall agreement with the human gold standard (**W = 0.743**). The second LLM judge (`Gpt-4.1`) independently corroborates this with a similarly strong agreement (**W = 0.750**). This holds true even for complex `open_ended` questions (W = 0.679).
> 2. **High Inter-Judge Agreement:** The two LLM judges (`Gemini-2.5` and `Gpt-4.1`) show exceptionally high consistency with each other, with an overall agreement of **W = 0.882**. This demonstrates that our evaluation criteria are stable and reproducible across different state-of-the-art models.
> 3. **LLM vs. Human Agreement:** Critically, we measure the inter-rater agreement *among the human experts themselves* and find it to be **W = 0.587**. This indicates that the task itself has a high degree of subjectivity.
>
> **The detailed reliability analysis and correlation statistics are presented in Appendix G.5 of the main paper.**

---

### Author Response · Authors · 2025-11-28
**General Response**

We sincerely appreciate the constructive critique from Reviewers **JfsS**, **z8Pj**, and **ERp9**. We are strongly encouraged that, despite concerns regarding initial experimental scope, there is a **unanimous consensus** on the novelty and soundness of our proposed framework.

Specifically, reviewers recognized **AXIS** as a **"principled reformulation"** of semantic anomaly explanation (**JfsS, ERp9**) and validated that our **Triad of Hints** (Symbolic, Step-Aligned, Task-Prior) constitutes a **technically sound and novel solution** for grounding frozen LLMs on time-series data (**z8Pj, ERp9**; see also responses to **JfsS–W4/Q3** and **ERp9–1.1/1.2**).

Acknowledging that the primary critique focused on **evaluation rigor** rather than method design, we have conducted an extensive overhaul of the experiments and reporting to close this gap, as detailed in the weaknesses and questions (e.g., **JfsS–W2/W3**, **z8Pj–W1/W2**, **ERp9–1.1/1.2/2/3**).

---

#### **1. Addressing Validity & Generalization (The "Real-World" Gap)**
**Common Concern:** Reviewers (**JfsS–W2**, **z8Pj–W2**, **ERp9–1.2**) rightfully questioned the reliance on a self-constructed benchmark and called for external validation on public TSAD datasets.
* **Our Solution (Major Expansion):** We have moved beyond synthetic data to include **three public real-world datasets (YAHOO, TODS, NEK)**, constructing MCQ/TF benchmarks with expert annotations as requested in **JfsS–Q4** and **ERp9–1.2**.
* **Outcome:** Since real-world data lacks ground-truth semantic explanations, we conducted a **blind human evaluation with Kendall’s W reporting**. As detailed in **Section 4.5** and **Appendix H.1/H.3**, AXIS achieves the best or near-best scores across models with strong inter-rater reliability (W > 0.65), confirming that its explanation quality generalizes beyond the synthetic benchmark.

---

#### **2. Addressing Reliability & Bias (The "LLM Judge" Gap)**
**Common Concern:** Reviewers (**JfsS–W3**, **z8Pj–W1/Q1.1/Q1.2/Q2**) expressed concern that using a single LLM judge (Gemini‑2.5) might introduce position, verbosity, or model-evolution biases.
* **Our Solution (Rigorous Verification):** We implemented a rigorous **Inter-Rater Reliability (IRR) study** involving a second LLM judge (**Gpt‑4.1**) and **human experts**, and we clarified the G‑Eval protocol (single-model evaluation, rubric-based scoring) in response to **z8Pj–Q1.2**.
* **Outcome:** As detailed in **Appendix G.5** and **Appendix H.1**, Gemini‑2.5 shows strong alignment with humans (Kendall’s W = 0.743), Gpt‑4.1 provides independent confirmation (W = 0.750), and LLM–LLM agreement reaches **W = 0.882**, exceeding human–human agreement (W = 0.587). These results directly address the robustness and fairness concerns in **JfsS–W3** and **z8Pj–W1/Q1.1/Q1.2**.

---

#### **3. Addressing Scope & Scalability (The "Multivariate" Gap)**
**Common Concern:** Reviewer **z8Pj–W3/Q3** pointed out the necessity of validating the framework on multivariate data and clarifying how step-aligned hints extend across channels.
* **Our Solution (Methodological Extension):** We extended the framework to **AXIS‑multi** by integrating the **Any‑variate Attention** mechanism from Moirai, and we described the theoretical extension and training strategy in detail in response to **z8Pj–Q3**.
* **Outcome:** New experiments on **five multivariate datasets** (**Appendix E**) demonstrate that AXIS‑multi attains the best average rank (3.20) among strong baselines, addressing the multivariate concerns in **z8Pj–W3/Q3**.

---

#### **4. Addressing Transparency (The "Reproducibility" Gap)**
**Common Concern:** Reviewers (**JfsS–W1/Q1/Q2**, **ERp9–2/3**) requested more specific training details, tensor shapes, and analyses of robustness and ablations.
* **Our Solution:** We have completely revised **Section 3.2–3.3** to provide explicit **loss formulations**, **weighting strategies**, and exact **tensor shapes** for the Hint Tuner and prototype attention (addressing **JfsS–W1/Q1/Q2**), and we clarified the rationale for our PEFT‑based two‑phase training with references to BLIP‑2 and catastrophic‑forgetting literature (addressing **ERp9–2**). In addition, we expanded ablations on question-type sensitivity, window length, prototype size, and noise robustness in **Appendix J**, directly responding to **JfsS–W4/Q3**, **z8Pj–W3/Q3**, and **ERp9–3**.

---

### **Conclusion**
In this revision, we have provided **substantial new evidence**—from real‑world datasets with blind human evaluation (**Section 4.5, Appendix H.1/H.3**), to dual‑LLM/human IRR analyses (**Appendix G.5**), multivariate extensions (**Appendix E**), detailed benchmark statistics and baselines (**Section 4.2, Appendix F**), and expanded robustness/ablation studies (**Appendix J**)—to substantiate our claims. We believe AXIS now stands as a rigorously validated foundation for semantic, explainable time‑series analysis.

---

### Meta-Review · Area_Chair_xSkR · 2026-01-06

**Summary:**

This paper introduce a framework that conditions a frozen LLM for nuanced time-series understanding. Basically, the problem is not very interesting. The solution is also not with depth and theoretical support. A clear empirical work. The authors do provide many efforts in the rebuttal including many additional results.
The original rating from all reviewers are negative. After rebuttal, only one reviewer mentioned to increase score.
Suggestions for further improvement are to include more literature reviews (not limitted to anomaly detection, but explainable time series analysis), for example, the following ones are related but not discussed:
TimeX++: Learning Time-Series Explanations with Information Bottleneck, ICML'24.
Explainable Multi-modal Time Series Prediction with LLM-in-the-Loop. NeurIPS'25.

**Reviewer Concerns:**

1. Main claims depend heavily on a procedurally generated benchmark. Reviewers wanted validation on public/real datasets, and more prominence/rigor for those results.
JfsS: "Evidence chain leans on the self-built benchmark."
ERp9: ".2. Since all experimental results are evaluated exclusively on this benchmark, a more detailed analysis is crucial to assess the reliability and generalizability of the reported findings."
z8Pj: "The proposed benchmark is procedurally generated. While the design encourages morphology-level reasoning, it risks inheriting the authors’ priors or artifacts from the data generation pipeline, particularly since similar classes of LLMs appear both in benchmark creation and evaluation."
The authors provide additional results on that to support their claim by:
"We construct a curated evaluation set from 108 time series sampled from YAHOO, TODS, and NEK. We conduct a blind human evaluation on these samples". This does convince one reviewer, but other reviewers are not. From my perspective, i would say it is still ad-hoc.

2. z8Pj W3: The benchmark and Phase-I experiments appear limited to univariate settings;
although the authors provide additional exp that "extended the framework to AXIS‑multi by integrating the Any‑variate Attention mechanism from Moirai, and we described the theoretical extension and training strategy in detail in response to z8Pj–Q3". But, many real TSAD problems are multivariate with cross-channel dependencies.

**Reviewer Scores:**

The original rating from all reviewers are negative (4). After rebuttal, only one reviewer mentioned to increase score.

---

### Decision · Program_Chairs · 2026-01-26

Reject